# Collateral deletion of the mitochondrial AAA+ ATPase ATAD1 sensitizes cancer cells to proteasome dysfunction

Jacob M Winter[1], Heidi L Fresenius[2], Corey N Cunningham[1], Peng Wei[1], Heather R Keys[3], Jordan Berg[1], Alex Bott[1], Tarun Yadav[1], Jeremy Ryan[4], Deepika Sirohi[5], Sheryl R Tripp[5], Paige Barta[1], Neeraj Agarwal[6], Anthony Letai[4], David M Sabatini[7], Matthew L Wohlever[2], Jared Rutter[1,6,8]*

[1]Department of Biochemistry, University of Utah, Salt Lake City, United States; [2]Department of Chemistry & Biochemistry, University of Toledo, Toledo, United States; [3]Whitehead Institute for Biomedical Research, Cambridge, United States; [4]Dana-Farber Cancer Institute, Harvard Medical School, Boston, United States; [5]University of Utah and ARUP Laboratories, Salt Lake City, United States; [6]Huntsman Cancer Institute, University of Utah, Salt Lake City, United States; [7]Department of Biology, Massachusetts Institute of Technology, Cambridge, United States; [8]Howard Hughes Medical Institute, Salt Lake City, United States

*For correspondence: rutter@biochem.utah.edu

**Abstract** The tumor suppressor gene *PTEN* is the second most commonly deleted gene in cancer. Such deletions often include portions of the chromosome 10q23 locus beyond the bounds of *PTEN* itself, which frequently disrupts adjacent genes. Coincidental loss of *PTEN*-adjacent genes might impose vulnerabilities that could either affect patient outcome basally or be exploited therapeutically. Here, we describe how the loss of *ATAD1*, which is adjacent to and frequently co-deleted with *PTEN*, predisposes cancer cells to apoptosis triggered by proteasome dysfunction and correlates with improved survival in cancer patients. ATAD1 directly and specifically extracts the pro-apoptotic protein BIM from mitochondria to inactivate it. Cultured cells and mouse xenografts lacking ATAD1 are hypersensitive to clinically used proteasome inhibitors, which activate BIM and trigger apoptosis. This work furthers our understanding of mitochondrial protein homeostasis and could lead to new therapeutic options for the hundreds of thousands of cancer patients who have tumors with chromosome 10q23 deletion.

## Editor's evaluation

The authors identify co-deletion of the mitochondrial AAA+ ATPase ATAD1 with the tumor suppressor PTEN as a factor modifying cancer prognosis, based on a new mechanism of increasing sensitivity to proteotoxic stress induced by proteasome inhibition. The authors also identify the mitochondrial E3 ubiquitin ligase MARCH5 as a gene whose deletion is synthetically lethal with ATAD1. These findings suggest that the use of proteasome-targeting agents may be useful in patients with tumors dually deleted for ATAD1 and PTEN. The study is based on convincing evidence and makes an innovative contribution to the understanding of the biology of tumors with 10q23 deletions.

## Introduction

The tumor suppressor gene *PTEN* is deleted in more than 33% of metastatic prostate tumors and nearly 10% of glioblastoma multiforme and melanoma (*Worby and Dixon, 2014*). These deletions are

**eLife digest** Cancer cells have often lost genetic sequences that control when and how cell division takes place. Deleting these genes, however, is not an exact art, and neighboring sequences regularly get removed in the process. For example, the loss of the tumor suppressor gene *PTEN*, the second most deleted gene in cancer, frequently involves the removal of the nearby *ATAD1* gene. While hundreds of thousands of human tumors completely lack *ATAD1*, individuals born without a functional version of this gene do not survive past early childhood. How can tumor cells cope without *ATAD1* – and could these coping strategies become the target for new therapies?

Winter et al. aimed to answer these questions by examining a variety of cancer cells lacking *ATAD1* in the laboratory. Under normal circumstances, the enzyme that this gene codes for sits at the surface of mitochondria, the cellular compartments essential for energy production. There, it extracts any faulty, defective proteins that may otherwise cause havoc and endanger mitochondrial health. Experiments revealed that without ATAD1, cancer cells started to rely more heavily on an alternative mechanism to remove harmful proteins: the process centers on MARCH5, an enzyme which tags molecules that require removal so the cell can recycle them. Drugs that block the pathway involving MARCH5 already exist, but they have so far been employed to treat other types of tumors. Winter et al. showed that using these compounds led to the death of cancerous *ATAD1*-deficient cells, including in human tumors grown in mice.

Overall, this work demonstrates that cancer cells which have lost *ATAD1* become more vulnerable to disruptions in the protein removal pathway mediated by MARCH5, including via already existing drugs. If confirmed by further translational work, these findings could have important clinical impact given how frequently *PTEN* and *ATAD1* are lost together in cancer.

imprecise and typically include *PTEN* as well as neighboring genes at the 10q23 locus. Prior research has demonstrated that the 'collateral' deletion of neighboring genes along with tumor suppressor genes can generate context-dependent vulnerabilities specific to tumor cells (*Muller et al., 2015*; *Kryukov et al., 2016*; *Mavrakis et al., 2016*). In some cases, such vulnerabilities can be exploited in a way that is toxic to mutant cancer cells but not to genetically intact host cells, which represents a therapeutic opportunity. Whether genomic deletions involving *PTEN* generate targetable vulnerabilities through the loss of neighboring genes is unknown, but would have relevance to a significant proportion of cancer patients given the frequency at which these deletions occur.

Only 40 kb upstream of *PTEN* is *ATAD1*, which encodes a AAA+ ATPase involved in protein homeostasis on the outer mitochondrial membrane (OMM) (*Chen et al., 2014*; *Okreglak and Walter, 2014*; *Nakai et al., 1993*; *Wang and Walter, 2020*; *Zhang et al., 2011*). ATAD1 hydrolyzes ATP to directly remove substrate proteins from the OMM (*Wang and Walter, 2020*; *Wang et al., 2022*). ATAD1 appears particularly suited to extract tail-anchored (TA) proteins, which harbor a C-terminal, single-pass transmembrane domain. In contexts unrelated to cancer, it has been shown that the absence of ATAD1 leads to the accumulation of TA proteins on the OMM and significant mitochondrial dysfunction (*Chen et al., 2014*). This housekeeping role of ATAD1 is important for cellular health, as evidenced by the findings that *ATAD1* is essential for life in mammals and has been conserved over the 1 billion years of evolution separating yeast and humans (*Zhang et al., 2011*; *Ahrens-Nicklas et al., 2017*). Here, we describe how the collateral deletion of *ATAD1* along with *PTEN* sensitizes cells to apoptosis induced by dysfunction of the ubiquitin proteasome system.

## Results

### ATAD1 and PTEN are co-deleted in many human cancers

Because the *PTEN* and *ATAD1* genes are adjacent on human Chr10q23.31 (*Figure 1A*; *Poluri and Audet-Walsh, 2018*), we assessed whether *ATAD1* is co-deleted with *PTEN* using immunohistochemistry on prostate adenocarcinoma tumors (*Chung et al., 2019*). We analyzed tumors that were *PTEN*-null by targeted sequencing, along with *PTEN*-wild-type (WT) controls. ATAD1 protein was undetectable in 21 of the 37 *PTEN*-null tumors analyzed, but was present in all 15 PTEN-WT control tumors (*Figure 1—figure supplement 1A-C*). Analysis of genomic data from The Cancer Genome

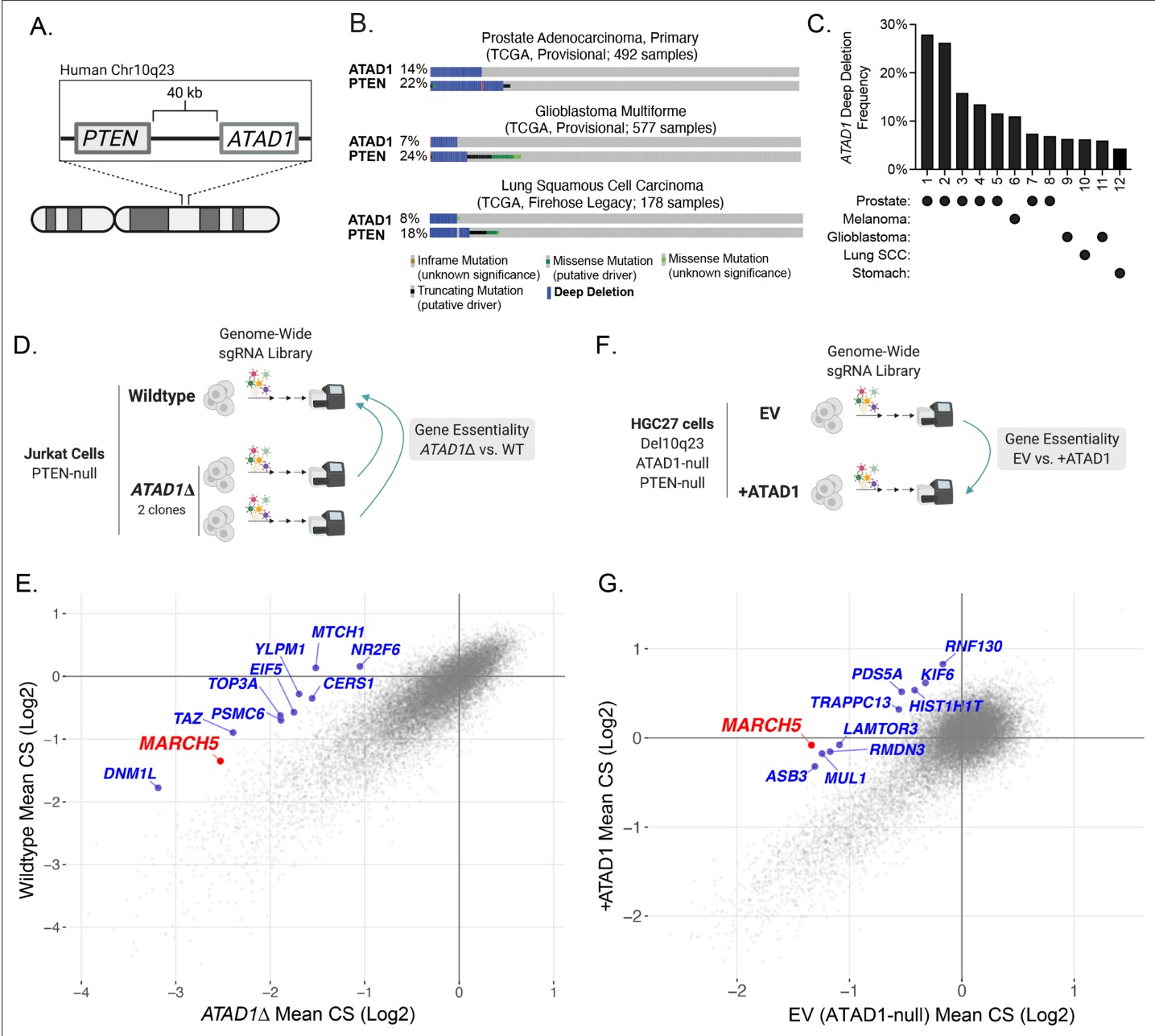

**Figure 1.** *ATAD1* is co-deleted with *PTEN* in cancer and its loss confers synthetic lethal vulnerabilities. (**A**) Schematic of *PTEN* and *ATAD1* loci. (**B**) Oncoprint plots from three TCGA studies of cancer. *ATAD1* and *PTEN* alteration frequencies are shown, with blue bars indicating deep deletions. (**C**) Frequency of *ATAD1* deep deletions across various cancer types; data from cBioPortal. (**D**) CRISPR screen design for wild-type (WT) and *ATAD1Δ* Jurkat cells. (**E**) Jurkat CRISPR screen results; each point represents one gene. CRISPR score (CS) values were calculated by taking the average log$_2$ fold-change in relative abundance of all sgRNAs targeting a given gene over 14 population doublings. WT CS values are shown on the y-axis. The CS values per gene for each of the two *ATAD1Δ* clones were averaged and are plotted on the x-axis. The top 10 genes that were differentially essential between WT and *ATAD1Δ* are labeled in blue, with *MARCH5* labeled in red. (**F**) CRISPR screen design for HGC27 cells (Chr10q23 deletion, *ATAD1*-null) comparing gene essentiality in *ATAD1* complemented cells or empty vector (EV) (*ATAD1*-null) control. (**G**) HGC27 CRISPR screen results; CS values are as described for (**E**). The x-axis depicts CS for the *ATAD1*-null condition of EV-transduced cells, and the y-axis depicts CS for the *ATAD1*-complemented (+ATAD1) condition. Labels are as described for (**E**).

The online version of this article includes the following source data and figure supplement(s) for figure 1:

**Source data 1.** Source data used to make *Figure 1*.

**Figure supplement 1.** *ATAD1* is co-deleted with PTEN as a passenger.

**Figure supplement 1—source data 1.** Source data used to make *Figure 1—figure supplement 1*.

*Figure 1 continued on next page*

**Figure supplement 2.** Characterization of co-deleted genes on Chr10q23.

**Figure supplement 3.** Supporting data for Jurkat CRISPR screens.

**Figure supplement 4.** Estimated number of deaths worldwide by cancer type.

Atlas corroborated these protein-level findings, as the majority of tumors harboring deep deletions in *PTEN* also had deep deletions in *ATAD1* (*Figure 1B*). Importantly, *ATAD1* is almost never deleted in the absence of *PTEN* deletion (*Figure 1B*), nor does it feature recurrent inactivating point mutations (*Figure 1—figure supplement 1D,E*), which argues that ATAD1 is not a tumor suppressor. Therefore, we hypothesize that *ATAD1* deletion is simply a 'hitchhiker' with the oncogenic driver deletion of *PTEN*. These deletions most frequently span the 0.5 Mb surrounding *PTEN*, leading to collateral deletion of *KLLN* and *RNLS* in addition to *ATAD1* (*Figure 1—figure supplement 2*). Altogether, *ATAD1* is deleted at a high frequency across many tumor types, including in more than 25% of prostate cancer, 11% of melanoma, 7% of glioblastoma, and 4% of gastric adenocarcinoma (*Figure 1C*). Given the established role of ATAD1 in mitochondrial protein homeostasis, we hypothesized that a hitchhiker deletion of *ATAD1* might confer unique vulnerabilities on tumors (*Muller et al., 2015*; *Kryukov et al., 2016*; *Mavrakis et al., 2016*; *Muller et al., 2012*).

## ATAD1 *is synthetic lethal with* MARCH5

We conducted genome-wide CRISPR knockout screens to identify genes that are selectively essential in *ATAD1Δ* cells. Such genes represent pathways whose inhibition could be selectively toxic to *ATAD1*-deficient tumor cells in a patient. We generated two *ATAD1Δ* clones in the *PTEN*-null Jurkat T-cell acute lymphoblastic lymphoma cell line using transient expression of Cas9 and one of two sgRNAs targeting distinct exons of *ATAD1* (*Figure 1—figure supplement 3A and B*). Jurkat cells were chosen as an experimentally tractable system that has been validated in genetic screening and as a cell line with background *PTEN* deficiency. *ATAD1* deletion did not affect basal proliferation rate (*Figure 1—figure supplement 3C*). We conducted parallel screens on the WT Jurkat parental cell line and each of the two *ATAD1Δ* clonal cell lines, the comparison of which enabled us to minimize idiosyncrasies inherent to clonal cell lines (*Figure 1D*). A CRISPR score (CS) is assigned to each gene, and represents the mean log$_2$ fold-change in relative abundance of sgRNAs targeting that gene. CS values for the two *ATAD1Δ* clones were averaged and compared against those of the WT cells (*Figure 1E*). Genes that are selectively essential in the *ATAD1Δ* background represent *ATAD1* synthetic lethal candidate genes. As expected, the differential CRISPR score (dCS) values for each *ATAD1Δ* clone vs. WT significantly correlated with each other (p=2.16 × 10$^{-16}$; *Figure 1—figure supplement 3D*). The top 10 candidates for *ATAD1* synthetic lethality include five genes that encode mitochondrial proteins (*MARCH5*, *TAZ*, *MTCH1*, *TOP3A*, *DNM1L*) and two components of the ubiquitin proteasome system (*MARCH5*, *PSMC6*), which are both processes with clear relevance to the known functions of ATAD1 (*Calvo et al., 2016*; *Figure 1E*). *MARCH5* (also known as MITOL) was a particularly interesting hit, since it is an E3 ubiquitin ligase that promotes protein degradation on the OMM (*Nakamura et al., 2006*).

Many properties of cell lines can affect their particular genetic dependencies, including tissue of origin, driver, and passenger mutations, and even the media in which they grow (*Hart et al., 2015*; *Rossiter et al., 2021*). Therefore, we conducted an additional genetic screen in a different cellular context to gain a broader perspective on how *ATAD1* deficiency creates synthetic lethal vulnerabilities. Rather than screening another pair of *ATAD1*-WT and engineered knockout cells, we sought to use cells that naturally have the Chr10q23 deletion seen in human tumors. Given the lack of available prostate cancer cell lines that harbor Del(10q23), we used a gastric adenocarcinoma cell line, HGC27, which is *ATAD1* and *PTEN* deficient. *ATAD1* is deleted in 4.1% of gastric cancer, a disease that causes nearly 800,000 deaths worldwide each year (*Ferlay et al., 2019*), therefore, we estimate that 32,000 patients die every year from ATAD1-deficient gastric cancer. This ranks second only to prostate cancer in terms of the number of deaths attributable to ATAD1-deficient tumors (*Figure 1—figure supplement 4*).

We made ATAD1-proficient and -deficient HGC27 lines by transducing with lentiviral ATAD1-FLAG or empty vector (EV), and subsequently conducted genome-wide CRISPR screens. In this case, genes

that are selectively essential for EV cells (ATAD1-deficient) represent putative *ATAD1* synthetic lethal candidate genes (*Figure 1F*). The top candidate for synthetic lethality with *ATAD1* was *MARCH5* (*Figure 1G*), a gene also identified in the Jurkat screen described above. Deletion of *MARCH5* had a negligible effect on fitness in ATAD1-proficient HGC27 cells, but was an essential gene (by MAGeCK) in the EV cells (ATAD1-deficient). A second OMM-localized ubiquitin E3 ligase, *MUL1*, was also a top hit for synthetic lethality with ATAD1. In summary, our two complementary and unbiased genetic screens indicate that dysfunction of the ubiquitin proteasome system is preferentially lethal to cells lacking *ATAD1*.

We were particularly intrigued by the interaction of *ATAD1* and *MARCH5* for two main reasons. First, the same synthetic lethal interaction between *ATAD1* and *MARCH5* emerged as the top hit of CRISPR screens using two vastly different cellular contexts. Second, *MARCH5* encodes a ubiquitin E3 ligase that ubiquitinates OMM proteins to trigger their extraction by p97/VCP and subsequent degradation by the proteasome (*Nakamura et al., 2006*; *Cherok et al., 2017*). MARCH5/p97 and ATAD1 mediate two parallel pathways by which OMM proteins are removed from mitochondria. Hence, it was intuitive that *ATAD1* and *MARCH5* could be synthetic lethal, given that they both contribute to protein homeostasis on the OMM, and synthetic lethal interactions classically involve two redundant pathways.

## An imbalance of BCL2 family proteins underlies the synthetic lethality of *ATAD1* and *MARCH5*

It has recently become clear that the key function of MARCH5 is to suppress apoptosis (*Djajawi et al., 2020*; *Haschka et al., 2020*; *Subramanian et al., 2016*; *Arai et al., 2020*). Apoptosis is regulated by OMM-localized BCL2 family proteins and requires the permeabilization of the OMM by BAX/BAK (*Kale et al., 2018*). Pro-survival proteins such as MCL1 bind to and inhibit BAX/BAK to prevent inappropriate cell death (*Greaves et al., 2019*). A variety of stressors activate BH3-only proteins (e.g. BIM), which trigger apoptosis by binding and inhibiting pro-survival proteins like MCL1 and in some cases by directly activating BAX/BAK (*Letai, 2017*). BH3-only proteins serve as sentinels for cellular stress and, upon activation, initiate mitochondrial outer membrane permeabilization (*Bhatt et al., 2020*; *Llambi et al., 2011*).

MARCH5 acts as a 'guardian' of MCL1 through an incompletely understood mechanism that involves the degradation of the pro-apoptotic BH3-only proteins BIM and/or NOXA (*Kale et al., 2018*; *Letai, 2017*; *Llambi et al., 2011*; *Czabotar et al., 2014*; *Lin et al., 2022*). We hypothesized that ATAD1 antagonizes these OMM-localized pro-apoptotic factors in parallel to MARCH5, such that simultaneous loss of both ATAD1 and MARCH5 leads to a lethal accumulation of pro-apoptotic proteins on the OMM. Indeed, BIM structurally resembles known substrates of ATAD1/Msp1 in that it is TA, has an intrinsically disordered region N-terminal to the transmembrane domain, and has basic residues at the extreme C-terminus (*Castanzo et al., 2020*; *Li et al., 2019*). Consistent with this hypothesis, the abundance of $BIM_{EL}$ (the predominant isoform of BIM) was increased in *ATAD1Δ* cells (*Figure 2A*). $BIM_{EL}$ can also be inactivated by phosphorylation by cytosolic kinases such as ERK. Deletion of *ATAD1* decreased $BIM_{EL}$ phosphorylation, as assessed by decreased mobility in SDS-PAGE (*Figure 2B*), and with phospho-specific antibodies for residues Ser69 and Ser77 (*Figure 2—figure supplement 1*). While ATAD1 activity promoted the inhibitory phosphorylation of BIM at Ser69 and Ser77, it did not affect BIM phosphorylation at Thr112, a phosphorylation site that potentiates the pro-death activity of BIM (*Figure 2—figure supplement 1*). These data suggested that ATAD1 might act on BIM to promote its degradation and inhibitory phosphorylation.

Since MARCH5 regulates the BIM/NOXA/MCL1 axis, we assessed abundance of these proteins in the context of single and double deletion of *ATAD1* and *MARCH5*. Deletion of *ATAD1* increased BIM levels, while deletion of *MARCH5* increased NOXA levels (*Figure 2C*). Accordingly, deletion of both *MARCH5* and *ATAD1* increased the abundance of both NOXA and BIM, which work together to antagonize the pro-survival protein MCL1 (*Figure 2C*). We hypothesized that synergistic antagonism of MCL1 explained, at least in part, the synthetic lethality of *ATAD1* and *MARCH5*.

We tested whether BIM was required for the synthetic lethal interaction of *ATAD1* and *MARCH5*. Deletion of *MARCH5* caused a modest decrease in viability in WT Jurkat cells (*Figure 2D*). This effect was similar when we deleted *MARCH5* in polyclonal BIM (*BCL2L11*) knockout cells (*Figure 2D*; generated by stably expressing Cas9 and sgRNA targeting *BCL2L11*). *ATAD1Δ* cells, however, were

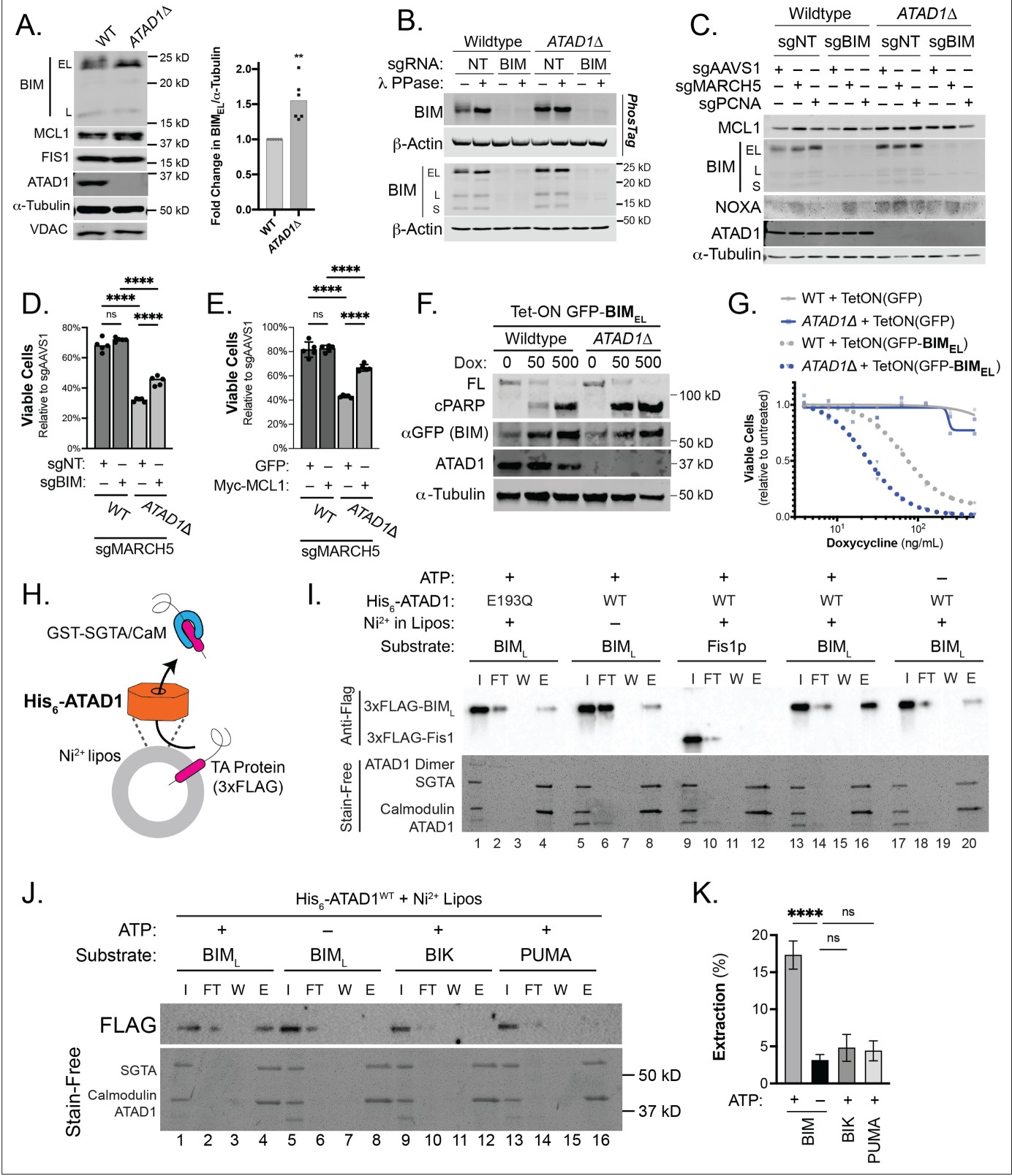

**Figure 2.** *ATAD1/MARCH5* synthetic lethality is partially mediated by BIM, which is a novel ATAD1 substrate. (**A**) Western blot of Jurkat cell lines, with quantification of $BIM_{EL}$ levels normalized to alpha-tubulin; one sample t and Wilcoxon test. (**B**) Western blot of whole cell lysates from wild-type (WT) or *ATAD1Δ* Jurkat cells stably expressing sgNT or sgBIM with Cas9-T2A-GFP. Lysates were mock treated or treated with lambda phosphatase ($\lambda$ PPase) and analyzed by PhosTag/SDS-PAGE. (**C**) Western blots of Jurkat cell lines stably expressing Cas9-T2A-GFP with sgNT or sgBIM, harvested 4 days after

*Figure 2 continued on next page*

*Figure 2 continued*

transduction with additional indicated sgRNAs. (**D**) Viability of Jurkat cells after deletion of *MARCH5*, using different genetic backgrounds. Viability at 4 days post-transduction was normalized to that of cells transduced with sgRNA targeting *AAVS1*. Data analyzed by two-way ANOVA with Tukey's multiple comparisons. (**E**) Viability of Jurkat cells stably expressing GFP or Myc-tagged MCL1 after deletion of *MARCH5* and normalized as in (**D**). (**F**) Viability of Jurkat cells transduced with tetracycline-inducible GFP or GFP-BIM$_{EL}$ fusion; t=48 hr, normalized to viability of cells without doxycycline. (**G**) Western blot of cell lines as described in (**D**), treated with doxycycline (Dox) for 24 hr. (**H**) Schematic of in vitro extraction assay; 'Ni$^{2+}$ lipos' indicates the use of nickel chelating headgroups of lipids in the liposomes; the star symbolizes a GST tag on the soluble chaperones, calmodulin (CaM) and SGTA, which are included to catch extracted TA substrates. (**I**) Extraction assay using His-ATAD1 and 3xFLAG-BIM$_L$ (lanes 1–8, 13–20) or the negative control yeast TA protein, 3xFLAG-Fis1p (lanes 9–12); E193Q indicates the use of a catalytically inactive mutant of ATAD1; in samples shown in lanes 5–8, Ni$^{2+}$ chelating lipids were omitted; in samples shown in lanes 17–20, ATP was omitted; 'I'=Input, 'FT'=flow-through, 'W'=final wash, 'E'=elution. Eluted fractions represent TA proteins extracted by ATAD1 and bound by GST-tagged chaperones; compare elution 'E' to input 'I'. (**J**) Extraction assay as described in (**H**) but comparing different BH3-only proteins, BIM, BIK, and PUMA. (**K**) Quantification of assays as shown in (**I**), n=6 independent experiments.

The online version of this article includes the following source data and figure supplement(s) for figure 2:

**Source data 1.** Source data and uncropped blots used to make *Figure 2*.

**Figure supplement 1.** BIM phosphorylation in Jurkat cells.

**Figure supplement 1—source data 1.** Source data used to make *Figure 2—figure supplement 1*.

**Figure supplement 2.** Synthetic lethality of *ATAD1/MARCH5* partially depends on BIM and can be suppressed by MCL1.

**Figure supplement 2—source data 1.** Source data used to make *Figure 2—figure supplement 2*.

**Figure supplement 3.** Functional and physical interaction of ATAD1 and BIM.

**Figure supplement 3—source data 1.** Source data used to make *Figure 2—figure supplement 3*.

**Figure supplement 4.** Validation of proteoliposome extraction assay.

**Figure supplement 4—source data 1.** Source data used to make *Figure 2—figure supplement 4*.

**Figure supplement 5.** Comparing intrinsically disordered regions of BIM$_L$, BIK, and PUMA.

**Figure supplement 6.** ATAD1 promotes non-mitochondrial localization of GFP-tagged BIM$_{EL}$ΔBH3.

**Figure supplement 7.** GFP-BIM puncta do not colocalize with mitochondria labeled with mito-mCherry.

**Figure supplement 8.** GFP-BIM puncta do not colocalize with peroxisomes.

**Figure supplement 9.** GFP-BIM puncta do not colocalize with lysotracker blue.

hypersensitive to *MARCH5* deletion, as expected from our CRISPR screen results. Interestingly, BIM knockout partially rescued the hypersensitivity of *ATAD1Δ* cells to *MARCH5* deletion (*Figure 2D*). These results validate the major finding from our CRISPR screens and demonstrate that BIM is partially responsible for the synthetic lethality of *ATAD1* and *MARCH5*.

If MCL1 antagonism underlies *ATAD1/MARCH5* synthetic lethality, then we reasoned that overexpression of MCL1 might rescue this phenotype. Indeed, we found that the synthetic lethal interaction of *ATAD1* and *MARCH5* was suppressed in cells stably overexpressing MCL1 (*Figure 2E*). Again, we observed increased BIM$_{EL}$ and NOXA in *ATAD1Δ* cells, and deletion of *MARCH5* led to a further increase in NOXA levels (*Figure 2—figure supplement 2A*). These effects were limited in cells overexpressing MCL1, which may reflect regulation of NOXA degradation by MCL1 binding. As a positive control for testing gene essentiality, we transduced cells with an sgRNA targeting the pan-essential gene *PCNA*, which is necessary for DNA replication (*Girish and Sheltzer, 2020*). Deletion of *PCNA* was similarly toxic to WT and *ATAD1Δ* cells, and was not rescued by BIM deletion nor by MCL1 overexpression (*Figure 2—figure supplement 2B and C*). These data indicate that ATAD1 protects cells from *MARCH5* deletion specifically, rather than making them generally more resistant to perturbation of important genes.

Altogether, our CRISPR screens identified a synthetic lethal interaction between *ATAD1* and *MARCH5*, both of which enact protein extraction from the OMM. Loss of *ATAD1* and *MARCH5* antagonizes the anti-apoptotic function of MCL1 via increased abundance and activity of BIM and NOXA, which partially explains the synthetic lethality.

We next asked if BIM was sufficient to trigger apoptosis preferentially in *ATAD1Δ* cells. We generated Jurkat cell lines expressing a tetracycline-inducible GFP-BIM$_{EL}$ fusion protein. The dose-dependent increase in GFP-BIM expression was equivalent in WT and *ATAD1Δ* cells (anti-GFP blot, *Figure 2F*). Maximal expression of GFP-BIM$_{EL}$ using 500 ng/mL doxycycline killed cells regardless of

the presence or absence of ATAD1 (*Figure 2G*; *Figure 2—figure supplement 3A*). However, *ATAD1Δ* cells were hypersensitive to intermediate expression of ectopic $BIM_{EL}$, as measured by cell viability assays (*Figure 2G*), cleaved PARP immunoblots (*Figure 2F*), and live cell imaging using the Incucyte platform (*Figure 2—figure supplement 3A*). Thus, endogenous ATAD1 protects against BIM but ATAD1 can be overwhelmed with sufficiently high levels of BIM. These results demonstrate that BIM is sufficient to induce apoptosis preferentially in cells lacking *ATAD1*. We further studied how ATAD1 affected apoptotic priming (the propensity of a cell to undergo intrinsic apoptosis) using BH3 profiling. H4 glioma cells (Del(10q23); *ATAD1*-null; *Figure 2—figure supplement 3B*) were highly sensitive to BIM peptide, which is rescued by re-expression of $ATAD1^{WT}$ but not the catalytically dead $ATAD1^{E193Q}$ mutant (*Figure 2—figure supplement 3C*). ATAD1 therefore appears to suppress overall apoptotic priming, at least in this context, as measured by sensitivity to BIM BH3 peptide.

## ATAD1 directly and specifically extracts BIM from membranes

We hypothesized that BIM might be a direct substrate of the ATAD1 dislocase, which could explain how ATAD1 suppresses BIM-induced apoptosis. Consistent with BIM being an ATAD1 substrate, GFP-$BIM_{EL}$ co-immunoprecipitated with FLAG-tagged ATAD1 in H4 cells (*Figure 2—figure supplement 3D*). Inversely, ATAD1-FLAG co-immunoprecipitated with endogenous BIM (*Figure 2—figure supplement 3E*). Thus, reciprocal co-immunoprecipitation argues that ATAD1 and BIM physically interact in cells.

We further tested whether ATAD1 can directly extract BIM from a membrane using an in vitro system with purified components (*Wohlever et al., 2017*). We used $BIM_L$ because it is more soluble than $BIM_{EL}$ but shares the key structural features that would likely mediate ATAD1 recognition, including the tail-anchor and juxtamembrane regions (*Ley et al., 2005*; *Liu et al., 2019*; *Chi et al., 2020*). We were unable to purify active, full-length ATAD1. Instead, we swapped the N-terminal transmembrane domain with a $His_6$ tag, which anchored $His_6$-ATAD1 to liposomes doped with phospholipids containing nickel-chelated headgroups ('Ni Lipos', *Figure 2H*). In this extraction assay, TA proteins that are extracted from liposomes by ATAD1 bind soluble GST-tagged chaperones (SGTA and calmodulin), which are purified on a glutathione column and detected by immunoblotting (*Figure 2H*; *Wohlever et al., 2017*). We validated the Ni-His anchoring strategy using full-length or truncated yeast Msp1 (the yeast homolog of ATAD1) and positive and negative control substrates (*Figure 2—figure supplement 4A–C*).

$His_6$-ATAD1 directly and efficiently extracted 3xFLAG-$BIM_L$ from liposomes in this assay (*Figure 2I*; compare lanes with elution 'E' to input 'I'; lanes 13–16). As expected, this activity was ATP-dependent (*Figure 2I*, lanes 17–20) and was abolished when we used the catalytically inactive mutant, $ATAD1^{E193Q}$ (*Figure 2I*, lanes 1–4). Omission of Ni-chelating lipids from the liposomes ('Mito Lipos', which cannot anchor $His_6$-ATAD1; *Figure 2I*, lanes 5–8) prevented ATAD1 from extracting BIM, demonstrating that ATAD1 requires membrane anchoring for its dislocase activity. Importantly, ATAD1 did not extract yeast Fis1, consistent with previous reports that Fis1 is not an Msp1 substrate (*Li et al., 2019*; *Figure 2I*, lanes 9–12).

Many BH3-only proteins share key structural features, including a tail-anchor, so we next asked whether ATAD1 could extract other members of this protein family (*Wilfling et al., 2012*). In addition to BIM, we tested BIK, PUMA, and NOXA, since these have been proposed to mediate apoptosis triggered by proteotoxic stress. NOXA did not incorporate into proteoliposomes, which precluded it from our assay, and is consistent with a report that it lacks a transmembrane domain (*Andreu-Fernández et al., 2016*). ATAD1 extracted BIM in an ATP-dependent manner, as expected, but failed to extract BIK or PUMA under the same conditions (*Figure 2J and K*). While it is not clear how ATAD1 distinguishes between these substrates, BIM differs from BIK and PUMA in that it has a positively charged C-terminus and an intrinsically disordered region N-terminal to the transmembrane domain, which are important features for substrate recognition by Msp1, the yeast homolog of ATAD1 (*Figure 2—figure supplement 5*; *Castanzo et al., 2020*; *Li et al., 2019*). Taken together, we demonstrate that ATAD1 and BIM physically and genetically interact in cells, and that ATAD1 can directly extract BIM – but not other, related, BH3-only proteins – from membranes in a reconstituted system.

These data raise the question of what happens to BIM after it has been extracted by ATAD1. We transduced SW1088 cells, which are a Del(10q23) cell line suitable for imaging, with either EV or ATAD1-FLAG. These cells were then transduced with TetON(GFP-$BIM_{EL}$ΔBH3), in which four point

mutations in the BH3 domain of BIM neutralize its pro-apoptotic activity to permit live cell imaging. We assessed GFP-BIM$_{EL}$ΔBH3 localization using live cell confocal microscopy in the presence and absence of ATAD1, using MitoTracker Red to label mitochondria. ATAD1 altered the localization of BIM under basal conditions, generating GFP-positive puncta that did not colocalize with mitochondria (*Figure 2—figure supplement 6A*). Since BIM is regulated by proteasomal degradation, we additionally treated cells with bortezomib, a proteasome inhibitor. Treatment with bortezomib exacerbated this phenotype and resulted in larger, brighter GFP-positive puncta only in ATAD1 expressing cells (*Figure 2—figure supplement 6B and C*). We saw the same phenomenon when we genetically labeled mitochondria with mCherry, ruling out the possibility that these GFP-BIM puncta are merely depolarized mitochondria that cannot accumulate Mitotracker dye (*Figure 2—figure supplement 7*). GFP-positive puncta also did not colocalize with lysosomes or peroxisomes (*Figure 2—figure supplements 8 and 9*). In addition to localization, BIM is regulated by inhibitory phosphorylation, which we had previously observed to be affected by *ATAD1* status. Bortezomib treatment led to the accumulation of phosphorylated BIM$_{EL}$ in SW1088 cells expressing ATAD1, while BIM$_{EL}$ accumulated in an unphosphorylated state in SW1088 cells transduced with EV (*Figure 3—figure supplement 1A*). We next used the PC3 prostate cancer cell line, which is PTEN-null and has a partial deletion of *ATAD1* (*Figure 3—figure supplement 1B*). In parallel to our findings with SW1088 cells, BIM accumulated in a phosphorylated state after treatment with bortezomib in PC3 cells with *ATAD1* present (sgNT), but this phosphorylation was abrogated in PC3 cells with *ATAD1* deleted (*Figure 3—figure supplement 1C and D*). Thus, in the context of proteasome inhibition, ATAD1 shifts the localization of BIM from mitochondria to cytoplasmic puncta and promotes inhibitory phosphorylation of BIM.

## Proteasome inhibition is preferentially toxic to ATAD1-deficient cells

We next sought to pharmacologically exploit *ATAD1* deficiency in relevant cancer models, drawing on our discovery of synthetic lethality with *MARCH5*. Since MARCH5 is a ubiquitin E3 ligase, we hypothesized that disrupting the ubiquitin proteasome system downstream of MARCH5 might also be preferentially toxic to cells lacking *ATAD1*. Consistent with this idea, proteasome inhibitors are known to increase BIM and NOXA abundance, thereby triggering apoptosis (*Baou et al., 2010*; *Meller et al., 2006*). There were hints from our CRISPR screens that *ATAD1*-deficient cells might be more sensitive to ubiquitin proteasome system dysfunction generally, with *PSMC6* (encoding a proteasome subunit; Jurkat screen) and *MUL1* (encoding a mitochondrial E3 ligase; HGC27 screen) also scoring among the top hits for synthetic lethality (*Figure 1E and G*).

Re-expression of *ATAD1* in the Del(10q23) cell lines HGC27 (gastric cancer), SW1088 (glioma), or RPMI7951 (melanoma) suppressed toxicity caused by various structurally distinct proteasome inhibitors (*Figure 3A–C*, *Figure 3—figure supplement 2A–E*). That the phenotype was common to multiple proteasome inhibitors with different mechanisms of action increases our confidence that this is due to an on-target effect. Deletion of *ATAD1* increased sensitivity to bortezomib in PC3 cells (*Figure 3D*). Inversely, overexpressing ATAD1$^{WT}$, but not ATAD1$^{E193Q}$, made PC3 cells more resistant to bortezomib (*Figure 3—figure supplement 2F*).

We hypothesized that apoptosis was responsible for the sensitivity of *ATAD1*-deficient cells to proteasome inhibition. Indeed, bortezomib treatment induced robust PARP cleavage in the same *ATAD1*-deficient cells described above, but not in cognate *ATAD1*-positive cells (*Figure 3E–G*, *Figure 3—figure supplement 3A-C*). Bortezomib treatment caused polyubiquitinated proteins to accumulate to the same extent in the presence or absence of *ATAD1*, indicating that ATAD1 affects how the cell responds to proteotoxic stress, rather than blocking the proteotoxic insult itself (*Figure 3E–G*, *Figure 3—figure supplement 3B*). Thus, *ATAD1* appears to be essential for viability in cells subjected to ubiquitin proteasome system dysfunction.

Apoptotic cell death is only one of many mechanisms underlying proteasome inhibitor toxicity (*Tsvetkov et al., 2019*; *Schneider and Bertolotti, 2015*; *Huang et al., 2020*). We next asked whether ATAD1 affects proteasome inhibitor sensitivity via some apoptosis-independent pathway, in which case ATAD1 re-expression and caspase inhibition (which blocks apoptosis) would have an additive effect in mitigating proteasome inhibitor toxicity. Treatment of *ATAD1* knockout PC3 cells with the caspase inhibitor zVAD-FMK rescued the effects of bortezomib essentially back to WT levels (*Figure 3H*; *Figure 3—figure supplement 3E*). Additionally, *ATAD1* re-expression phenocopied treatment with zVAD-fmk in RPMI7951 cells (Del(10q23)) treated with bortezomib (*Figure 3—figure supplement 3F*

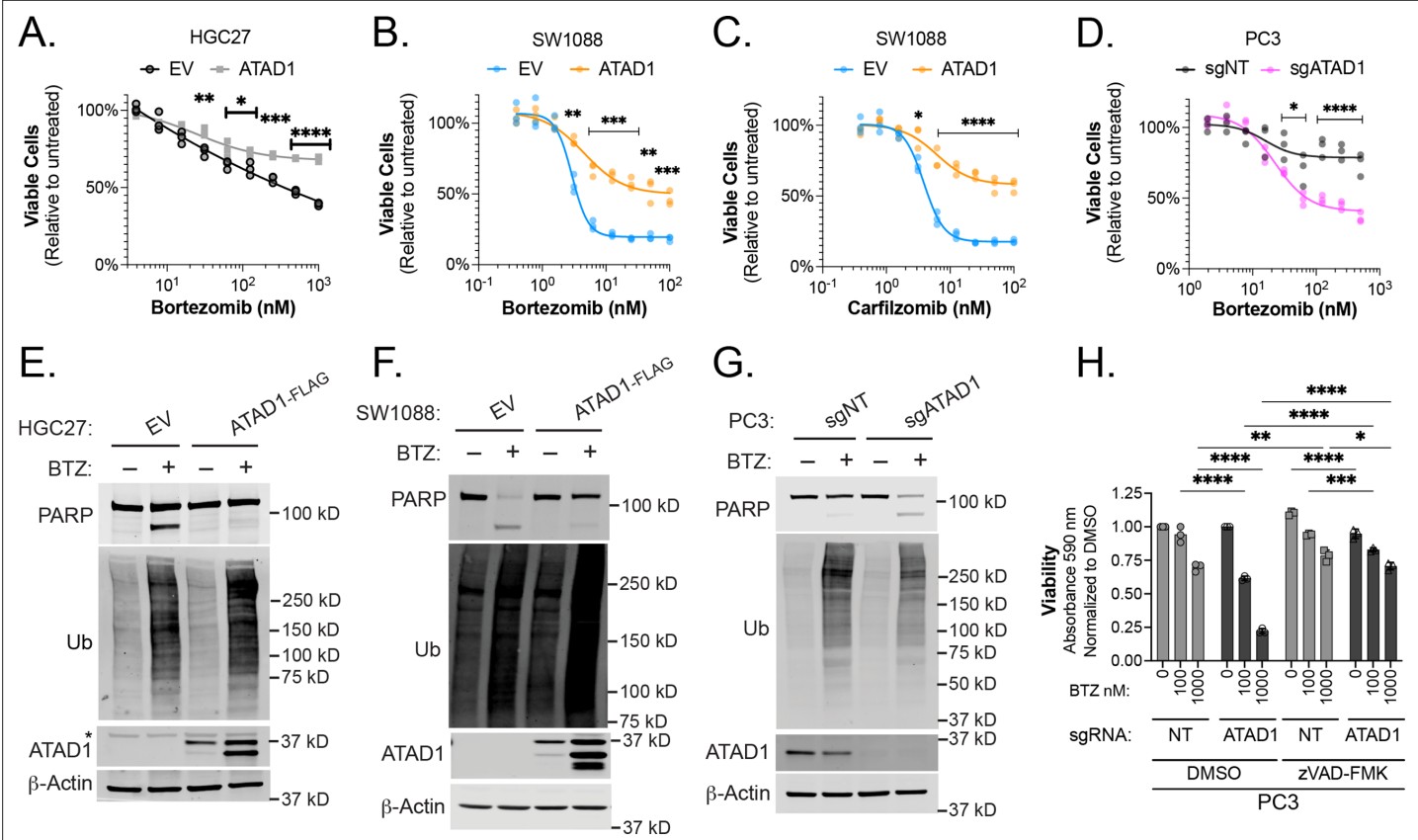

**Figure 3.** ATAD1 protects cells from apoptosis triggered by proteasome inhibition. (**A**) Viability of HGC27 cells treated with bortezomib (BTZ) for 16 hr. (**B**) Viability of SW1088 cells treated with BTZ for 24 hr. (**C**) Viability of SW1088 cells treated with carfilzomib, a different proteasome inhibitor, for 24 hr. (**D**) Viability of PC3 cells treated with BTZ for 16 hr. (**E**) Western blots of HGC27 cells screen treated with 1 μM BTZ for 16 hr. (**F**) Western blots of SW1088 cells transduced with empty vector (EV) or ATAD1-FLAG and treated with 100 nM BTZ for 16 hr. (**G**) Western blots of PC3 cells transduced with non-targeting sgRNA or *ATAD1* sgRNA and treated with 1 μM BTZ for 16 hr. (**H**) Viability as measured by normalized crystal violet staining (Abs 590 nm) in PC3 cells transduced with sgNT vs. sgATAD1, treated with BTZ for 16 hr in the presence or absence of 40 μM zVAD-FMK. Data analyzed by two-way ANOVA with Tukey's multiple comparisons.

The online version of this article includes the following source data and figure supplement(s) for figure 3:

**Source data 1.** Source data used to make *Figure 3*.

**Figure supplement 1.** ATAD1 promotes BIM_EL phosphorylation in response to proteasome inhibition.

**Figure supplement 1—source data 1.** Source data used to make *Figure 3—figure supplement 1*.

**Figure supplement 2.** *ATAD1* status and proteasome inhibition in cancer cell lines.

**Figure supplement 3.** ATAD1 protects cells from proteasome inhibition by blocking apoptosis, specifically.

**Figure supplement 3—source data 1.** Source data used to make *Figure 3—figure supplement 3*.

**Figure supplement 4.** Effect of BIM knockout in PC3 cells treated with bortezomib (BTZ).

**Figure supplement 4—source data 1.** Source data used to make *Figure 3—figure supplement 4*.

*and G*). Moreover, there was no additive effect of zVAD-fmk and ATAD1, suggesting that ATAD1 and zVAD-fmk act in a linear pathway to prevent the death of cells subjected to proteasome inhibition (*Figure 3H*, *Figure 3—figure supplement 3G and H*).

Although caspase inhibition completely rescued the bortezomib phenotype of ATAD1-deficient cells, the same was not true for deletion of BIM in PC3 cells, implying that other factors in addition to BIM mediate this phenotype (*Figure 3—figure supplement 4A and B*). We examined several OMM-localized or TA proteins in PC3 cells with and without bortezomib, and reduced BIM phosphorylation was the only consistent change caused by *ATAD1* deletion (*Figure 3—figure supplement 4C*). Other BH3-only proteins reported to be activated by proteotoxic stress include BIK, PUMA, but our in vitro

extraction assay ruled out a direct action of ATAD1 to extract these proteins. Increased mitochondrial fragmentation in ATAD1-null cells could be one potential explanation, since mitochondrial dynamics are intimately connected to apoptosis (*Chen et al., 2014*). Nonetheless, these results indicate that the protective effects of ATAD1 during proteasome inhibition can be explained exclusively by limiting apoptosis, with BIM extraction playing a key role.

## ATAD1 loss limits tumor progression, particularly under proteotoxic stress

We next tested whether deficiency of *ATAD1* sensitized cancer cells to proteasome inhibition in mouse tumor xenografts. Bortezomib had no effect on growth of PC3 xenografts transduced with sgNT, but significantly decreased the growth rate of tumors in which *ATAD1* was deleted (*Figure 4A and B*). Bortezomib treatment induced a significant increase in NOXA levels in sgATAD1 tumors, but not in sgNT tumors (*Figure 4C*; *Figure 4—figure supplement 1*). NRF1/TCF11 is a transcription factor that is activated upon proteasome dysfunction, and we measured NRF1 levels by immunoblot to assess bortezomib activity (*Radhakrishnan et al., 2010*). NRF1 was equally induced by bortezomib in sgNT and sgATAD1 tumors (*Figure 4C*; *Figure 4—figure supplement 1*), again supporting our conclusion that ATAD1 protects cells from death triggered by protein stress rather than preventing the protein stress itself. Altogether, these in vivo results demonstrate that bortezomib is preferentially toxic to PC3 tumors that lack *ATAD1*.

ATAD1 was not necessary for basal tumor growth in PC3 cells, so we assessed the effect of *ATAD1* on tumor growth of SW1088 cells, a Del(10q23) glioma cell line that is non-tumorigenic in SCID mice (*Jiang et al., 2017*; *Mercapide et al., 2003*). As expected, SW1088 cells transduced with EV failed to form tumors in 17 out of 17 NOD/SCID mice. However, SW1088 cells transduced with ATAD1 grew palpable tumors in 20 out of 21 mice (*Figure 4—figure supplement 2A and B*). ATAD1 thus promotes growth of SW1088 glioma xenografts, despite not affecting their proliferation in 2D culture (*Figure 4—figure supplement 2C*).

Finally, we examined clinical outcomes in patients with tumors that were (i) *PTEN*-null, (ii) both *PTEN*-null and *ATAD1*-null, or (iii) neither ('unaltered' at these loci). We queried two studies of patients with metastatic, castration-resistant prostate cancer (mCRPC) (*Robinson et al., 2015*; *Abida et al., 2019*). The median overall survival of mCRPC patients with *PTEN*-null *ATAD1*-null tumors (77 months) was more than double that of patients with only *PTEN*-null or 'unaltered' tumors (37 months, *Figure 4D*). In a subset of patients, we were able to additionally assess overall survival after initial hormone therapy or chemotherapy. Patients whose tumors lacked both *ATAD1* and *PTEN* had longer survival post-therapy than did the patients whose tumors lacked only *PTEN* (*Figure 4E and F*). Altogether, these data suggest that ATAD1 exerts a pro-survival effect in tumor cells (with *ATAD1* deficiency decreasing tumor fitness and improving patient survival) in both murine xenografts and human mCRPC patients.

## Discussion

In this work we describe how the collateral deletion of *ATAD1*, a mitochondrial protein extractase, sensitizes cancer cells to apoptosis induced by proteasome dysfunction. It might appear counter-intuitive that ATAD1 protects against apoptosis despite Chr10q23 deletion occurring so frequently in cancer. This may be partly explained by the strong selection for deletion of *PTEN*, which virtually always co-occurs with deletion of *ATAD1*. PTEN loss activates AKT, which has pro-survival effects, including decreasing transcription of *BCL2L11* (BIM) by inhibiting FOXO3A, and inactivating the BH3-only protein BAD by direct phosphorylation (*Gilley et al., 2003*; *Datta et al., 1997*). Therefore, co-deletion of *PTEN* likely buffers some of the apoptotic priming induced by *ATAD1* loss. Even so, neutral or detrimental alleles that act as genetic 'hitchhikers' when they are physically linked to advantageous alleles are a well-described phenomenon in evolutionary biology, so the same could be expected in the evolutionary arena of cancer (*Lang et al., 2013*). Furthermore, genetic lesions that prime cells for apoptosis are not always selected against in cancer. On the contrary, certain oncogenes (including MYC paralogs) are strongly selected for, despite their known effect of priming for apoptosis (*Mason et al., 2008*; *Dammert et al., 2019*). While we cannot rule out the possibility that ATAD1 loss could be beneficial to tumor cells under some circumstances, our data clearly demonstrate its role as a pro-survival factor in cancer cells of diverse origins. These data also illustrate that ATAD1 is particularly

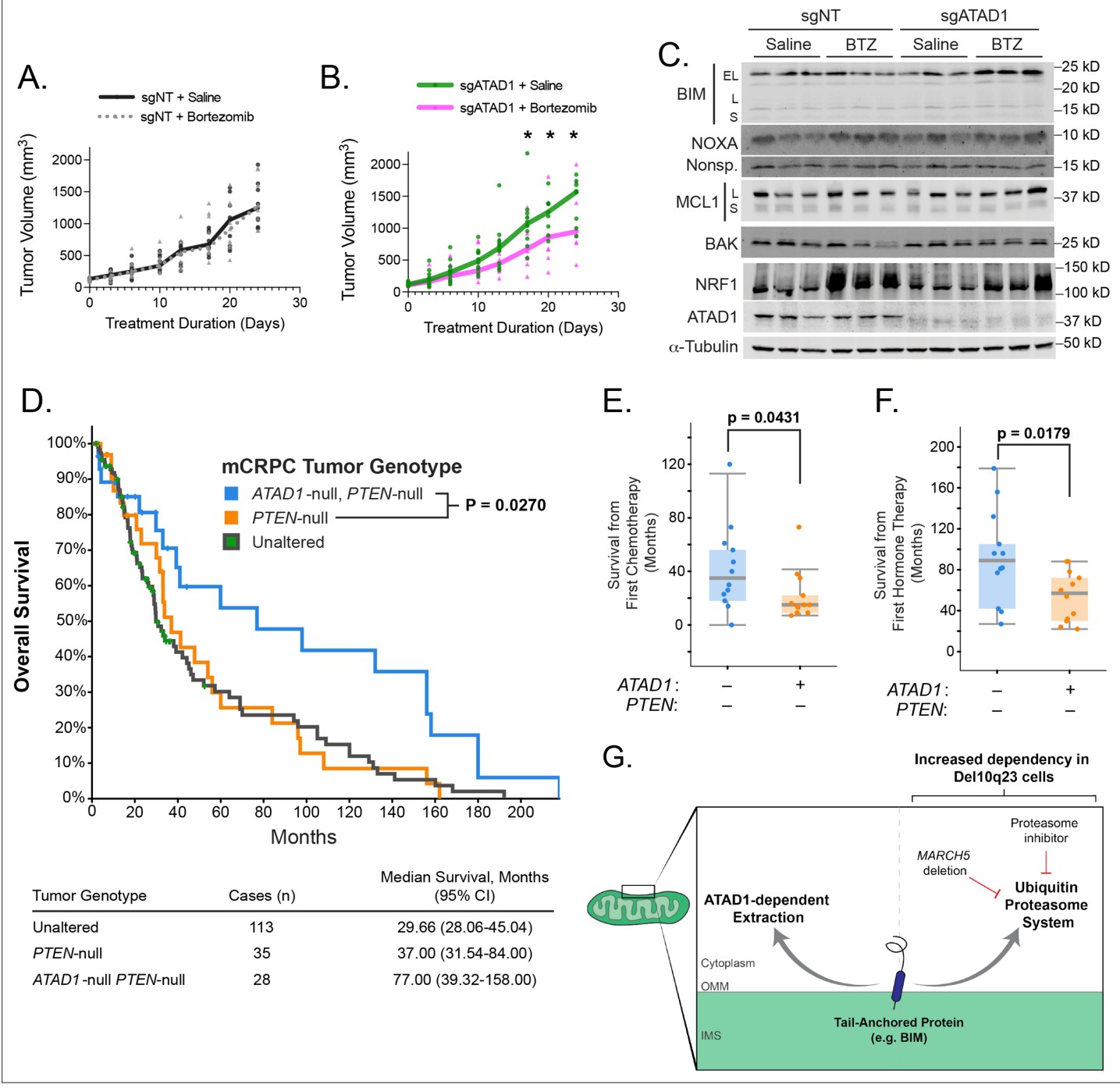

**Figure 4.** *ATAD1* loss sensitizes PC3 xenografts to proteasome inhibition and predicts improved survival in patients with metastatic prostate cancer. (**A**) Tumor volume over time for mice with flank xenografts of PC3 cells treated with saline (vehicle) or 1 mg/kg bortezomib (BTZ). (**B**) Tumor volume over time for mice with flank xenografts of *ATAD1*-knockout PC3 cells treated with saline (vehicle) or 1 mg/kg BTZ. (**C**) Western blots of whole cell lysates from tumor samples taken from animals as in (**A,B**) sacrificed 24 hr after receiving saline/BTZ. (**D**) Kaplan-Meier curve of overall survival from patients with metastatic, castrate-resistant prostate cancer (mCRPC), stratified based on tumor genotype at the *ATAD1* and *PTEN* loci, with accompanying table below. (**E**) Survival (months) after initiating chemotherapy or hormone therapy (**F**) in patients with mCRPC, based on tumor genotype. (**G**) Graphical summary.

The online version of this article includes the following source data and figure supplement(s) for figure 4:

**Source data 1.** Source data used to make *Figure 4*.

**Figure supplement 1.** Quantification of western blots from PC3 xenograft lysates.

**Figure supplement 2.** *ATAD1* re-expression confers tumorigenicity to SW1088 cells.

important for cellular health in the context of ubiquitin proteasome system dysfunction, which can be at least partly explained by ATAD1 antagonizing BIM. Patients with metastatic prostate cancer tumors that lack ATAD1 live significantly longer, and appear to respond better to therapy, which suggests that ATAD1 is also important in the context of human tumors subjected to different stressors.

The effect of ATAD1 on apoptosis likely extends beyond *PTEN*-null cancers into a broader physiological context. ATAD1 has not been explicitly linked to apoptosis, but it was originally discovered in mammals via a genetic screen for factors that prevent neuronal cell death (*Dai et al., 2010*). Further, *Atad1* phenocopies *Bcl2* and *Bcl2l1* in the middle cerebral artery occlusion mouse model of ischemic stroke: deletion of *Atad1* or *Bcl2* exacerbates, and overexpression of *Atad1*, *Bcl2*, or *Bcl2l1* decreases ischemic cell death (*Zhang et al., 2019*; *Martinou et al., 1994*; *Parsadanian et al., 1998*; *Broughton et al., 2009*). We speculate that antagonism of BIM may contribute to the neuroprotective function of ATAD1 in vivo (*Zhang et al., 2011*).

One limitation of our study is that deletion of BIM does not completely rescue the effects of *ATAD1*. Even though we demonstrate that BIM is an ATAD1 substrate and that this interaction is important for the pro-survival effects of ATAD1 (such as the synthetic lethality of *ATAD1* and *MARCH5*), there must be other ATAD1 substrates that contribute to the proteasome inhibition phenotype of *ATAD1*-null cells. We speculate that a multitude of ATAD1 substrates accumulate on the OMM during proteasome inhibition, and deletion of a single one (e.g. BIM) is insufficient to counteract the summative stress. Further clarity on the substrate repertoire of ATAD1 will be an important direction for future research. Future work should also clarify whether a tumor's tissue of origin influences the dependencies generated by ATAD1 deficiency, although our use of multiple cell lines from different tumor types suggests a phenotype with reasonably high penetrance.

Overall, these data show that cells lacking ATAD1 have an increased dependency on the ubiquitin proteasome system. We therefore propose a model in which mitochondrial TA proteins have (at least) two fates: extraction by ATAD1, and ubiquitination and proteasomal degradation (*Figure 4G*). This model is strongly supported by the finding that ATAD1 becomes important for cellular viability only in the context of a dysfunctional UPS, such as upon *MARCH5* deletion or pharmacological inhibition of the proteasome. That caspase inhibition completely rescues the absence of ATAD1 in proteasome inhibitor treated cells argues that ATAD1 is important in this context specifically for preventing apoptosis, as opposed to a general loss of cell fitness. Proteins extracted by ATAD1 may or may not be subsequently degraded by the proteasome, but ATAD1 is beneficial to the cell even when – and especially when – the proteasome is inhibited.

That ATAD1 loss sensitizes to proteasome dysfunction could have therapeutic implications for hundreds of thousands of cancer patients with Del(10q23) tumors, especially considering that drugs targeting the proteasome are already approved for the treatment of cancer. Previous trials of proteasome inhibitors in unselected patients with prostate cancer have shown activity in a subset of patients, consistent with the possibility that this subset was enriched for patients with *ATAD1*-null tumors (*Papandreou et al., 2004*; *Naik et al., 2018*; *Morris et al., 2007*). A retrospective analysis of *ATAD1* status in tumors of 'responders' and 'non-responders' from these trials would be a valuable first step toward the clinical translation of our findings. We predict that pharmacological inhibitors of MARCH5 could achieve an even better therapeutic window than proteasome inhibitors in treating tumors with *ATAD1* deletion, and would be an exciting area for drug development. In sum, our work uncovers a new connection between mitochondrial protein homeostasis and cell death, and has the potential for clinical impact for patients with *PTEN*/*ATAD1* co-deleted tumors.

## Materials and methods

### Lead contact

Further information and requests for resources and reagents should be directed to and will be fulfilled by the lead contact, Jared Rutter (rutter@biochem.utah.edu).

### Materials availability

All unique/stable reagents generated within this study are available from the lead contact upon request without restriction.

## Data and code availability

The CRISPR screening datasets generated during this study are available in the supplemental materials. The human mCRPC survival data are available in supplemental materials. The uncropped source data files for every western blot image are included in the 'source data' files.

## Experimental model and subject details

### *ATAD1* knockout cell lines

Jurkat E6.1 human T-ALL cells (ATCC TIB-152) were grown in RPMI1640 with 10% FBS and 100 U/mL Pen/Strep (Thermo Fisher). Cells were electroporated using Lonza SE Cell Line 4D-Nucleofector X Kit L according to the manufacturer's specifications and protocol optimized for Jurkat E6.1 cells. Px458-derived plasmids encoding sgRNA targeting *ATAD1* were transiently expressed in Jurkat cells via electroporation. One of two sgRNAs were used, targeting either exon 2 (sense oligo: 5'-CCGACTCA AAGGACGAGAAA-3') or exon 5 (sense oligo: 5'-CGGTCAGTGTCGAAGGCTGA-3'). Three days later, GFP-positive cells were sorted (BD FACSAria) and plated as single cells in 96-well plates. Clonal cell lines were grown, harvested, and evaluated for *ATAD1* deletion via immunoblot using a knockout-validated monoclonal antibody (NeuroMab).

PC3 cells were transduced with lentivirus encoding LentiCRISPRv2-GFP (LCv2G) with non-targeting sgRNA (sgNT, which contains a 15 nt sgRNA sequence, 5'-GAGACGGACGTCTCT-3', that does not precede any 5'-NGG-3' sites in the human genome. This was determined by using GAGACGGACGTC TCTNGG as input for ncbi BLAST, and filtering results to 100% query coverage, 100% identity) or LCv2G with sgATAD1 (targeting exon 5: 5'-CGGTCAGTGTCGAAGGCTGA-3'). Three days after transduction, GFP-positive cells were sorted (BD FACSAria) and maintained as a polyclonal population. Editing was confirmed by immunoblot as above.

### *ATAD1* re-expression in Del(10q23) cell lines

H4 and PC3 cells were transduced with retrovirus (pQCXIP transfer plasmid) encoding ATAD1 with C-terminal FLAG and HA tags. Two days after transduction, cells were selected with 1 µg/mL puromycin for 4 days. RPMI7951, HGC27, and SW1088 cells were transduced with lentivirus (pLenti-Blast transfer plasmid) encoding ATAD1-FLAG, and selected with 8 µg/mL blasticidin for 6 days. Cells were grown in media containing the selective antibiotic upon thawing stocks, but no experiments were conducted using media that contained selective antibiotics.

## Method details

### Immunohistochemistry

#### ATAD1 (using NeuroMab #75-157 mouse monoclonal antibody)

The ATAD1 immunohistochemical staining was performed on 4 µm thick sections of formalin-fixed, paraffin-embedded tissues. Sections were air-dried and then melted in a 60°C oven for 30 min. Slides were loaded onto the Leica Bond III automated staining instrument (Leica Biosystems, Buffalo Grove, IL) and de-paraffinized with the Bond Dewax solution. The antigen retrieval performed was done with Bond Epitope Retrieval Buffer 2 (ER2, pH 8.0) for 20 min at 95°C. The ATAD1 primary antibody concentration of 1:400 was applied at an incubation time of 30 min at room temperature. Positive signal was visualized using the Bond Polymer Refine Detection kit-DAB, which is a goat anti-mouse/anti-rabbit secondary HRP/polymer detection system, utilizing DAB (3,3' diaminobenzidine) as the chromogen. Tissue sections were counterstained with hematoxylin for 10 min. The slides were removed from the immunostainer and placed in a $dH_2O$/DAWN mixture. The sections were gently washed in a mixture of deionized water and DAWN solution to remove any unbound reagent. The slides were gently rinsed in deionized water until all of wash mixture was removed. The slides were de-hydrated in graded ethanols, cleared in xylene and then coverslipped.

#### PTEN (using rabbit anti-human monoclonal antibody): clone 138G6, catalog #9559L, Cell Signaling, Danvers, MA

The PTEN immunohistochemical staining was performed on 4 µm thick sections of formalin-fixed, paraffin-embedded tissues. Sections were air-dried and then melted in a 60°C oven for 30 min. Slides were loaded onto the Ventana BenchMark Ultra automated staining instrument (Ventana Medical

Systems, Tucson, AZ), de-paraffinized with the EZ Prep solution. The antigen retrieval performed was done with a citrate buffer (pH 6.0) in a pressure cooker (BioCare Medical, Concord, CA) for 4 min at 100°C then cooled in hot buffer for 30 min. The PTEN primary antibody concentration of 1:50 was applied at an incubation time of 2 hr at room temperature. The Ventana Amplification kit was applied to increase the antibody signal. Positive signal was visualized using the UltraView DAB detection kit, which is a goat anti-mouse/anti-rabbit secondary HRP/polymer detection system, utilizing DAB as the chromogen. Tissue sections were counterstained with hematoxylin for 16 min. The slides were removed from the immunostainer and placed in a dH$_2$O/DAWN mixture. The sections were gently washed in a mixture of deionized water and DAWN solution to remove any unbound reagent and coverslip oil applied by the automated instrument. The slides were gently rinsed in deionized water until all of wash mixture was removed. The slides were de-hydrated in graded ethanols, cleared in xylene, and then coverslipped.

## Cell culture

Jurkat cells were cultured in RPMI1640 with 10% FBS (Sigma) and 100 U/mL Pen/Strep. Cells were counted regularly and typically split at a concentration of approximately 1–1.5 × 10$^6$ cells/mL, but always before reaching a concentration of 3×10$^6$ cells/mL. Jurkat cells transduced with tet-inducible vectors were cultured in RPMI1640 with 10% 'Tet System Approved FBS' (Takara) instead of standard FBS. Adherent cell lines were maintained in subconfluent cultures in the following media: RPMI1640 (PC3), DMEM (H4 and SW1088), EMEM (RPMI7951 and HGC27), all with 10% FBS and 100 U/mL Pen/Strep. Cells were periodically tested for mycoplasma contamination using a MycoAlert kit and were negative.

## Cloning

All cloning was conducted via traditional PCR/restriction enzyme 'cut and paste' methods and verified by Sanger sequencing.

### ATAD1 constructs

Retroviral plasmids encoding ATAD1-FLAG/HA and ATAD1$^{E193Q}$-FLAG/HA were published previously (*Chen et al., 2014*). Lentiviral vectors were made, using the pLenti-BLAST backbone, by PCR-amplifying the ATAD1 CDS from the above retroviral vectors, but truncating the construct by replacing the HA tag with a stop codon, and ligating between SalI and XbaI sites.

### GFP-BIM constructs

The pLVXTet-One vector was purchased from Takara. The coding sequence for EGFP was PCR-amplified and ligated into the MCS using AgeI/BamHI sites. A fusion of EGFP-BIM$_{EL}$ was generated using SOEing PCR and ligated using AgeI/BamHI sites. EGFP-BIM$_{EL}$ was also ligated into pEGFP-C3 for transient transfection.

A gene fragment encoding BIM$_{EL}$ΔBH3 was synthesized by GeneWiz and subcloned into the pLVX-Tet-One vector described above to make an N-terminal GFP fusion.

## CellTiterGlo viability assay

Viability was determined by CellTiterGlo (Promega) according to the manufacturer's recommendation, with some modifications. Cells were plated at a density of 5×10$^3$ cells/well (adherent cell lines) or 2–4 × 10$^4$ cells/well (Jurkat) in 100 µL in 96-well plates with white walls and clear bottoms (Corning #3610). CellTiterGlo reagent was reconstituted, diluted 1:4 using sterile PBS, and stored at –20°C in 10 mL aliquots. The outer wells of the 96-well plates were filled with media but not with cells, due to concerns of edge effects. Luminescence was measured using a Biotek Synergy Neo2 microplate reader. Luminescence values were normalized on each plate to untreated cells on the same plate, and expressed as percent. Viability experiments were conducted with multiple biological replicates and repeated with independent experiments.

## Incucyte

Jurkat cells stably transduced with TetON(GFP-BIM$_{EL}$) were seeded at a density of 10$^3$ cells/well in clear-bottom, black 96-well plates (Corning) with different concentrations of doxycycline. Cells

were imaged with an Incucyte SX5 system and monitored by phase contrast microscopy, with five images taken per well, every 4 hr, for approximately 4 days. Confluence was normalized to t=0 and is expressed as fold-change. Three replicate wells were used for each condition, and three independent experiments were conducted.

## Spinfection

Jurkat cells were routinely spinfected in 12-well plates, with $2–5 \times 10^6$ cells per mL in 1 mL containing 10 µg/mL polybrene. Centrifugation was conducted at 30°C and $1100 \times g$ for 60 min.

## CRISPR-based genetic interaction experiments

Jurkat cells (*ATAD1Δ* or WT) were transduced with LentiCRISPSRv2-GFP encoding a non-targeting sgRNA (see above) or sgRNA targeting *BCL2L11* (encoding BIM; gRNA: 5'-GTTGTGGCTCTGTCTG TAGGG-3') and sorted by FACS. For MCL1 overexpression experiments, Jurkat cells transduced with LentiCRISPRv2GFP-sgNT were subsequently transduced with viral particles packaged with pLenti-GFP or pLenti-Myc-MCL1, which encodes human MCL1 with an N-terminal Myc tag, and selected with 1 µg/mL puromycin.

Viral particles were made using the LRCherry2.1 transfer plasmid, which encodes U6-sgRNA and mCherry. Cells were spinfected using 10 µg/mL polybrene and viral supernatant. The same number of cells was used for each transduction. One day after spinfection (day 1), cells were resuspended in fresh RPMI media and cultured until day 3, when they were split or seeded into 96-well plates for subsequent analysis by CellTiterGlo (Promega). Viability was calculated by dividing CellTiterGlo values (day 4 or day 7) for a given cell line transduced with sgMARCH5 (5'-GCACTGAGGACATGCCACTC -3') or sgPCNA (5'-CTACCGCTGCGACCGCAACC-3') by the values for the same cell line on the same plate transduced with sgAAVS1 (5'-ACTGTTGACGGCGGCGATGT-3'; *Girish and Sheltzer, 2020*). Cell pellets were collected on day 4 for western blot.

## Crystal violet staining

Cells cultured in 12- or 6-well plates were washed twice with PBS then fixed with 4% paraformalde-hyde (Sigma-Aldrich) for 30 min at room temperature. Wells were washed with ddH$_2$O three times, then stained with 0.1% (w/v) crystal violet solution in 20% methanol for 30 min at room temperature. Wells were again washed with ddH$_2$O three times, inverted to dry, and plates were photographed against a white background using an iPhone X. For quantification, glacial acetic acid was added to each well to elute the dye, and plates were incubated at room temperature on a rotary shaker for 30 min. Absorbance was measured at 590 nm using a Biotek Synergy Neo2 microplate reader, and values were normalized to those from untreated cells of the same genotype on each plate.

## SDS-PAGE and immunoblotting

Whole cell lysates were prepared by scraping cells directly into RIPA buffer (or adding RIPA buffer to Jurkat cell pellets) supplemented with protease and phosphatase inhibitors (Sigma-Aldrich P8340, Roche Molecular 04906845001), incubated on ice for 30 min with vortexing every 10 min, and then spun at $16,000 \times g$ for 10 min at 4°C to remove insoluble material. Supernatant was saved as lysate and concentrations were normalized for total protein content after measuring with a BCA (Thermo Scientific 23225). Samples were resolved by SDS-PAGE or Tris-glycine gels (Invitrogen XP04205BOX) and transferred to nitrocellulose or PVDF (extraction assay) membranes. Immunoblotting was performed using the indicated primary antibodies which are listed in the Key resources table according to the manufacturers' recommendations, and analyzed by Licor Odyssey or Azure C500 (extraction assay). Note that the detector for the Azure C500 has several columns of pixels which appear to be non-functional. This gives the appearance of thin vertical white lines in some images. This can be readily viewed in raw data files by over-adjusting the contrast.

## Co-immunoprecipitation

H4 cells (expressing EV or ATAD1-FLAG/HA) were transfected with GFP-BIM$_{EL}$ in pEGFP-C3 (GFP-BIM), 10 µg plasmid for 10 cm plate, in the presence of 20 µM zVAD-fmk. Transient expression proceeded overnight (approximately 16 hr). Cells were washed with cold PBS and lysed with HN buffer supplemented with protease inhibitor cocktail and 1% CHAPS (HNC buffer). Magnetic anti-FLAG

beads (Sigma-Aldrich) were equilibrated with HNC buffer and then mixed with lysate (after removing 10% volume as input). Bead-lysate mixtures were incubated on a rotator at 4°C for 2–4 hr. Beads were washed 3× with HNC buffer, then heated at 65°C in 30 µL 1× Laemmli buffer for 10 min.

## Cell counting

Cells were counted using Bio-Rad TC20 cell counter. At least two samples were taken from a culture any time a count was to be made and the mean was recorded. For proliferation experiments with Jurkat cells (cell counts over time), a hemocytometer was used.

## Jurkat CRISPR screen

Jurkat cells (WT parental, *ATAD1Δ* #1, and *ATAD1Δ* #2) were transduced by spinfection with a genome-wide lentiviral sgRNA library (Addgene #1000000100; *Wang et al., 2015*) that also encoded Cas9 and a puromycin resistance cassette. Transduction was optimized to achieve an approximate transduction efficiency of 30%, and cells were selected with puromycin (0.5 µg/mL) for 3 days, allowed to recover without puromycin for 2 days, then maintained in a lower dose of puromycin (0.2 µg/mL) for the duration of the screen. An initial sample of cells ($8 \times 10^7$) were collected and frozen at the endpoint of puromycin selection (6 days post-transduction). Cells were then maintained in culture for 14 cumulative population doublings (CPDs). Cells were passaged every 2 days and seeded into new flasks at a density of $2 \times 10^5$ cells/mL. After 14 population doublings, representative samples were collected ($8 \times 10^7$ cells). As described elsewhere (*Adelmann et al., 2019*), cell pellets were processed using a QIAamp DNA Blood Maxiprep, sgRNA sequences were amplified by PCR, and amplicons were sequenced for 40 cycles by Illumina HiSeq NGS at the Whitehead Institute Genome Technology Core Facility.

Sequencing reads were aligned to the sgRNA library, given a pseudocount of 1, the counts from each sample were normalized for sequencing depth, and the relative abundance of each sgRNA was calculated as described previously (*Wang et al., 2015*; *Kanarek et al., 2018*). sgRNAs with fewer than 50 reads, and genes with fewer than 4 sgRNAs, in the initial reference dataset were omitted from downstream analyses. The $\log_2$ fold-change in abundance of each sgRNA between the final and initial reference populations was calculated and used to define a CS for each gene. The CS is the average $\log_2$ fold-change in abundance of all sgRNAs targeting a given gene. To achieve a direct comparison of gene essentiality in an *ATAD1Δ* clone to that in the WT control, we omitted sgRNAs that were not adequately represented (i.e. <50 reads at the initial time point) in both groups. This step enables a paired analysis of sgRNA changes in abundance, and avoids including a given sgRNA that 'scored' in one genetic background but whose effects cannot be assessed in another. Data were analyzed and plotted using ggplot2 with R version 4 and RStudio version 1.1.442.

## HGC27 CRISPR screen

The HGC27 CRISPR screen was conducted using the Brunello sgRNA library, which was obtained from Addgene and amplified according to the instructions provided by the depositing lab. HGC27 cells transduced with pLentiBLAST-EV or ATAD1-FLAG were selected and propagated as described above. Cells were spinfected with viral sgRNA library, and 1 day later were treated with puromycin (0.5 µg/mL) for 2 days. The following day (day 4), cell pellets were collected for the initial time point (CPD = 0). Mock-transduced cells were included as a control and demonstrated complete death in response to puromycin. Pellets (80e6 cells) were collected at CPDs of 0 and 14. Genomic DNA was harvested as described above. Sequencing libraries were prepared by University of Utah Genomics Core. Sequencing reads were aligned and quantified using BBtools Seal. Counts were normalized for library size and a pseudocount was added to each value. CS per gene were calculated as the mean $\log_2$-transformed fold-change in sgRNA abundance between the final time point (CPD 14) and the initial time point post-infection and antibiotic selection (CPD 0). Selectively essential genes were ranked by dCS:

$$\mathrm{dCS} \; = \; (\mathrm{CS_{EV}} - \mathrm{CS_{ATAD1+}})$$

## BH3 profiling

BH3 profiling was conducted using a FACS-based method to directly monitor cytochrome C release/retention in cells, as described previously (*Ryan and Letai, 2013*).

## Confocal microscopy

SW1088 cells transduced with EV or ATAD1-FLAG and TetON(GFP-BIM$_{EL}$ΔBH3) were seeded at a density of $3.5 \times 10^4$ cells per dish, in 35 mm Fluorodish plates (World Precision Instruments). Approximately 16 hr later, media was removed and replaced with media containing 100 ng/mL doxycycline. After 24 hr, cells were treated with 20 nM MitoTracker Red for 15 min and then imaged on a Zeiss LSM 880 confocal laser scanning microscope for 45 min in 5% $CO_2$ at 37°C. Imaging on the Zeiss LSM 880 confocal laser scanning microscope was performed with a Plan-Apochromat 63×/1.40 Oil DIC f/ELYR objective. Alternatively, cells were treated with 100 nM bortezomib for 90 min prior to imaging, were stained with MitoTracker Red as described above, and imaged for 30 min. Doxycycline concentrations were maintained throughout the staining and imaging process. All images were Airyscan processed using the Zeiss Zen Desk software.

Microscopy was conducted by an investigator (C Cunningham) who was blinded to genotype (EV vs. ATAD1) and treatment (presence or absence of bortezomib). Experiments were repeated for three independent replicates (both with and without bortezomib) and two additional replicates (without bortezomib only). At least 30 images (representing approximately 40–60 cells) were taken per condition, per replicate. GFP-positive, MitoTracker Red-negative puncta were counted using FIJI with the multi-point tool and were graphed using GraphPad Prism 9 for MacOS.

Co-localization of GFP-BIM$_{EL}$ΔBH3 and mito-mCherry was analyzed by Coloc2 package (FIJI). Regions of interest were defined by selecting entire cells but excluding nuclei using the GFP channel. All images were identically processed by background subtraction and smoothening. Each data point represents the cytoplasmic region of interest for one cell.

## Mouse xenografts

SW1088 cells (transduced with EV or ATAD1-FLAG) were grown under normal culture conditions, as described above. Cells ($3 \times 10^6$) were mixed 1:1 with Matrigel (Corning) and injected into one flank per mouse. Mice were male NOD/SCID aged 13–15 weeks. Tumor volumes were monitored biweekly using a Biopticon TumorImager. Animal experiments were conducted in accordance with The University of Utah IACUC.

PC3 cells (transduced with LentiCRISPRv2-sgNT or sgATAD1) were grown as described above. Cells ($1.8 \times 10^6$) were mixed 1:1 with Matrigel (Corning) and injected into one flank per mouse (12-week-old, male, NRG). Once tumors had established, mice were randomized into bortezomib or vehicle groups and treatment occurred by tail-vein injection twice weekly for 4 weeks. At the conclusion of the experiment, mice were sacrificed within 24 hr of receiving an IV injection. Tumors were harvested and snap-frozen on liquid nitrogen. Tumor fragments were homogenized in RIPA buffer using ceramic beads on a Omni Bead Ruptor 24 Bead Mill Homogenizer (2 cycles of 45 s at 6 m/s, 4°C). Homogenates were centrifuged (16,000 RCF, 10 min, 4°C) and supernatants were recovered, then used for downstream analysis by immunoblot. Data were analyzed by mixed effects model with Tukey's multiple comparisons. Animal experiments were conducted in accordance with The University of Utah IACUC.

## Patient data

Outcome data from patients with mCRPC were downloaded from TCGA via cBioPortal. Patients were stratified into three groups based on status of *ATAD1* and *PTEN* (unaltered vs. null). Raw data are available as a table in *Source data 3*.

## Bacterial transformation

For cloning, *Escherichia coli* DH5α competent cells (New England Biolabs) were transformed according to the manual provided by the manufacturer and grown on LB agar plates at 37°C overnight. For cloning of lentiviral and retroviral vectors, NEB Stable competent cells were used (NEB C3040I).

### E. cloni cells

For cloning, E. cloni10G competent cells were transformed according to the manual provided by the manufacturer (Lucigen) and grown on LB agar plates at 37°C overnight.

### BL21-DE3 pRIL cells

For protein expression, *E. coli* BL21(DE3) containing a pRIL plasmid and a protein expression vector were grown in terrific broth at 37°C until an $OD_{600}$ of 0.6–1.0. Cultures were induced with isopropyl-1-thio-β-D-galactopyranoside (IPTG) at a final concentration of 1 mM and grown at room temperature for an additional 3–4 hr.

## Production of soluble constructs

### Δ1-32 Msp1 and Δ1-39ATAD1

The gene encoding the soluble region of *Saccharomyces cerevisiae* Msp1 (Δ1–32) was PCR-amplified from genomic DNA and subcloned into a pET28a derivative (Novagen) encoding an N-terminal 6×His tag followed by a TEV protease cleavage site. The soluble region of *Rattus norvegicus* ATAD1 (Δ1-39) was PCR-amplified from a plasmid containing ATAD1 cDNA (GE Healthcare). All insertions and deletions were performed by standard PCR techniques. Site-specific mutagenesis was carried out by QuickChange PCR. All constructs were verified by DNA sequencing.

Plasmids encoding soluble Msp1, ATAD1, or their mutants were purified as described previously (*Wohlever et al., 2017*). Plasmids were transformed into *E. coli* BL21(DE3) containing a pRIL plasmid and expressed in terrific broth at 37°C until an $OD_{600}$ of 0.6–1.0, cultures were induced with 1 mM IPTG and grown at room temperature for an additional 3–4 hr. Cells were harvested by centrifugation, and resuspended in Msp1 Lysis Buffer (20 mM Tris pH 7.5, 200 mM KAc, 20 mM imidazole, 0.01 mM EDTA, 1 mM DTT) supplemented with 0.05 mg/mL lysozyme (Sigma), 1 mM phenylmethanesulfonyl fluoride (PMSF) and 500 U of universal nuclease (Pierce), and lysed by sonication. The supernatant was isolated by centrifugation for 30 min at 4°C at 18,500 × *g* and purified by Ni-NTA affinity chromatography (Pierce) on a gravity column. Ni-NTA resin was washed with 10 column volumes (CV) of Msp1 Lysis Buffer and then 10 CV of Wash Buffer (Msp1 Lysis buffer with 30 mM imidazole) before elution with Lysis Buffer supplemented with 250 mM imidazole. Purification of soluble ATAD1 also included the addition of ATP to stabilize the protein. ATP was added to a final concentration of 2 mM after sonication and again after elution from the nickel resin.

The protein was further purified by size exclusion chromatography (SEC) (Superdex 200 Increase 10/300 GL, GE Healthcare) in 20 mM Tris pH 7.5, 200 mM KAc, 1 mM DTT. Peak fractions were pooled, concentrated to 5–15 mg/mL in a 30 kDa MWCO Amicon Ultra centrifugal filter (Pierce) and aliquots were flash-frozen in liquid nitrogen and stored at –80°C. Protein concentrations were determined by $A_{280}$ using a calculated extinction coefficient (Expasy).

### GST-SGTA and GST-calmodulin

GST-tagged SGTA was expressed and purified as described previously (*Mateja et al., 2015*). The original calmodulin plasmid was a kind gift of the Hegde lab (*Shao and Hegde, 2011*). Calmodulin was cloned into pGEX6p1 plasmid by standard methods. GST-SGTA and GST-calmodulin were expressed as described above for soluble Msp1 constructs. Cells were harvested by centrifugation and resuspended in SGTA Lysis Buffer (50 mM HEPES pH 7.5, 150 mM NaCl, 0.01 mM EDTA, 1 mM DTT, 10% glycerol) supplemented with 0.05 mg/mL lysozyme (Sigma), 1 mM PMSF and 500 U of universal nuclease (Pierce), and lysed by sonication. The supernatant was isolated by centrifugation for 30 min at 4°C at 18,500 × *g* and purified by glutathione affinity chromatography (Thermo Fisher) on a gravity column. Resin was washed with 20 CV of SGTA Lysis Buffer and then eluted with 3 CV of SGTA Lysis Buffer supplemented with 10 mM reduced glutathione. The protein was further purified by SEC (Superdex 200 Increase 10/300 GL, GE Healthcare) in 20 mM Tris pH 7.5, 100 mM NaCl, 0.1 mM TCEP. Peak fractions were pooled, concentrated to 10 mg/mL in a 30 kDa MWCO Spin Concentrator (Pierce) and aliquots were flash-frozen in liquid nitrogen and stored at –80°C. Protein concentrations were determined by $A_{280}$ using a calculated extinction coefficient (Expasy).

## Production of membrane proteins

### BIM and Fis1

*Homo sapiens* BimL, *S. cerevisiae* Fis1 TMD ±5 flanking amino acids (residues 126–155), *H. sapiens* Bik, or *H. sapiens* Puma was cloned in place of the Sec22 TMD in the SumoTMD construct described previously (*Wang et al., 2010*; *Wohlever et al., 2017*). These constructs have N-terminal $His_6$ and 3× Flag tags and a C-terminal opsin glycosylation site (11 residues). A 3C protease site was added immediately after the His tag by standard PCR methods. The resulting constructs are $His_6$-3C-3xFlag-Sumo-thrombin-BimL-Opsin and $His_6$-3C-3xFlag-Sumo-thrombin-Fis1(126-155)-Opsin, $His_6$-3C-3xFlag-Sumo-thrombin-Bik-Opsin, and $His_6$-3C-3xFlag-Sumo-thrombin-Puma-Opsin.

Expression plasmids for SumoTMD were transformed into *E. coli* BL21(DE3) containing a pRIL plasmid and expressed in terrific broth at 37°C until an $OD_{600}$ of 0.6–0.8, cultures were induced with 0.4 mM IPTG and grown at 20°C for an additional 3–4 hr. Cells were harvested by centrifugation, and resuspended in SumoTMD Lysis Buffer (50 mM Tris pH 7.5, 300 mM NaCl, 10 mM $MgCl_2$, 10 mM imidazole, 10% glycerol) supplemented with 0.05 mg/mL lysozyme (Sigma), 1 mM PMSF and 500 U of universal nuclease (Pierce), and lysed by sonication. Membrane proteins were solubilized by addition of n-dodecyl-β-D-maltoside (DDM) to a final concentration of 1% and rocked at 4°C for 30′. Lysate was cleared by centrifugation for at 4°C for 1 hr at 35,000 × *g* and purified by Ni-NTA affinity chromatography.

Ni-NTA resin was washed with 10 CV of SumoTMD Wash Buffer 1 (50 mM Tris pH 7.5, 500 mM NaCl, 10 mM $MgCl_2$, 10 mM imidazole, 5 mM β-mercaptoethanol (BME), 10% glycerol, 0.1% DDM). Resin was then washed with 10 CV of SumoTMD Wash Buffer 2 (same as Wash Buffer 1 except with 300 mM NaCl and 25 mM imidazole) and 10 CV of SumoTMD Wash Buffer 3 (same as Wash Buffer 1 with 150 mM NaCl and 50 mM imidazole) and then eluted with 3 CV of SumoTMD Elution Buffer (same as Wash Buffer 3 except with 250 mM imidazole).

The protein was further purified by SEC (Superdex 200 Increase 10/300 GL, GE Healthcare) in 50 mM Tris pH 7.5, 150 mM NaCl, 10 mM $MgCl_2$, 5 mM BME, 10% glycerol, 0.1% DDM. Peak fractions were pooled and concentrated in a 30 kDa MWCO spin concentrator (Pierce). Sample was then incubated with 3C Protease at a 1:100 ratio at 4°C overnight to remove the His tag. The following day, the sample was run over Ni-NTA resin equilibrated in Lysis Buffer to remove 3C protease, His tag, and uncleaved proteins. Flow through was collected, aliquoted, and flash-frozen in liquid nitrogen and stored at –80°C. Protein concentrations were determined by $A_{280}$ using a calculated extinction coefficient (Expasy).

### Msp1

Full-length *S. cerevisiae* Msp1 was PCR-amplified from genomic DNA, subcloned into a pET21b derivative with a C-terminal 6×His tag, and expressed as described above for the soluble constructs. Cells were lysed by sonication and the insoluble fraction was harvested by centrifugation for 1 hr at 4° C at 140,000 × *g*. After resolubilizing for 16 hr in Msp1 Lysis Buffer containing 1% DDM (Bioworld), the detergent-soluble supernatant was isolated by centrifugation for 45 min at 142,000 × *g* and purified by Ni-NTA affinity chromatrography and SEC as described above for the soluble constructs, except that all buffers contained 0.05% DDM. Peak fractions were concentrated in 100 kDa MWCO Amicon Ultra centrifugal filter (Millipore). Protein concentrations were determined by $A_{280}$ using a calculated extinction coefficient (Expasy) and aliquots were flash-frozen in liquid nitrogen.

## Reconstitution of Msp1 activity in proteoliposomes

### Liposome preparation

Liposomes mimicking the lipid composition of the yeast OMM were prepared as described (*Kale et al., 2014*). Briefly, a 25 mg lipid film was prepared by mixing chloroform stocks of chicken egg phosphatidyl choline (Avanti 840051C), chicken egg phosphatidyl ethanolamine (Avanti 840021C), bovine liver phosphatidyl inositol (Avanti 840042C), synthetic DOPS (Avanti 840035C), and synthetic TOCL (Avanti 710335C) at a 48:28:10:10:4 molar ratio with 1 mg of DTT. Nickel liposomes were made as described above, except, 1,2-dioleoyl-*sn*-glycero-3-[*N*-((5-amino-1-carboxypentyl)iminodiacetic acid)succinyl] Nickel salt (Avanti 790404) was used at a molar ratio of 2% and DOPS was dropped from 10% to 8%.

Chloroform was evaporated under a gentle steam of nitrogen and then left on a vacuum (<1 mTorr) overnight. Lipid film was resuspended in Liposome Buffer (50 mM HEPES KOH pH 7.5, 15% glycerol, 1 mM DTT) to a final concentration of 20 mg/mL and then subjected to five freeze-thaw cycles with liquid nitrogen. Liposomes were extruded 15 times through a 200 nm filter at 60°C, distributed into single-use aliquots, and flash-frozen in liquid nitrogen.

## Proteoliposome preparation

For extraction assays with full-length Msp1, proteoliposomes were prepared by mixing 1 µM Msp1, 1 µM TA protein (SumoTMD), and 2 mg/mL of mitochondrial liposomes in Reconstitution Buffer (50 mM HEPES KOH pH 7.5, 200 mM potassium acetate, 7 mM magnesium acetate, 2 mM DTT, 10% sucrose, 0.01% sodium azide, and 0.1% deoxy big chaps). For extraction assays with soluble Msp1/ATAD1, proteoliposomes were prepared by mixing 1 µM TA protein (SumoTMD), and 2 mg/mL of Nickel liposomes in Reconstitution Buffer. Detergent was removed by adding 25 mg of biobeads and rotating the samples for 16 hr at 4°C. After removing biobeads, unincorporated TA protein was pre-cleared by incubating the reconstituted material with excess (5 µM) GST-SGTA and GST-calmodulin and passing over a glutathione spin column (Pierce #16103); the flow through was collected and used immediately for dislocation assays.

## Extraction assay

Extraction assays contained 60 µL of pre-cleared proteoliposomes, 5 µM GST-SGTA, 5 µM calmodulin, and 2 mM ATP and the final volume was adjusted to 200 µL with Extraction Buffer (50 mM HEPES KOH pH 7.5, 200 mM potassium acetate, 7 mM magnesium acetate, 2 mM DTT, 0.1 µM calcium chloride). Samples were incubated at 30°C for 35 min and then loaded onto a glutathione spin column. Columns were washed 4× with Extraction Buffer and eluted with the same buffer supplemented with 20 mM glutathione pH 8.5. Samples were loaded onto stain-free gels, imaged, and then transferred to a PVDF membrane and blotted as indicated in the Key resources table. To account for variability in reconstitution efficiency and western blotting, a new reconstitution and dislocation assay with WT Msp1 was done in parallel with each mutant Msp1. Figures are representative of N>3 separate reconstitutions. Note that the 'input' lane is diluted 5× relative to the 'elution' lane.

## Quantification and statistical analysis

To account for variability in reconstitution efficiency and western blotting, a new reconstitution and dislocation assay with WT Msp1 was done in parallel with each Msp1 mutant. Figures are representative of N>3 separate reconstitutions. Dislocation efficiency was quantified by comparing the amount TA protein in the 'elution' lane with the amount of substrate in the 'input' lane.

## Acknowledgements

The authors acknowledge University of Utah Core Facilities, particularly James Marvin, PhD, and the Flow Cytometry Core, David Lum, PhD, and the Preclinical Research Resource, the DNA/Peptide Synthesis Core, Brian Dalley, PhD, and the High-Throughput Genomics Core, as well as the Whitehead Institute Genome Technology Core Facility. We thank Keren Hilgendorf for equipment. We thank Derick Torres for technical lab support. We thank Florian Muller for helpful conversations and insight. We thank Jeff Morgan, Sarah Fogarty, and other members of the Rutter lab for helpful discussions and comments on the manuscript.

## Additional information

### Competing interests

Jacob M Winter, Jared Rutter: has filed a patent related to this work. Reference: WO2021/257910. The other authors declare that no competing interests exist.

## Funding

| Funder | Grant reference number | Author |
|---|---|---|
| National Institutes of Health | 1F30CA243440 | Jacob M Winter |
| National Institutes of Health | 1T32DK11096601 | Jordan Berg |
| National Institutes of Health | 1F99CA253744 | Jordan Berg |
| National Institutes of Health | 5T32DK091317 | Corey N Cunningham |
| National Institutes of Health | 1F32GM140525 | Corey N Cunningham |
| National Institutes of Health | K00CA212445 | Alex Bott |
| National Institutes of Health | R35GM137904 | Matthew L Wohlever |
| National Institutes of Health | CA228346 | Jared Rutter |
| National Institutes of Health | R35GM131854 | Jared Rutter |
| Howard Hughes Medical Institute | | Jared Rutter |

The funders had no role in study design, data collection and interpretation, or the decision to submit the work for publication.

## Author contributions

Jacob M Winter, Conceptualization, Data curation, Formal analysis, Funding acquisition, Validation, Visualization, Methodology, Writing – original draft, Writing – review and editing; Heidi L Fresenius, Heather R Keys, Formal analysis, Investigation, Methodology; Corey N Cunningham, Jordan Berg, Alex Bott, Formal analysis, Investigation; Peng Wei, Jeremy Ryan, Sheryl R Tripp, Paige Barta, Investigation; Tarun Yadav, Deepika Sirohi, Data curation, Formal analysis; Neeraj Agarwal, Resources, Data curation; Anthony Letai, David M Sabatini, Resources; Matthew L Wohlever, Conceptualization, Resources, Supervision, Funding acquisition, Methodology, Writing – review and editing; Jared Rutter, Conceptualization, Resources, Supervision, Funding acquisition, Writing – review and editing

## Author ORCIDs

Jacob M Winter  http://orcid.org/0000-0001-7152-183X
Heather R Keys  http://orcid.org/0000-0003-1371-2288
Alex Bott  http://orcid.org/0000-0003-2273-8922
Jeremy Ryan  http://orcid.org/0000-0002-3327-1283
Matthew L Wohlever  http://orcid.org/0000-0002-9406-3410
Jared Rutter  http://orcid.org/0000-0002-2710-9765

## Ethics

All of the animals were handled according to approved institutional animal care and use committee (IACUC protocol # 18-11004) protocols of the University of Utah. Every effort was made to minimize suffering.

## Decision letter and Author response

Decision letter https://doi.org/10.7554/eLife.82860.sa1
Author response https://doi.org/10.7554/eLife.82860.sa2

# Additional files

## Supplementary files
• MDAR checklist

- Source data 1. Gene-level data from Jurkat CRISPR screen, used to make *Figure 1E*.
- Source data 2. Gene-level data from HGC27 CRISPR screen, used to make *Figure 1G*.
- Source data 3. Raw survival data from patient with metastatic, castrate-resistant prostate cancer (mCRPC), used to make *Figure 4D*.

## Data availability

All data and source data generated or analyzed are included as supplementary files. CRISPR screening data and human mCRPC survival data are provided as supplementary files.

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

# Appendix 1

### Appendix 1—key resources table

| Reagent type (species) or resource | Designation | Source or reference | Identifiers | Additional information |
|---|---|---|---|---|
| Cell line (human) | HEK293T cells | ATCC | #CRL-11268, RRID:CVCL_1926 | |
| Cell line (human) | Jurkat E6.1 | ATCC | TIB-152 | |
| Cell line (human) | H4 | ATCC | HTB-148 | |
| Cell line (human) | RPMI-7951 | ATCC | HTB-66 | |
| Cell line (human) | SW1088 | ATCC | HTB-12 | |
| Cell line (human) | PC3 | ATCC | CRL-1435 | |
| Cell line (human) | HGC27 | HGC27 | 94042256 | |
| Antibody | Anti-Flag Mouse mAB | Sigma-Aldrich | #F7425, RRID:AB_439687 | 1:5000 |
| Antibody | Anti-V5 Mouse mAB | Abcam | #ab9116, RRID:AB_307024 | 1:5000 |
| Antibody | Anti-GFP Rabbit mAB | Cell Signaling | #2956S | 1:5000 |
| Antibody | Anti-ATAD1 Mouse mAB | NeuroMab | 75–157 | 1:1000 |
| Antibody | Anti-PTEN Rabbit mAB | CST | #9188 | 1:1000 |
| Antibody | Anti-beta actin Rabbit mAB | CST | #4970 | 1:20,000 |
| Antibody | Anti-alpha tubulin Mouse mAB | CST | #3873 | 1:20,000 |
| Antibody | Anti-MCL1 Rabbit mAB | CST | 94296 | 1:1000 |
| Antibody | Anti-BCLXL Rabbit mAB | CST | 2764 | 1:1000 |
| Antibody | Anti-pBIM(Ser69) Rabbit mAB | CST | 4585 | 1:1000 |
| Antibody | Anti-pBIM(Ser77) Rabbit mAB | CST | 12433 | 1:1000 |
| Antibody | Anti-pBIM(Thr112) Rabbit mAB | Thermo Fisher | PA5-64655 | 1:1000 |
| Antibody | Anti-NRF1/TCF11 Rabbit mAB | CST | 8052 | 1:1000 |
| Antibody | Anti-BID Rabbit mAB | CST | 2002 | 1:1000 |
| Antibody | Anti-MAVS Rabbit mAB | CST | 24930 | 1:1000 |
| Antibody | Anti-MFF Rabbit mAB | Abcam | AB129075 | 1:1000 |
| Antibody | Anti-FIS1 Rabbit mAB | Abcam | AB156856 | 1:1000 |
| Antibody | Anti-BIM Rabbit mAB | CST | #2933 | 1:1000 |
| Antibody | Anti-BAK Rabbit mAB | CST | #12105 | 1:1000 |

*Appendix 1 Continued on next page*

*Appendix 1 Continued*

| Reagent type (species) or resource | Designation | Source or reference | Identifiers | Additional information |
|---|---|---|---|---|
| Antibody | Anti-ubiquitin Rabbit mAB | CST | #43124 | 1:1000 |
| Antibody | Anti-ubiquitin Mouse mAB | Abcam | #ab7254 | 1:1000 |
| Antibody | Anti-GAPDH Mouse mAB | CST | #97166S | 1:5000 |
| Antibody | Anti-PARP Rabbit mAB | CST | #9532 | 1:1000 |
| Antibody | Anti-Caspase 3 Rabbit mAB | CST | #14220S | 1:1000 |
| Antibody | Goat Anti-Mouse IgG (H&L) Antibody Dylight 800 Conjugated | Rockland | #610-145-002-0.5 | 1:10,000 |
| Antibody | Donkey anti-Rabbit IgG (H+L) Highly Cross-Adsorbed Secondary Antibody, Alexa Fluor 680 | Invitrogen | #A10043 | 1:10,000 |
| Antibody | Goat Anti-Mouse IgG (H+L) Antibody, Alexa Fluor 680 Conjugated | Invitrogen | #A21057, RRID:AB_141436 | 1:10,000 |
| Antibody | Donkey Anti-Rabbit IgG (H&L) Antibody Dylight 800 Conjugated | Rockland | #611-145-002-0.5, AB_11183542 | 1:10,000 |
| Antibody | Goat anti-Mouse IgG (H+L), Superclonal Recombinant Secondary Antibody, HRP | Thermo Fisher | #A28177, RRID:AB_2536163 | 1:10,000 |
| Antibody | Goat anti-Rabbit IgG (H+L), HRP | ProteinTech | RRID:AB_2722564 | 1:10,000 |
| Recombinant DNA reagent | psPAX2 | Addgene | | |
| Recombinant DNA reagent | pMD2.G | Addgene | | |
| Recombinant DNA reagent | pSpCas9(BB)–2A-GFP (PX458) | Addgene | | |
| Recombinant DNA reagent | LentiCRISPRv2GFP-sgNT | Addgene | | |
| Recombinant DNA reagent | Px458_sgATAD1_1 | This study | sgRNA targeting ATAD1 in Px458 vector; see Materials and methods | |
| Recombinant DNA reagent | Px458_sgATAD1-2 | This study | sgRNA targeting ATAD1 in Px458; see Materials and methods | |
| Recombinant DNA reagent | LRCherry2.1-sgMARCH5_10 | This study | sgRNA targeting MARCH5 in LRCherry2.1; see Materials and methods | |
| Recombinant DNA reagent | LRCherry2.1-sgPCNA | Addgene | | |
| Recombinant DNA reagent | LRCherry2.1-sgAAVS1 | Addgene | | |
| Recombinant DNA reagent | pLenti-Blast | Addgene | | |
| Recombinant DNA reagent | pQCXIP | Clontech | | |
| Recombinant DNA reagent | pQCXIP-ATAD1-FLAG/HA | *Chen et al., 2014* | | |
| Recombinant DNA reagent | pQCXIP-ATAD1^E193Q-FLAG/HA | *Chen et al., 2014* | | |
| Recombinant DNA reagent | pQCXIP-mito-mCherry | This study | Cox8-MTS upstream of mCherry, in PQCXIP | |

*Appendix 1 Continued on next page*

*Appendix 1 Continued*

| Reagent type (species) or resource | Designation | Source or reference | Identifiers | Additional information |
|---|---|---|---|---|
| Recombinant DNA reagent | pQCXIP-mRFP-SKL | This study | SKL amino acids fused to C-terminus of mRFP in pQCXIP | |
| Recombinant DNA reagent | pLenti-Blast-ATAD1-FLAG | This study | ATAD1 CDS with C-terminal FLAG tag cloned into pLenti-Blast | |
| Recombinant DNA reagent | pLVX-TetOne-Puro | Takara | | |
| Recombinant DNA reagent | pLVX-TetOne-Puro-GFP | This study | GFP CDS cloned into pLVX-TetOne-Puro | |
| Recombinant DNA reagent | pLVX-TetOne-Puro-GFP-BIM$_{EL}$ | This study | GFP-BIM$_{EL}$ fusion cloned into pLVX-TetOne-Puro | |
| Recombinant DNA reagent | pLVX-TetOne-Puro-GFP-BIM$_{EL}$ΔBH3 | This study | GFP-BIM$_{EL}$ fusion with 3 amino acids in BH3 domain mutated, cloned into pLVX-TetOne-Puro | |
| Recombinant DNA reagent | pLenti-GFP-Puro | Addgene | | |
| Recombinant DNA reagent | pLenti-Myc-MCL1 | This study | MCL1 with N-terminal Myc tag swapped with GFP in pLenti-GFP-Puro | |
| Recombinant DNA reagent | Brunello CRISPR knockout sgRNA library | Addgene | | |
| Recombinant DNA reagent | pET21: Msp1-His (*S. cerevisiae*) | *Wohlever et al., 2017* | | |
| Recombinant DNA reagent | pET28: His-TEV-Δ1–32-Msp1 (*S. cerevisiae*) | *Wohlever et al., 2017* | | |
| Recombinant DNA reagent | pET28: His-TEV-Δ1-39-ATAD1 (*R. norvegicus*) | This study | Wohlever Lab; see Materials and methods | |
| Recombinant DNA reagent | pET28: His-Flag-Sumo-Sec22 TMD-Opsin | *Wang et al., 2010* | | |
| Recombinant DNA reagent | pET28: His-Flag-Sumo-BimL-Opsin | This study | Wohlever Lab; see Materials and methods | |
| Recombinant DNA reagent | pET28: His-Flag-Sumo-Fis1 TMD-Opsin | This study | Wohlever Lab; see Materials and methods | |
| Recombinant DNA reagent | pET28: His-Flag-Sumo-Bik-Opsin | This study | Wohlever Lab; see Materials and methods | |
| Recombinant DNA reagent | pET28: His-Flag-Sumo-Puma-Opsin | This study | Wohlever Lab; see Materials and methods | |
| Recombinant DNA reagent | pGEX6p1: GST-SGTA | *Mateja et al., 2015* | | |
| Recombinant DNA reagent | pGEX6p1: GST-Calmodulin | *Shao and Hegde, 2011* | | |
| Commercial assay or kit | Pierce BCA | Thermo | 23225 | |
| Commercial assay or kit | CellTiterGlo Luminescent Viability Assay | Promega | G7572 | |
| Chemical compound, drug | DDM | GoldBio | DDM25 | |
| Chemical compound, drug | Lipofectamine 3000 | Thermo Fisher | L3000008 | |
| Chemical compound, drug | Lipofectamine RNAiMAX | Invitrogen | 13778150 | |

*Appendix 1 Continued on next page*

*Appendix 1 Continued*

| Reagent type (species) or resource | Designation | Source or reference | Identifiers | Additional information |
|---|---|---|---|---|
| Chemical compound, drug | MitoTracker Red CMXRos | Invitrogen | M7512 | |
| Chemical compound, drug | LysoTracker Blue DND-22 | Invitrogen | L7525 | |
| Chemical compound, drug | MitoTracker Deep Red FM | Invitrogen | M22426 | |
| Chemical compound, drug | Crystal Violet | Sigma | C0775 | |
| Chemical compound, drug | RIPA Buffer | Cell Signaling | 9806 | |
| Chemical compound, drug | SE Cell Line 4D-Nucleofector X Kit L | Lonza | V4XC-1012 | |
| Chemical compound, drug | Adenosine Triphosphate | Acros Organics | AC10280-0100 | |
| Chemical compound, drug | Bovine liver phosphatidyl inositol | Avanti | 840042C-10mg | |
| Chemical compound, drug | Synthetic DOPS | Avanti | 840035C-10mg | |
| Chemical compound, drug | Synthetic DOGS-Ni-NTA | Avanti | 790404C-5mg | |
| Chemical compound, drug | Chicken egg phosphatidyl ethanolamine | Avanti | 840021C-25mg | |
| Chemical compound, drug | Chicken egg phosphatidyl choline | Avanti | 840051C-200mg | |
| Chemical compound, drug | Synthetic TOCL | Avanti | 710335C-25mg | |
| Chemical compound, drug | Bortezomib | EMD Millipore | 5043140001 | |
| Chemical compound, drug | Carfilzomib | Selleck Chem | S2853 | |
| Chemical compound, drug | Marizomib | Selleck Chem | S7504 | |
| Chemical compound, drug | zVAD-FMK | Sigma-Aldrich | V116 | |
| Software, algorithm | metap | Michael Dewey, 2020 | | |
| Software, algorithm | R | R Core Team | | |
| Software, algorithm | Ggplot2 | Wickham, 2009 | | |
| Other | SuperSep PhosTag precast gels, 12.5% ac | Wako/Fujifilm | 195-17991 | |

