## [Editor Report]

The authors identify co-deletion of the mitochondrial AAA+ ATPase ATAD1 with the tumor suppressor PTEN as a factor modifying cancer prognosis, based on a new mechanism of increasing sensitivity to proteotoxic stress induced by proteasome inhibition. The authors also identify the mitochondrial E3 ubiquitin ligase MARCH5 as a gene whose deletion is synthetically lethal with ATAD1. These findings suggest that the use of proteasome-targeting agents may be useful in patients with tumors dually deleted for ATAD1 and PTEN. The study is based on convincing evidence and makes an innovative contribution to the understanding of the biology of tumors with 10q23 deletions.

---

## [Decision Letter]

**Decision letter after peer review:**

Thank you for submitting your article "Co-deletion of ATAD1 with PTEN sensitizes cancer cells to protein stress" for consideration by *eLife*. Your article has been reviewed by 2 peer reviewers, one of whom is a member of our Board of Reviewing Editors, and the evaluation has been overseen by Erica Golemis as the Senior Editor. The reviewers have opted to remain anonymous.

Essential revisions:

The reviewers took note of the fact that this manuscript previously underwent two rounds of review; reviews and responses to the reviews were included with the submission, clarifying the evolution of the study based on very extensive commentary. The opinion of the reviewers was that the main requirement for revision is the tempering of some claims in the absence of new experiments addressing the points summarized below, together with some experiments requested as controls and clarifications of mechanistic points. The authors are strongly encouraged to address the points made by the reviewers as recommendations, at the bottom of this review.

*Reviewer #1 (Recommendations for the authors):*

1. The manuscript is mainly about ATAD1 depletion and how that sensitizes cells to proteasome inhibition. Maybe the title should better reflect that? I understand the inclusion of PTEN in the title reflects the co-deletion of the two genes, but since that isn't really studies, I'd recommend changing to better focus on ATAD1 deletion being the critical determinant in dictating sensitivity to proteasome inhibition (protein stress may be a bit broad as well).

2. It would be useful to include more arrows to highlight the puncta across different conditions in EV9-12. This is included in one panel, but considering the fact that puncta are also reported to be observed in other panels where ATAD1 is overexpressed, it would be useful to include more arrows to guide the reader.

3. The phos-tag gels are convincing in showing alterations in BIM phosphorylation by ATAD1 status, but the phosphor-specific antibodies were more easily visualized for readers not familiar with looking at phos-tag gels. Is there a reason why these antibodies were not utilized more to follow phosphorylation?

4. I'm confused by Figures 3D and E16A. In both cases, it appears that PC3 cells are depleted of ATAD1 and then treated with increasing concentrations of proteasome inhibitors. However, in Figure 3D ATAD1 depletion reduces the sensitivity of viability to bortezomib, while no effect is observed in E16A under the same conditions. Shouldn't those conditions be the same? If yes, why aren't they? Other experiments support the importance of ATAD1 depletion on btz sensitivity in PC3 cells (e.g., Figure 3G PARP cleavage).

5. The authors indicate a 'trend' for increased BIM levels in ATAD1-depleted xenografts treated with BTZ (line 311). While it may look that way in the representative blot (Figure 4C), this is not really the case in Figure EV17D. There could be changes in phosphorylation (not measured), but it seems like a bit of a stretch to indicate any strong evidence for increases in BIM.

6. It would be useful to include a figure showing the expression of ATAD1 across different cell models used to confirm various statements in the text.

7. Similarly, an immunoblot confirming depletion of ATAD1 and/or MARCH5 in the synthetic lethality studies would be valuable.

*Reviewer #2 (Recommendations for the authors):*

1. In Figure 1, the π shows in the initial panels that ATAD1 is co-deleted with PTEN in a significant percentage of prostate and other solid tumors. Then, beginning in Figure 1D, the authors present data from the Jurkat T-cell lymphoblastic leukemia line. This raises several questions that should be addressed. First, what is the ATAD1/PTEN deletion pattern in lymphoblastic lymphomas? Second, why switch from prostate and other solid tumor types to Jurkat cells? At present, the rationale for the study and the relevance of the Jurkat versus the solid tumor findings is unclear.

2. In Figure 1F the authors do a rescue experiment, replacing ATAD1-FLAG with HGC27 ATAD1-null cells. While this is a useful experiment, the results may not be comparable to knocking out ATAD1 in a cell model that evolved in the presence of this protein, because of selective pressure differences. This may affect interpretations.

3. As *eLife* has more relaxed word limits on articles, and appeals to a general audience, it would be useful to add text around lines 116 and following explaining what the process of OMM extraction is, in greater detail. Similarly, a sentence around line 128 introducing BH3-only proteins versus other family members would be useful. Finally, including a figure to summarize the model for ATAD1 action at the end of the manuscript would also be a good idea, to increase clarity.

4. The discussion in the paragraph beginning at line 131 about the relationship of MARCH5, ATAD1, and BIM, is confusing. It seems as if multiple mechanisms are being proposed for ATAD1 to influence BIM. In particular, the idea that ATAD1 is regulating phosphorylation of BIM-EL through ERK is stated; but, this seems as if it might be a very indirect or secondary process. This statement would benefit from more data to address the mechanism.

5. Line 53 – the sentence regarding deletion in polyclonal BIM knockout cells is unclear.

6. Returning to the original observation, ATAD1 is deleted selectively in the context of PTEN deletion. More information is needed about the nature of deletions – that is, if both PTEN and ATAD1 are deleted, how extensive is the deletion (versus a PTEN-only deletion)? Are other genes pertinent to the mechanism under study also being deleted?

7. The role of PTEN loss is pertinent to some of the interpretations of the mechanisms of ATAD1 action, where it is not clear whether experiments are being performed in parallel in PTEN-positive versus PTEN-null settings. This is particularly important, given the discussion focuses on proteotoxic stress, and there is ample literature demonstrating that loss of PTEN also contributes to proteotoxic stress. This causes one to wonder what might inhibition of PI3K do to some of the synthetic lethal interactions observed in PTEN-deficient cells, with or without loss of PTEN.

---

## [Author Response]

Reviewer #1 (Recommendations for the authors):1. The manuscript is mainly about ATAD1 depletion and how that sensitizes cells to proteasome inhibition. Maybe the title should better reflect that? I understand the inclusion of PTEN in the title reflects the co-deletion of the two genes, but since that isn't really studies, I'd recommend changing to better focus on ATAD1 deletion being the critical determinant in dictating sensitivity to proteasome inhibition (protein stress may be a bit broad as well).

This is an excellent suggestion by the reviewer. In response, we have changed the title of the manuscript to “Collateral deletion of the mitochondrial AAA+ ATPase ATAD1 sensitizes cancer cells to proteasome dysfunction.”

2. It would be useful to include more arrows to highlight the puncta across different conditions in EV9-12. This is included in one panel, but considering the fact that puncta are also reported to be observed in other panels where ATAD1 is overexpressed, it would be useful to include more arrows to guide the reader.

We thank the reviewer for this suggestion. We have now added many more arrows to highlight the puncta in various panels. Adding arrows to indicate each individual punctum on some images becomes quite distracting, so we hope that our pointing out more examples of puncta will help the reader see them throughout the images.

3. The phos-tag gels are convincing in showing alterations in BIM phosphorylation by ATAD1 status, but the phosphor-specific antibodies were more easily visualized for readers not familiar with looking at phos-tag gels. Is there a reason why these antibodies were not utilized more to follow phosphorylation?

We thank the reviewer for this helpful critique. BIM phosphorylation can be either activating (pro-death) or neutralizing (pro-survival), depending on the residue. There are multiple phosphoresidues (e.g. Ser69, Ser77) that neutralize BIM, and ATAD1 activity increases the phosphorylation of each of them. Alternatively, phosphorylation at Thr112 stimulates pro-death activity of BIM, and this phosphorylation is unaffected by the presence or absence of ATAD1. Once we used phospho-specific antibodies to determine these findings, we used phos-tag gels to show overall phosphorylation of BIM, since BIM exists in mono, di, and poly phosphorylated states. Using phos-tag gels, we are able to detect BIM in its multiply phosphorylated states, which gives an additional layer of information. Additionally, using phos-tag gels we can detect phosphorylated and un-phosphorylated states on the same gel, enabling a clearer picture of the ratio of active:inactive BIM.

4. I'm confused by Figures 3D and E16A. In both cases, it appears that PC3 cells are depleted of ATAD1 and then treated with increasing concentrations of proteasome inhibitors. However, in Figure 3D ATAD1 depletion reduces the sensitivity of viability to bortezomib, while no effect is observed in E16A under the same conditions. Shouldn't those conditions be the same? If yes, why aren't they? Other experiments support the importance of ATAD1 depletion on btz sensitivity in PC3 cells (e.g., Figure 3G PARP cleavage).

We neglected to indicate statistical significance on the plot in E16A and we thank the reviewer for pointing out this oversight. In E16A, as in Figure 3D, we see that deletion of ATAD1 sensitizes PC3 cells to bortezomib treatment. There is some minor variability in the exact shapes of the curves based on when the experiment was done, normalization, batch of bortezomib, batch of FBS, etc., but the phenotype is completely consistent. As pointed out by the reviewer, we also confirm this phenotype using cPARP and in vivo tumor growth. In Author response image 1, we include both independent repeats (3 biological replicates each) for the sgBIM experiments, showing all 4 groups on the top row and then the same data but with the sgBIM groups omitted on the bottom row. As we hope the reader can appreciate, we used the more conservative plot for the figure in our paper (leftmost plot). We modified the figure in our manuscript to include an indication of statistical significance.

**Author response image 1. sa2fig1:** 

5. The authors indicate a 'trend' for increased BIM levels in ATAD1-depleted xenografts treated with BTZ (line 311). While it may look that way in the representative blot (Figure 4C), this is not really the case in Figure EV17D. There could be changes in phosphorylation (not measured), but it seems like a bit of a stretch to indicate any strong evidence for increases in BIM.

This is an accurate critique and we have now removed any suggestion of an increase in BIM levels in the western blots from the xenografts. The sentence now reads: “Bortezomib treatment induced a significant increase in NOXA levels in sgATAD1 tumors, but not in sgNT tumors (Figure 4C; Figure 4 Supplement 1).”

6. It would be useful to include a figure showing the expression of ATAD1 across different cell models used to confirm various statements in the text.

We have now included a western blot of ATAD1 and PTEN across cell types used, which is Figure 1 Supp 3 in the revised version of the manuscript. Additionally, we routinely probe for ATAD1 and include those nearly every western blot in the paper, based on abundance of lysate.

7. Similarly, an immunoblot confirming depletion of ATAD1 and/or MARCH5 in the synthetic lethality studies would be valuable.

This is now demonstrated by western blot in Figure 2 Supp 2D in the revised manuscript.

Reviewer #2 (Recommendations for the authors):1. In Figure 1, the π shows in the initial panels that ATAD1 is co-deleted with PTEN in a significant percentage of prostate and other solid tumors. Then, beginning in Figure 1D, the authors present data from the Jurkat T-cell lymphoblastic leukemia line. This raises several questions that should be addressed. First, what is the ATAD1/PTEN deletion pattern in lymphoblastic lymphomas? Second, why switch from prostate and other solid tumor types to Jurkat cells? At present, the rationale for the study and the relevance of the Jurkat versus the solid tumor findings is unclear.

We thank the reviewer for this comment and have added a sentence explaining rationale: “Jurkat cells were chosen as an experimentally tractable system that has been validated in genetic screening and also as a cell line with background PTEN deficiency.” Part of why we used Jurkat cells is that we performed this initial screen in 2018, when CRISPR screens were not commonly performed in adherent cell lines. At that time, Jurkat cells were the best option as a well-validated screening model (e.g. Birsoy et al. 2014, Birsoy et al. 2015), and had the bonus of being a PTEN-deficient background. By using a PTEN-deficient background and deleting ATAD1 (with comparison to ATAD1^WT^), we were able to isolate the variable of ATAD1 deficiency.

We used Jurkat cells as a model system rather than with the intent to study biology intrinsic to T-ALL. For the sake of completeness, we analyzed *ATAD1* and *PTEN* deletions in T-ALL. We used a dataset from Liu et al. (*Nat Genet* 2017) that characterized the genomic landscape of TALL using 264 samples. We defined shallow deletions and deep deletions in the standard fashion. All told, we found that *ATAD1* is deep deleted in 1.1% of T-ALL and has shallow deletions in 1.5%. *PTEN* is also deleted at a lower frequency in this disease compared to other cancer types, with 3% deep deletion and 5.7% shallow deletion. The raw values are shown in Author response table 1, with a total of n = 264 samples used in the analysis.

**Author response table 1. sa2table1:** 

Gene	Deep Deletion	Shallow Deletion	Amplification	Gain	No CNAs
*PTEN*	*8*	*15*	*5*	*3*	*233*
*ATAD1*	*3*	*4*	*5*	*3*	*249*

2. In Figure 1F the authors do a rescue experiment, replacing ATAD1-FLAG with HGC27 ATAD1-null cells. While this is a useful experiment, the results may not be comparable to knocking out ATAD1 in a cell model that evolved in the presence of this protein, because of selective pressure differences. This may affect interpretations.

We agree with the reviewer that the results of deleting *ATAD1* in cells that evolved with it is not the same as re-expressing *ATAD1* in cells that evolved without it. In fact, that is why we took both approaches to ask the question of: What genetic vulnerabilities are present in a cell that lacks ATAD1? When used together, we think that these two approaches complement each other and enable us to isolate the variable of ATAD1 proficiency vs. deficiency. Indeed, we agree with the reviewer that taking only one of these approaches (i.e. only studying WT vs. KO cells) would risk biasing results based on the evolutionary history of that cell line. That we found the same genetic interaction with ATAD1 and MARCH5 using two fundamentally different approaches strengthens our confidence in the external validity of this finding. As described in our manuscript: “Therefore, we conducted an additional genetic screen in a different cellular context to gain a broader perspective on how *ATAD1* deficiency creates synthetic lethal vulnerabilities.”

3. As eLife has more relaxed word limits on articles, and appeals to a general audience, it would be useful to add text around lines 116 and following explaining what the process of OMM extraction is, in greater detail. Similarly, a sentence around line 128 introducing BH3-only proteins versus other family members would be useful. Finally, including a figure to summarize the model for ATAD1 action at the end of the manuscript would also be a good idea, to increase clarity.

We thank the reviewer for the pointing out this oversight in our manuscript, and offering a helpful suggestion to expand on key points in our manuscript. Our resubmission includes added background information on OMM protein extraction. First, we expanded the introduction to two paragraphs, the second of which provides more detail on ATAD1-dependent OMM protein extraction. Second, we added the following sentence when we describe ATAD1/MARCH5 synthetic lethality: “Thus, MARCH5/p97 and ATAD1 are known to represent two parallel pathways by which OMM proteins are removed from mitochondria.”

We also added the following sentence to help give background on BH3-only proteins: “BH3-only proteins serve as sentinels for cellular stress and, upon activation, initiate mitochondrial outer membrane permeabilization.”

4. The discussion in the paragraph beginning at line 131 about the relationship of MARCH5, ATAD1, and BIM, is confusing. It seems as if multiple mechanisms are being proposed for ATAD1 to influence BIM. In particular, the idea that ATAD1 is regulating phosphorylation of BIM-EL through ERK is stated; but, this seems as if it might be a very indirect or secondary process. This statement would benefit from more data to address the mechanism.

We apologize for unclear writing. Indeed, we propose that ATAD1 can influence BIM activity through multiple mechanisms, regulating both the degradation and phosphorylation of BIM. To be clear, ATAD1 only has one action on BIM –extraction– but there are multiple fates of BIM thereafter. It is clear that ATAD1 promotes the phosphorylation of BIM-EL, but, as the reviewer suggests, this must be an indirect process since ATAD1 is not a kinase. Instead, we propose that ATAD1 extracts BIM-EL and this facilitates subsequent phosphorylation by kinases such as ERK. We know that loss of ATAD1 decreases BIM-EL phosphorylation and that ATAD1 can directly extract unphosphorylated BIM from reconstituted proteoliposomes, which both argue that ATAD1-mediated extraction is upstream of phosphorylation of BIM.

5. Line 53 – the sentence regarding deletion in polyclonal BIM knockout cells is unclear.

We thank the reviewer for this critique. We have edited the sentence to read (changes in bold):

“This effect was similar when we deleted *MARCH5* in polyclonal BIM (*BCL2L11*) knockout cells (Figure 2D; generated by stably expressing Cas9 and sgRNA targeting *BCL2L11*).”

6. Returning to the original observation, ATAD1 is deleted selectively in the context of PTEN deletion. More information is needed about the nature of deletions – that is, if both PTEN and ATAD1 are deleted, how extensive is the deletion (versus a PTEN-only deletion)? Are other genes pertinent to the mechanism under study also being deleted?

This is an excellent question. We have returned to our genomic datasets and better characterized the deletions at Chr10q23.31 using metastatic prostate cancer patient data (the same datasets used for our survival analysis in figure 4; virtual study: https://bit.ly/3VS0eud). We have generated a figure to outline these data, whereby we plot deletion frequency of each gene within ≈2 Mb of *PTEN* and overlay with a CNV plot. In this plot (which is now included in the manuscript as Figure 1 Supp 2), the darkness of horizontal blue bars directly correlates with depth of deletion of the corresponding region of the chromosome, where white bars indicate no deletion, light bars indicate partial/shallow deletion, and dark blue indicates complete/deep deletion. It is immediately apparent that the most frequently deleted gene in this region is, as expected, *PTEN* (approximately 27.5% of samples in this dataset). As one examines genes either proximal or distal to *PTEN* on the chromosome, the frequency of deep deletion decreases. In this dataset, *ATAD1* is deleted at a frequency of 18%. On the other side of *PTEN* (in the 3’ direction of *PTEN*), *RNLS* is deleted at a frequency of 21.7%. At a distance of approximately 2 Mb from PTEN, the deletion frequency decreases to 3.5% (*GRID1*) or 3.3% (*PANK1*). We now mention these other genes in our manuscript: “These deletions most frequently span the 0.5 Mb surrounding *PTEN*, leading to collateral deletion of *KLLN* and *RNLS* in addition to *ATAD1* (Figure 1 Supplement 2)”. Whether these other genes that are co-deleted with PTEN influence the biology of these tumors is an interesting and valid question, but we unfortunately cannot address it within the scope of the current manuscript.

7. The role of PTEN loss is pertinent to some of the interpretations of the mechanisms of ATAD1 action, where it is not clear whether experiments are being performed in parallel in PTEN-positive versus PTEN-null settings. This is particularly important, given the discussion focuses on proteotoxic stress, and there is ample literature demonstrating that loss of PTEN also contributes to proteotoxic stress. This causes one to wonder what might inhibition of PI3K do to some of the synthetic lethal interactions observed in PTEN-deficient cells, with or without loss of PTEN.

The reviewer presents an interesting question, but it is outside the scope of the current study. We focus on the background of PTEN deletion because that is the context in which ATAD1 is deleted in cancer. Regarding the contributions of PTEN deletion to our findings, it is possible that the dual deletion of ATAD1 and PTEN make cells even more sensitive to proteasome inhibition than the loss of either alone. Even so, we clearly and repeatedly demonstrate that loss of ATAD1 in and of itself increases sensitivity to proteasome inhibition. Interestingly, there is some evidence in yeast that Msp1 deletion (yeast homolog of ATAD1) is synthetic sick with Rpn10 deletion (encoding a subunit of the proteasome; Basch et al., 2020 MBoC), suggesting that there is evolutionary conservation of this general principle, and making it less likely that this is a phenotype specific to PTEN-null cells.

[Editors' note: we include below the reviews that the authors received from another journal, along with the authors’ responses.]

Referee #1 (Remarks to the Author):The authors have made the interesting and unexpected finding that co-deletion of the outer mitochondrial membrane (OM) AAA+ ATPase ATAD1 with the tumour suppressor PTEN correlates with improved cancer patient survival. Their study proposes that the “passenger/hitchhiker” deletion of ATAD1 in PTEN null cancers confers a vulnerability of cancer cells to apoptosis. ATAD1 extracts tail anchored proteins from the OM, for example during quality control to remove mistargeted proteins, and is known to be neuroprotective, however Winter et al. present a novel role for ATAD1 in the regulation of apoptosis. Mechanistically, they show that one function of ATAD1 is to extract the pro-apoptotic BH3-only protein BIM from the OM. The potential for inhibiting ATAD1 activity in cancers (also synergistically with proteasomal inhibitors) make these intriguing findings of potentially broad relevance.My main concern surrounds the conclusion that ATAD1 is cytoprotective in cancer specifically via BIM extraction. The expression of ATAD1 in SW1088 cells is sufficient to induce xenograft tumour growth. This is consistent with the in vitro data but I feel more evidence is required to support the model that it is specifically an ATAD1-BIM anti-apoptotic axis that supports tumour growth. Especially considering the partial rescue of ATAD1 KO cell viability in absence of BIM (Figure 2C, D). Do BIM1 depleted SW1088 cells also grow as tumours? Ideally, a BIM construct that cannot be recognised by ATAD1 could be used but this is admittedly difficult to achieve. It would be worthwhile to perform xenograft experiments in tumour forming cells that have endogenous ATAD1 expression and deplete ATAD1 to try and block tumour growth. If so, is this BIM dependent? This is required to show that ATAD1 deficiency decreases tumour fitness via BIM and would be more relevant for future potential clinical application. Of note, the authors did not further study the specificity of ATAD1 and it appears likely that additional substrates of ATAD1 exist in the OM.

We thank the reviewer for their insightful comments and suggestions. We are glad to hear that the reviewer considers our findings “intriguing” and “of potentially broad relevance.” The reviewer states that their main concern “surrounds the conclusion that ATAD1 is cytoprotective in cancer specifically via BIM extraction.” We have taken several steps to address this issue. The first set of experiments address the importance of BIM in the genetic relationship of *ATAD1* and *MARCH5*. We demonstrate that the synthetic lethality of *ATAD1* and *MARCH5* can be partially rescued by deletion of BIM (*BCL2L11*). This result indicates that, although there may be multiple mechanisms that contribute to the lethal phenotype of losing both *ATAD1* and *MARCH5*, BIM is an important component of that phenotype.

In related new experiments, we showed that deletion of BIM protects *ATAD1*∆ cells, but not WT cells, from subsequent *MARCH5* deletion. Therefore, BIM deletion is particularly beneficial to *ATAD1*-deficient cells, compared to WT cells. This result suggests that, in WT cells, ATAD1 suppresses BIM activity – but when ATAD1 is absent, BIM activity is elevated. Indeed, we have included several examples of direct evidence that loss of ATAD1 activates BIM. In the absence of ATAD1, BIM levels increase, BIM phosphorylation decreases, and ectopic expression of BIM is more toxic.

Next, we took a complementary approach to study the functional relationship of ATAD1 and BIM more thoroughly. We reasoned that if BIM contributes to the lethal phenotype of *ATAD1*/*MARCH5* double deletion, then neutralizing BIM in alternative ways should also rescue this phenotype. In order to test this prediction of our model, we needed a way to neutralize BIM other than directly knocking out BIM. To that end, we overexpressed *MCL1*, which encodes a pro-survival protein that directly inhibits BIM. As expected, overexpression of *MCL1* also partially rescued *ATAD1*/*MARCH5* double deletion. Together, these results reveal that neutralizing BIM is a key function of ATAD1 in the context of protein homeostasis stress on the outer mitochondrial membrane (*MARCH5* deletion).

It is clear, however, that the pro-survival effects of ATAD1 cannot be solely explained by BIM extraction, since *MCL1* overexpression was actually *more* protective than BIM deletion (Figure 2E vs. D; Extended Data Figure 5C vs. B). MCL1 can inhibit factors downstream of BIM, such as BAX and BAK, so it could be that ATAD1 can extract other BH3-only proteins that converge on BAX/BAK activation. Consistent with this notion, BIM deletion shows modest to no rescue of proteasome inhibitor hypersensitivity of *ATAD1* knockout cells, even though caspase inhibition completely rescues the phenotype. We therefore conclude that although ATAD1 directly and specifically antagonizes BIM, ATAD1 likely also extracts other substrates that are relevant in different contexts.

To assess whether ATAD1 has inherent specificity for BIM as a substrate, we compared the ability of ATAD1 to extract BIM versus other BH3-only proteins in our in vitro extraction assay. As we demonstrated in our initial submission, ATAD1 directly extracts BIM from lipid membranes in an ATP-dependent manner. Remarkably, ATAD1 did not extract BIK or PUMA under the same conditions. In considering which other factors ATAD1 might extract to explain its pro-survival function, BIK and PUMA were prime candidates. They are both tail-anchored, and have both been described as important for apoptosis triggered by proteotoxic stress. That we ruled out these two important candidates strongly supports the conclusion that BIM is a key ATAD1 substrate, especially in the context of proteotoxic stress-induced apoptosis.

NOXA is the other candidate BH3-only protein that one might propose to be extracted by ATAD1 in these contexts, since NOXA is activated by proteasome inhibition and directly inhibits MCL1. However, multiple lines of evidence convinced us that NOXA is not a direct substrate of ATAD1. First, NOXA does not have a bona fide transmembrane domain, so it is difficult to envision how it could be a substrate of ATAD1 (a membrane protein extractase). Second, *ATAD1* re-expression is strongly protective against proteasome inhibition in SW1088 cells, despite NOXA being undetectable in this cell line (western blot shown in Author response image 2). Third, although *ATAD1* loss increases NOXA levels in Jurkat cells (Figure 2C, Extended Data Figure 5A), this is not seen in PC3 cells (Figure 4C, Extended Data Figure 16C,D).

Despite our ruling out all of the prime suspects, the genetic data suggest that other factors beyond BIM are involved in the pro-survival function of ATAD1. To acknowledge that ATAD1 likely acts through proteins in addition to BIM, we have now revised the manuscript to more precisely indicate that while BIM is clearly an important and direct substrate of ATAD1, there are likely other substrates that are relevant to the cytoprotective function of ATAD1. We state the following in our Results section:“Although caspase inhibition completely rescued the bortezomib phenotype of ATAD1deficient cells, the same was not true for deletion of BIM in PC3 cells, which means that there must be other factors in addition to BIM that mediate this phenotype (Extended Data Figure 16A,B). We examined several OMM-localized or tail-anchored proteins in PC3 cells with and without bortezomib, and reduced BIM phosphorylation was the only consistent change caused by *ATAD1* deletion (Extended Data Figure 16C). Other BH3-only proteins reported to be activated by proteotoxic stress include BIK, PUMA, and NOXA, but our in vitro extraction assay ruled out a direct action of ATAD1 to extract these proteins. Increased mitochondrial fragmentation in ATAD1-null cells^2^ could be one potential explanation, since mitochondrial dynamics are intimately connected to apoptosis. Nonetheless, these results indicate that the protective effects of ATAD1 during proteasome inhibition can be explained exclusively by limiting apoptosis, with BIM extraction playing a key role.”

The identities of these other hypothetical substrates are unknown. Deficiency of *ATAD1* has been previously shown to increase mitochondrial fragmentation (Chen et al., EMBOJ 2014), so it may be that ATAD1 extracts substrates that regulate mitochondrial dynamics. Changes in mitochondrial fission and fusion have an established role in regulating the apoptotic threshold, which has been known and studied for nearly 20 years (Youle and Karbowski, Nat Rev Mol Cell Bio, 2005). Our lab and others have shown that ATAD1 is an evolutionarily conserved protein quality control factor, so this more generic protective function could also contribute to the prosurvival role of ATAD1 in the context of protein stress. Given the strong synthetic lethality with *MARCH5* across multiple cell types, we are confident in our initial focus on a direct role in the regulation of apoptosis.

“The authors did not further study the specificity of ATAD1”

The revised manuscript now includes exciting new data regarding the specificity of ATAD1, where we compare BIM with other related BH3-only proteins BIK and PUMA using our in vitro extraction assay (Figure 2J,K). As described in more detail in the previous section, we found that ATAD1 is unable to extract BIK or PUMA, though it can extract BIM under the same conditions. This demonstrates an inherent substrate selectivity of ATAD1 for BIM, even compared to other tail-anchored proteins of the same family. In combination with the genetic and other data, we conclude that BIM is a specific, direct and important substrate of ATAD1.

“Do BIM depleted SW1088 cells also grow as tumours?”

We agree that this would be an interesting and potentially valuable experiment to show in yet another context that BIM is partially responsible for the pro-survival effects of ATAD1. After consultation with the editor, we jointly concluded that this was beyond the scope of the manuscript. More importantly, the original purpose of the SW1088 xenograft experiment was to test the hypothesis that ATAD1 deficiency sensitized tumors to proteasome inhibition– that the ATAD1-deficient SW1088 cells never engrafted precluded us from even testing that hypothesis. Therefore, we wanted to test our model in a system in which we could study established tumors lacking *ATAD1* and how they respond to treatment, since this most closely resembles the translational potential for our work. To that end, we conducted another xenograft experiment, this time using PC3 cells transduced with Cas9 + non-targeting sgRNA or sgRNA targeting *ATAD1*. In this context, loss of *ATAD1* was compatible with tumor engraftment (Figure 4B). We treated mice harboring *ATAD1*-proficient (sgNT) or deficient (sgATAD1) PC3 tumors with saline or bortezomib, and found that bortezomib slowed the growth of sgATAD1 tumors specifically (Figure 4B), while having no effect on the sgNT tumors (Figure 4A). These data represent a strong in vivo validation of our model in which ATAD1 protects cells from apoptosis induced by proteotoxic stress, which has direct implications for hundreds of thousands of cancer patients whose tumors have Chr10q23 deletion.

Further to this, the BIM levels barely change in the SW1088 cells expressing ATAD1-Flag compared to EV (Figure Extended Data 5 A). Can the authors comment on this and quantify BIM levels in these cells? ATAD1 protein levels could be shown more often in key experiments when comparing EV and ATAD1 expressing cells e.g in the SW1088 cells used for xenografts.

BIM protein levels increase upon loss of *ATAD1* in many cell contexts (for example, Figure 2A, 2B, Extended Data Figure 16C). Nevertheless, we have made the reproducible observation that ATAD1 regulates BIM via mechanisms beyond simply degradation of the protein. Specifically, we have added new experiments in the revised manuscript using high-resolution imaging, which demonstrates ATAD1 promotes the non-mitochondrial localization of BIM (Extended Data Figure 9-12). BIM is known to be inactivated by changes in localization (Kale, Osterlund, and Andrews, *Cell Death & Differentiation* 2018), but how it translocates from the OMM to the cytosol was previously unknown. BIM requires OMM localization to bind and inhibit anti-apoptotic proteins, such as MCL1, and to activate the pore-forming proteins BAX/BAK. Hence, BIM localization is a key parameter that dictates its pro-apoptotic activity.

Finally, we have made the exciting new discovery that ATAD1 promotes the phosphorylation of BIM_EL_ by cytosolic kinases (Figure 2B; Extended Data Figure 4A,B; Extended Data Figure 13). Phosphorylation of BIM_EL_ has been clearly established as an important mechanism by which kinases like ERK promote cell survival. Phosphorylation of BIM_EL_ reduces its toxicity by decreasing its binding affinity for pro-survival proteins like MCL1, preventing its activation of BAX, and accelerating its degradation (e.g., PMID: 17151701; PMID: 29775995; PMID: 18498746). We now show that *ATAD1* loss decreases BIM_EL_ phosphorylation, while reexpressing *ATAD1* increases BIM_EL_ phosphorylation, and identify residues Ser69 and Ser77 as being phosphorylated in this context (Figure 2B; Extended Data Figure 4A,B; Extended Data Figure 13A,D-F). Of course, ATAD1 is not a kinase, so our model is that ATAD1 extracts BIM and, by so doing, puts it into a biochemical environment that promotes its phosphorylation by kinases such as ERK and Aurora A/B. In the context of proteasome inhibition, ATAD1-null cells accumulate unphosphorylated (active) BIM, while ATAD1 positive cells accumulate phosphorylated (inactive) BIM (Extended Data Figure 13A,D-F).

We propose that ATAD1 extraction of BIM from the OMM prevents it from activating apoptosis, and subsequent phosphorylation of BIM enforces its inactivation via multiple mechanisms. We think this may partly explain how ATAD1 is protective against apoptosis even in the context of proteasome inhibition – if the sole fate of ATAD1-extracted substrates were immediate proteasomal degradation, then ATAD1 would be inconsequential in the context of an inhibited proteasome (e.g., bortezomib treatment). Instead, we clearly demonstrate that ATAD1 is most important to cell fitness in the context of ubiquitin-proteasome system dysfunction (Figure 1E, G; Figure 2D,E; Figure 3; Figure 4A,B).

Other comments:The authors observe genetic interaction between MARCH5 and ATAD1 indicating that they may act in parallel pathways. They argue that this supports the importance of BIM accumulation in ATAD1 KO cells but otherwise did not analyse any further the role of MARCH5 for the pathway. The model in Figure 4F only vaguely links the function of MARCH5 to ATAD1 and BIM extraction. How does the mild accumulation of BIM1 in ATAD1 KO cells (Figure 2A) compare with MARCH5 KO? Do the authors observe any additive effect upon loss of both proteins at a molecular level? Without further analysis of MARCH5 function in this pathway, the observation of a genetic interaction does not add much to the described apoptotic role of ATAD1.

We thank the reviewer for this comment and agree that this part of the manuscript was under-developed in our initial submission. We have since made significant progress in this area. First, we directly validated the *ATAD1/MARCH5* synthetic lethal interaction using focused CRISPR deletion experiments (Figure 1G; Figure 2C-E). Second, we conducted an additional, complementary CRISPR-based screen using HGC27 cells (Del10q23, *ATAD1*-null gastric cancer cell line) re-expressing *ATAD1* or transduced with the empty vector. Remarkably, the top hit for synthetic lethality with *ATAD1* in this second screen was *MARCH5*. We discuss the rationale behind this screen in more detail below. Third, we demonstrate that BIM deletion or *MCL1* overexpression partially rescues this synthetic phenotype (Figure 2D,E; Extended Data Figure 5). This genetic experiment demonstrates that BIM is partially responsible for *ATAD1*/*MARCH5* synthetic lethality. That the rescue is incomplete indicates that there must be other factors involved, which will be important to study in the future. We speculate that processes independent of apoptosis could also contribute to the functional relationship of ATAD1/MARCH5, such as mitochondrial fission. For instance, our lab and others have demonstrated that loss of *ATAD1* in human cells, *Atad1* in mouse embryonic fibroblasts, or *msp1∆* in yeast can lead to fragmented mitochondria (Chen et al., EMBOJ 2014). Increased mitochondrial fission has a well-established role in promoting apoptosis and could have other implications for cell fitness as manifested by changes in mitophagy, nutrient metabolism, and other pathways.

These experiments showed us that our initial model was incomplete, since the main effect of *MARCH5* deletion in this context was an increase of NOXA, rather than BIM. It is clear that loss of *MARCH5* increases BIM in some cell types (Arai et al. *eLife* 2020), but apparently not in Jurkat cells, which we used for our mechanistic follow up experiments. Based on our new data, we have revised our model to propose that MARCH5 inhibits NOXA, while ATAD1 inhibits BIM, and these two pathways converge on MCL1. Thus, co-deletion of *ATAD1* and *MARCH5* triggers a concomitant increase of BIM and NOXA, which combine to inhibit MCL1 and activate apoptosis (schematic in Figure 4G).

We conducted an additional genome-wide CRISPR-based synthetic lethality screen using a gastric cancer cell line, HGC27, which harbors the Chr10q23 deletion seen in cancer patients. *ATAD1* is deleted in 4.1% of gastric cancer cases. We elaborate on the relevance of cell lines from different cancer types further below.

We transduced HGC27 cells with lentiviral *ATAD1* or empty vector and conducted CRISPR-based screens using the Brunello sgRNA library. We performed an sgRNA enrichment/depletion screen as we had done before in Jurkat cells. The top hit for synthetic lethality with *ATAD1* was again *MARCH5* (Figure 1F,G). It is remarkable that we identified the same genetic interaction despite all of the differences between these two screens. Notably, with Jurkat cells we screened WT cells vs. *ATAD1-*knockout clones, while with HGC27 cells we screened Del10q23 cells +/- *ATAD1* re-expression. By pairing these different approaches, we controlled for: presence/absence of other genes that are co-deleted on Chr10q23; how cells might evolve differently if *ATAD1* deletion is an early event in tumorigenesis rather than an experimental perturbation in cell culture; and clonality of knockout cell lines. We highlight the other important differences between the two screens here in Author response table 2.

**Author response table 2. sa2table2:** 

	Jurkat Screen	HGC27 Screen
Tissue of origin	Blood; T-ALL	Gastric cancer
Growth substrate	Suspension	Adherent
Media	RPMI1640	EMEM
sgRNA library	Sabatini/Lander (10 sgRNA/gene)	Brunello (4 sgRNA/gene)
Location	Whitehead Institute	University of Utah
Year	2018	2021
CNA (Del)	CDKN2A/B, MSH2, MSH6	
CNA (Amp)	TERT	MYC, AKT1
Mutations	BAX,	PIK3CA, APC, JAK1

In an ideal world, we would have used a prostate cancer cell line, but there are no prostate cancer cell lines available that harbor Del10q23. We could have used PC3 cells with *ATAD1* deleted via CRISPR, but we felt that there was more to gain by doing the screen in a cell line with Chr10q23 deletion, which includes other co-deleted genes and does not simply recapitulate our Jurkat screen in a different cell type.

If one considers all cancer types based on two variables, frequency of *ATAD1* deletion and patient mortality, then gastric cancer is the second most “relevant” cancer type, behind only prostate cancer. Gastric cancer accounts for nearly 800,000 deaths worldwide each year and *ATAD1* is deleted in 4.1% of cases; Approximately 32,000 patients die every year from *ATAD1*deficient gastric cancer (GLOBOCAN 2020). For perspective, this is more than half of the total number of patients who die of melanoma annually (i.e., *all* deaths attributable to melanoma, irrespective of *ATAD1* status).

As a result of these issues, we consider the screens done using Jurkat and HGC27 cell lines as nicely complementary systems to study *ATAD1* deletion in cancer more generally, and we suggest that pairing these two very different contexts (knockout vs. re-expressing in a Del10q23 cell) is a powerful and rigorous approach.

– The qualitative assessment of apoptosis using example blots of PARP cleavage e.g. Figure 1I and Figure 2C and F are not convincing as a single measure of apoptosis, especially without quantification. Quantification of the lipid extraction assays, currently shown as representative examples, are also missing (which the authors indicate in methods have been done N>3 times). Similarly, quantification of MCL1 levels in Figure 2A, B is missing. Here, the analysis of steady state levels of other tail-anchored OM proteins would also be of interest.

We thank the reviewer for this suggestion. In the revised manuscript, we have now added multiple quantitative measures of cell viability, including CellTiterGlo and crystal violet staining followed by elution (Figure 2D,E,G; Extended Data Figure 5B,C; Extended Data Figure 6A, C; Extended Data Figure 15A-F; Figure 3A-D,H; Extended Data Figure 15 C-F; Extended Data Figure 16A). We confirm that changes in viability represent apoptosis by rescuing with caspase inhibitors (Figure 3H; Extended Data Figure 15C-F), BIM deletion (Figure 2D; Extended Data Figure 6D), or MCL1 overexpression (Extended Data Figure 6A,C). These rescue experiments functionally validate the conclusion that *ATAD1* loss promotes intrinsic apoptosis, which is roughly defined as caspase dependent cell death activated by mitochondrial outer membrane permeability.

In the revised manuscript, we have now added quantification of the liposome extraction assays, which represent n = 6 independent experiments (Figure 2K). These data clearly demonstrate that ATAD1 extracts BIM in a reconstituted proteoliposome setting. We demonstrate that ATAD1 extraction of BIM requires ATP, and that even in the presence of ATP, ATAD1 cannot extract the other structurally related tail-anchored proteins BIK and PUMA. Thus, our data demonstrates that ATAD1 extracts BIM from proteoliposomes in a direct and specific manner.

Western blots of BCL2, TA proteins, and other OMM proteins have been added for Jurkat cells (Figure 2C, Extended Data Figure 5A), cultured PC3 cells (Extended Data Figure 16C,D), and PC3 xenograft tumor lysates (Figure 4C). We demonstrate that loss of *ATAD1* leads to increased BIM (Figure 2A-C, Extended Data Figure 5A; Extended Data Figure 13D; Extended Data 16C). MCL1 and NOXA levels are higher in *ATAD1*-deficient Jurkat cells (Figure 2C; Extended Data Figure 5A) but not in cultured PC3 cells (Extended Data Figure 16C,D) nor in PC3 xenografts (Figure 4C; Extended Data Figure 17C,E). As suggested by the reviewer, we also examined steady state abundance of several tail-anchored and/or OMM proteins in *ATAD1*-proficient or *ATAD1*KO PC3 cells: BIM_EL_ (including phosphorylation status), BIM_L_, BIM_S_, MCL1, BCLXL, FIS1, MFF, MAVS, and NOXA (Extended Data Figure 16C,D). We observed modest increases in steady state BIM_EL_ levels upon deletion of *ATAD1* in PC3 cells, accompanied by a substantial decrease in phosphorylation, which we believe to be important for ATAD1 function as described above. Abundance of MAVS, MFF, FIS1, BCLXL, NOXA, BID, and MCL1 were unchanged upon deletion of *ATAD1* in PC3 cells. BCL2, BIK, and PUMA were undetectable by western blot in PC3 cells. We conclude from these extensive data that ATAD1 has a clear and consistent effect on BIM, but not on other members of the BCL2 family or other OMM proteins.

– Figure 1H, I should include the complementation of ATAD1 KO cells with WT and catalytic dead ATAD1 or at least using the additional ATAD1 KO clone used for screening.

Although both Jurkat clones show synthetic lethality with *MARCH5*, and both clones are hypersensitive to BIM overexpression, our recent experiments show that they differ in that the second KO clone is insensitive to the MCL1 inhibitor AMG176. Indeed, other cell lines also show inconsistency in response to AMG176, where ATAD1 protects HGC27 cells but not PC3 cells from this agent. Because of this inconsistency we have decided to omit the experiments involving AMG176. These experiments are superfluous now that we have direct genetic experiments demonstrating that deletion of *MARCH5* is synthetic lethal with *ATAD1* loss, and that this is partialy rescued by either BIM deletion or *MCL1* overexpression. We deem it to be likely that this inconsistency in AMG176 response probably reflects selective pressure in cultured cells to circumvent apoptosis, which could be achieved by upregulating other antiapoptotic proteins like BCL2 or BCLXL. Again, we now emphasize our findings of *MARCH5* synthetic lethality as opposed to sensitization to AMG176, since this genetic interaction is seen in both Jurkat using gene deletion clones and in our complementary CRISPR-based screen in HGC27 cells.

– What happens to the GFP-BIMEL-deltaBH3 construct in ATAD1 KO cells vs WT? Is there an accumulation on the mitochondria? Also, can the authors comment on the mitochondrial localization of GFP-BIMELdeltaBH3 in EV vs ATAD1 cells in Figure 3A. It looks to me as if it accumulates on mitochondria to similar extent in both conditions.

The revised manuscript includes multiple experiments that address this set of issues. First, we have demonstrated that GFP-BIMEL∆BH3 (BIMEL with the BH3 deleted, expressed as a fusion with GFP) shows increased localization to cytoplasmic puncta in ATAD1+ cells, while this localization is shifted to mitochondria in *ATAD1*-null cells. We quantified this effect using the coloc2 image quantification tool in FIJI and the data is now shown in Extended Data Figure 10B.

As described above, we also discovered that BIM is phosphorylated on Ser69 and Ser77

(and other residues) in a manner that is dependent on *ATAD1*. To be clear, we are not proposing that ATAD1 is a kinase, but rather that extraction by ATAD1 facilitates phosphorylation of BIM by cytoplasmic kinases, such as ERK1/2, which phosphorylates Ser59, Ser69, and Ser77. Phosphorylation of BIM_EL_ has been established as an important mechanism by which pro-survival pathways suppress apoptosis (e.g., PMID: 17151701; PMID: 29775995; PMID: 18498746).

Thus, our revised model provides mechanistic detail into the relationship between ATAD1 and BIM. ATAD1 directly extracts BIM from mitochondria and, by so doing, facilitates its phosphorylation, which is known to neutralize its pro-apoptotic activity via multiple mechanisms. It is possible that this phosphorylation step is important to prevent retargeting of BIM to mitochondria, which could generate a futile cycle of extraction and re-insertion. This will be an important topic to study in the future. The ATAD1-dependent extraction of BIM in the context of proteasome inhibition leads to BIM localization to cytoplasmic puncta.

– The efficiency of the immunoprecipitation in Figure 2I is difficult to assess, as input and precipitate are not analysed and shown on the same SDS-PAGE.

We have now repeated these experiments as well as added complementary approaches to demonstrate the direct interaction of ATAD1 and BIM. These experiments demonstrate that the efficiency of co-immunoprecipitation is low, which is exactly what would be predicted for a transient interaction between ATAD1 and its substrate BIM. Nonetheless, we reproducibly observed a physical interaction between ATAD1-FLAG and GFP-BIM_EL_.

More importantly, we have now included new data demonstrating that endogenous BIM co-precipitates ATAD1-FLAG (Extended Data Figure 6E), providing further evidence of a physical interaction in cells.

When combined with the data from the reconstituted biochemical extraction assay system, which clearly demonstrates specific extraction of BIM by ATAD1, these data provide strong evidence that BIM is a direct extraction substrate of ATAD1. These data dovetail perfectly with the genetic data that demonstrate the importance of BIM in the anti-apoptotic effects of ATAD1. We demonstrate that the pro-survival function of ATAD1 depends on BIM with genetic experiments, and we demonstrate that ATAD1 can directly and specifically extract BIM from membranes using biochemical approaches. Together, these results strongly support the model that one mechanism whereby ATAD1 protects cells from apoptosis induced by ubiquitin proteasome system dysfunction is the extraction and neutralization of BIM.

Referee #2 (Remarks to the Author):In the manuscript by Winter et al., the authors report that loss of ATAD1, which is adjacent to and frequently codeleted with PTEN, predisposes cancer cells to apoptosis and correlates with improved survival in cancer patients. ATAD1, an AAA-ATPase, is a major protein dislocase of the mitochondrial outer membrane and peroxisomes that extracts single-spanning membrane proteins from these organelles (Trends Cell Biol. 2021; S0962-8924). The authors tried to connect the death-regulatory activity of ATAD1 to its ability to extract the pro-apoptotic BIM protein from mitochondria. Although the link between ATAD1 and BIM is interestingly, the study appears relatively preliminary and the data do no fully support the conclusions drawn. It remains to be determined whether all the phenotypes associated with ATAD1 loss are entirely caused by BIM extraction and inactivation. The following suggestions might help the authors further strengthen their conclusions.

We thank the reviewer for their constructive review of our manuscript. We are encouraged that the reviewer finds the work “interesting,” and we now submit a much-improved revision that addresses each of the reviewer’s concerns.

We now present biochemical and genetic data convincingly demonstrating that BIM is a critically important substrate to mediate the cytoprotective function of ATAD1. First, genetic screens in two different cell line contexts (Jurkat and HGC27 cells, more on this below) independently demonstrate a synthetic lethal genetic interaction between *ATAD1* and *MARCH5*, which is known to protect cells from apoptosis via degradation of BIM and NOXA (Figure 1 D-G). Second, we found that deletion of BIM (*BCL2L11*) partially rescues the synthetic lethal phenotype of *ATAD1* and *MARCH5*, suggesting that BIM is an important downstream mediator of this genetic interaction (Figure 2C,D). Third, we demonstrate that *ATAD1* deletion increases the steady-state abundance of BIM in some cellular contexts (Figure 2A-C). Fourth, we demonstrate that endogenous levels of ATAD1 protect cells from a challenge with ectopic BIM expression (Figure 2F,G). Together with the BIM deletion experiments (Figure 2C,D) these results demonstrate that BIM is sufficient and partially necessary to explain the anti-apoptotic effects of ATAD1. Fifth, we demonstrate that ATAD1 and BIM physically interact in cells through reciprocal coimmunoprecipitation experiments: ATAD1-FLAG co-immunoprecipitates GFP-BIM_EL_, and endogenous BIM co-immunoprecipitates ATAD1-FLAG (Extended Data Figure 6D,E). Sixth, we demonstrate that ATAD1 relocalizes GFP-BIM from the OMM to cytoplasmic puncta in live cell imaging (Extended Data Figure 9, 10, 11, 12). Seventh, we show that ATAD1 promotes BIM phosphorylation on serine 69 and 77, and likely other residues, which is known to neutralize the pro-apoptotic activity of BIM (Figure 2B; Extended Data Figure 4A,B; Extended Data Figure 13A-E). Last, we demonstrate that ATAD1 can directly extract BIM from membranes in a reconstituted proteoliposome system. Remarkably, we find that ATAD1 specifically extracts BIM, but not two other structurally-related BH3-only proteins, PUMA or BIK (Figure 2J,K).

Together, these data strongly support a model in which ATAD1 protects cells from apoptosis by directly extracting and neutralizing BIM.

Nonetheless, it is clear that BIM extraction cannot account for all of the cytoprotective effects of ATAD1. We and others have described a general role of ATAD1 in mitochondrial protein quality control (Chen et al., EMBOJ 2014), and this function could be particularly important in the context of proteasome inhibition. Moreover, loss of *ATAD1* leads to increased mitochondrial fragmentation in cultured cells from humans and mice (Chen et al., EMBOJ 2014), and mitochondrial fragmentation has been shown to promote apoptosis. As a membrane protein extractase, ATAD1 could influence an array of diverse processes in the cell. However, our complementary genetic screens gave a clear signal that *MARCH5* and apoptosis are the most differentially essential processes in *ATAD1*-proficient and deficient states. Moreover, we demonstrate unequivocally that direct and specific extraction of BIM is an important part of how ATAD1 protects cells from apoptosis. This advance represents a completely novel substrate of ATAD1 and a new mechanism of apoptotic regulation. With our work, BIM becomes the only tail-anchored protein that has been shown to interact with ATAD1 at endogenous levels and the only tail-anchored protein to be directly extracted by ATAD1 using a reconstituted proteoliposome assay. Furthermore, that ATAD1 directly and specifically extracts BIM demonstrates a novel cellular role of ATAD1 as an anti-apoptotic protein, with direct implications for many thousands of cancer patients.

Specific Comments:1. Given that most BCL^-^2 family proteins have a similar C-terminal transmembrane domain as BIM, the authors should exclude the possibility that ATAD1 may also extract other BCL^-^2 family proteins from mitochondria and regulate apoptosis through these proteins. The extraction assay shown in Figure 2J should examine other BCL^-^2 family proteins. Similarly, the immunoblots in Figures 2A and S5A should include other BCL^-^2 family proteins.

We thank the reviewer for this insightful point regarding the structural similarities of

BCL2 family proteins. We conducted extensive experiments to directly address this concern.

As suggested, we purified other BH3-only proteins and examined the ability of ATAD1 to extract them from liposomes using our extraction assay. Since PUMA, BIK, and NOXA have all been proposed to mediate apoptosis triggered by different proteotoxic stressors, we examined each of these three proteins. Remarkably, ATAD1 was unable to extract PUMA or BIK, although it extracted BIM under identical conditions. This result indicates inherent substrate selectivity of ATAD1 for BIM, even amongst these structurally related proteins (Figure 2J,K). While we do not completely understand the structural features underlying the specificity of ATAD1 for BIM over PUMA and BIK, we suspect that it has to do with an intrinsically disordered region N-terminal to the TMD in BIM (Extended Data Figure 8). This will be an important area for future research.

We were unable to stably integrate NOXA into proteoliposomes, precluding it from analysis. The inability of NOXA to integrate into proteoliposomes is consistent with a report by Andreu-Fernández et al. (*JBC* 2016) demonstrating that NOXA does not have a bona fide transmembrane domain. As a result, we deem it unlikely that ATAD1, a membrane protein extractase, could act directly on NOXA.

Western blots of BCL2 family members, TA proteins, and other OMM proteins have been added for Jurkat cells (Figure 2D, Extended Data Figure 5A), cultured PC3 cells (Extended Data Figure 16C), and PC3 xenograft tumor lysates (Figure 4C; Extended Data Figure 17). We demonstrate that loss of ATAD1 leads to increased BIM (Figure 2A-C, Extended Data Figure 5A; Extended Data Figure 13B; Extended Data 16C). MCL1 and NOXA levels are higher in ATAD1-deficient Jurkat cells (Figure 2A,C; Extended Data Figure 5A) but not in cultured PC3 cells (Extended Data Figure 16C) nor in PC3 xenografts (Figure 4C; Extended Data Figure 19C,E). As suggested by the reviewer we examined steady state abundance of several tail-anchored and/or OMM proteins in *ATAD1*proficient or KO PC3 cells: BIM_EL_ (including assessment of phosphorylation status), BIM_L_, BIM_S_, MCL1, BCLXL, FIS1, MFF, MAVS, and NOXA (Extended Data Figure 16C,D). We observed modest increases in steady state BIM_EL_ levels upon deletion of ATAD1 in PC3 cells, accompanied by a substantial decrease in phosphorylation (Extended Data Figure 13D,E). Levels of MAVS, MFF, FIS1, BCLXL, NOXA, BID, and MCL1 were unchanged upon deletion of ATAD1 in PC3 cells. BCL2, BIK, and PUMA were undetectable by western blot in PC3 cells. We conclude from these data that ATAD1 has clear and consistent effects on BIM, but not on other members of the BCL2 family and other OMM proteins.

2. The data shown in Figure 2H that re-expression of wild-type ATAD1 protected cells from MOMP induced by BIM BH3 peptide (lacking the C-terminal transmembrane domain) would argue against the involvement of extraction activity of ATAD1. Alternatively, it may suggest that other BCL^-^2 family proteins are extracted by ATAD1, leading to the protection against BIM peptide-mediated cytochrome c release. The BH3 profiling should include BID BH3 peptide as a control. Furthermore, the authors should confirm comparable expression of ATAD1WT and ATAD1E193Q in these cells.

We appreciate that interpretation of the BH3 profiling assay can be confusing when viewed in the context of our other experiments. We have now included a much more explicit and clear description and interpretation. To summarize, we do not think that ATAD1 acts directly on the BH3 peptide derived from BIM. Rather, the data suggest that ATAD1 acts on endogenous BIM, which changes the apoptotic priming “set-point” underlying the response to challenge with exogenous BIM BH3 peptide. There are pools of pro and anti-apoptotic proteins on the OMM, and the relative abundances and activation states of these proteins dictates how a cell will respond to further pro-apoptotic stimuli, such as treatment with a BH3 peptide. Increased activity (de-phosphorylation, localization, and abundance) of BIM due to the loss of *ATAD1* will “fill up” the reservoirs of anti-apoptotic proteins and make the cell more sensitive to exogenous challenge with BIM peptide. This experiment simply demonstrates that the absence of ATAD1 primes cells for apoptosis, using an assay that is widely used in the field. Although these data support the overarching hypothesis, in the revised manuscript we have deemphasized this assay to help focus the attention of the reader on more easily interpretable data that also strongly support the model.

As requested, we added western blots of ATAD1^WT^ and ATAD1^E193Q^ (Extended Data Figure 6B), which demonstrate that the E193Q mutant is actually expressed at higher steady-state levels than ATAD1^WT^, as we have seen in other contexts (Chen et al., EMBOJ 2014). Thus, the difference between ATAD1^WT^ and ATAD1^E193Q^-expressing cell lines cannot be explained simply by lower expression of the mutant.

3. The authors should clarify whether loss of ATAD1 sensitizes H4 to MCL^-^1 inhibitors as Jurkat cells (Figures 1H and 1I). The BH3 profiling data shown in Figure 2H would argue against this. The authors should provide an explanation for the differential phenotypes in these two cell lines.

In studying additional cell lines, we find that MCL1 inhibition is not as effective or consistent as proteasome inhibition in exploiting *ATAD1* deficiency. We suspect that MCL1 inhibitors are effective in Jurkat cells because they are more primed for apoptosis generally, and have high steady-state abundance of BIM and NOXA relative to other cell types used (e.g. SW1088). Proteasome inhibitors can generate an imbalance in pro- and anti-apoptotic proteins, while BH3 mimetics can only exploit a pre-existing imbalance. For instance, if there is little BIM present at steady state, then there will be little BIM liberated upon antagonizing MCL1 with AMG176. In contrast, blocking proteasome function (or knockout of *MARCH5*) can increase BIM levels and challenge the cell in a distinct way. This type of differential sensitivity is routinely observed in the literature. Indeed, many solid tumors are insensitive to BH3 mimetics, which is thought to reflect lower basal apoptotic priming (e.g., Bhola et al. *Sci Signal* 2020; SánchezRivera et al. *PNAS* 2021; Cragg et al. *Nat Rev Cancer* 2009).

4. The authors should demonstrate that comparable levels of BIM were induced by doxycycline in WT and ATAD1-null cells (Figures 2E and 2F). Dose-dependent induction of BIM should be demonstrated in Figure 2G. To confirm the specificity of BIM, the authors should include inducible-tBID as a control.

As requested, we have performed anti-GFP immunoblots that demonstrate dosedependent induction of GFP-BIM_EL_ (**Figure 2F**). These experiments demonstrate that minimal basal expression of GFP-BIM is detectable in both WT and *ATAD1∆* cells in the absence of doxycycline. Increasing doxycycline concentration to 50 ng/mL or 500 ng/mL increases levels of GFP-BIM, but these increases are the same in WT and *ATAD1*∆ cells. Intermediate doses of GFP-BIM expression reveal that the loss of *ATAD1* significantly sensitizes cells to BIM.

Ectopic expression of tBID is an interesting idea for an additional experiment, but we have concluded that the interpretation of this experiment would be very complicated and would not explicitly test the model. tBID is a soluble BH3-only protein that can act similarly to BIM in that it binds anti-apoptotic proteins somewhat indiscriminately, making it a potent pro-apoptosis stimulus. Indeed, one would expect that tBID could exploit *ATAD1* deficiency indirectly, because of how ATAD1 affects the extent apoptotic priming. This concept is similar to how BIM BH3 peptide can exploit *ATAD1*-deficiency even though ATAD1 presumably cannot act on the BIM BH3 peptide itself, a soluble α helix. It is only in the context of the extensive additional data, including reciprocal co-IP, genetic data, and direct extraction in a reconstituted proteoliposome system, that ectopic expression of BIM strongly supports our model of ATAD1 extracting BIM and preventing MCL1 neutralization.

5. In figure 2I, the authors should include BIML and ATAD1E193Q for comparison.

We have now included data from a reciprocal Co-IP in which we immunoprecipitate endogenous BIM and blot for ATAD1-FLAG (Extended Data Figure 6E). We tried to include ATAD1^E193Q^, but found that it bound to the beads nonspecifically, leading to uninterpretable data. Although we did not directly assess binding between BIM_L_ and ATAD1, our in vitro extraction assay did use BIM_L_, so we know that BIM_L_ can be a direct substrate of ATAD1.

6. Because MitoTracker Red staining is mitochondrial membrane potential-dependent, the lack of MitoTracker Red staining in GFP-BIM puncta may simply reflect depolarized mitochondria. The authors should employ MitoTracker Green (mitochondrial potential-independent) and mCherry-fused BIM. If these GFP-BIM puncta are indeed not mitochondria, the authors should provide some explanation.

This is an excellent suggestion made by the reviewer. We now include imaging data wherein we compare GFP-BIM to mitochondria-targeted mCherry, and we observe the same phenomenon (Extended Data Figure 11). Specifically, we found that the presence of ATAD1 leads to accumulation of GFP-tagged BIM in the cytoplasm, and this phenotype is most striking under conditions of proteasome inhibition. What this result tells us is that ATAD1 promotes the nonmitochondrial localization of BIM, consistent with our other data that demonstrate ATAD1 can extract BIM from the OMM as a direct substrate. To reiterate, the observation that ATAD1 promoted non-mitochondrial localization of GFP-BIM strongly supports our model, and can be seen whether we label mitochondria with MitoTracker Red or genetically label them with a mitomCherry transgene.

We hypothesized that these non-mitochondrial GFP puncta might localize to the lysosome for degradation, but they do not colocalize with LysoTracker Blue (Extended Data Figure 12). We also wondered whether ATAD1 might extract BIM and relocalize it to peroxisomes, but again the GFP puncta did not colocalize with mRFP-SKL, which is imported into and labels peroxisomes (Extended Data Figure 10). Thus, the data suggest that these puncta represent aggregates of extracted GFP-BIM that accumulates upon inhibition of the proteasome.

7. The immunoblots in Figure 3C should include other BCL^-^2 family proteins especially MCL^-^1 and NOXA, which have been reported to play important roles in proteasome inhibitor-induced apoptosis.

We have now immunoblotted for other BCL2 family proteins in the context of proteasome inhibition and +/– ATAD1. In Jurkat cells, *ATAD1* loss increases basal NOXA levels (Figure 2C; Extended Data Figure 5A), but this does not occur in cultured PC3 cells, either basally or in response to bortezomib (Extended Data Figure 16C,D). There is also no difference in NOXA levels in PC3 xenograft tumors between wildtype and sgATAD1 tumors, either in the presence or absence of bortezomib (Figure 4C, Extended Data Figure 17C). Similarly, *ATAD1*-deficient Jurkat cells have increased MCL1 (Figure 2C, Extended Data Figure 5A), but *ATAD1*-deficient PC3 cells do not (Figure 4C, Extended Data Figure 16C,D, Extended Data Figure 17E). Since MCL1 can stabilize NOXA (Arai et al., *eLife* 2020), we suspect that these differences may reflect differential adaptations to *ATAD1* loss, where Jurkat cells upregulate MCL1, which stabilizes NOXA. It could also be that ATAD1 indirectly affects NOXA levels through some undescribed mechanism that is only operational in Jurkat cells. Given that NOXA doesn’t appear to have a transmembrane domain, it is very unlikely that NOXA is a direct substrate for ATAD1-mediated extraction.

8. The authors should include data of BIM KO or KD in Figure 4A.

We agree that this would be an interesting and potentially valuable experiment to show in yet another context that BIM is partially responsible for the pro-survival effects of ATAD1. After consultation with the editor, we jointly concluded that this was beyond the scope of the manuscript. More importantly, the original purpose of the SW1088 xenograft experiment was to test the hypothesis that ATAD1 deficiency sensitized tumors to proteasome inhibition– that the ATAD1-deficient SW1088 cells never engrafted precluded us from even testing that hypothesis. Therefore, we wanted to test our model in a system in which we could study established tumors lacking *ATAD1* and how they respond to treatment, since this most closely resembles the translational potential for our work. To that end, we conducted another xenograft experiment, this time using PC3 cells transduced with Cas9 + non-targeting sgRNA or sgRNA targeting *ATAD1*. In this context, loss of *ATAD1* was compatible with tumor engraftment (Figure 4B). We treated mice harboring *ATAD1*-proficient (sgNT) or deficient (sgATAD1) PC3 tumors with saline or bortezomib, and found that bortezomib slowed the growth of sgATAD1 tumors specifically (Figure 4B), while having no effect on the sgNT tumors (Figure 4A). These data represent a strong in vivo validation of our model in which ATAD1 protects cells from apoptosis induced by proteotoxic stress, which has direct implications for hundreds of thousands of cancer patients whose tumors have Chr10q23 deletion.

The authors should demonstrate that BIM protein levels are higher in ATAD1-null PTEN-null tumors than PTEN-null tumors shown in Figure 4C. The authors should also assess whether BIM protein levels affect the patient survival.

As described above, since submission of the initial manuscript, we have obtained data showing that ATAD1 can affect BIM via mechanisms beyond simply protein degradation. For instance, we demonstrate that ATAD1 promotes BIM_EL_ phosphorylation (Figure 2B, Extended Data Figure 4A,B, Extended Data Figure 13 A-D) and cytoplasmic localization (Extended Data Figure 9A-C, Extended Data Figure 10, Extended Data Figure 11A,B), both of which inactive its pro-apoptotic activity. In addition, we would expect that tumors that initiated and evolved in the *ATAD1*deficient state would adapt to that increased BIM abundance or activity over the course of their several years of growth and evolution in an *ATAD1*-deficient state. Second, although ATAD1 affects total BIM protein levels in many contexts, it appears that promoting BIM relocalization and phosphorylation are the universal features of how ATAD1 inactivates BIM. It would be interesting to assess these samples for phosphorylation and localization of BIM, but we see the strongest effects in the context of bortezomib treatment, which none of these patients experienced. To that end, we contacted the principal investigator of a different trial, in which prostate cancer patients were treated experimentally with the proteasome inhibitor carfilzomib. This was a phase II trial of 28 patients with metastatic castration resistant prostate cancer (NCT02047253). The lead investigator, Dr. Guru Sonpavde, was at University of Alabama Birmingham when the trial took place and has since moved to the Dana Farber Cancer Institute. He informed us that, unfortunately, they are unable to procure specimens for us to use for immunohistochemistry.

10. Time point information is missing in several figures.

We thank the reviewer for pointing out this important detail. We have now added time point information in every figure legend.

Referee #3 (Remarks to the Author):In this manuscript, by Winter et al., entitled ‘Co-deletion of ATAD1 with PTEN primes cells for BIM-mediated apoptosis’, the authors propose a novel cancer-specific vulnerability generated by the loss of ATAD1, which is adjacent to and frequently co-deleted with PTEN. ATAD1 functions to properly anchor proteins in the mitochondrial membrane. The authors provide clinical evidence that ATAD1 is lost in some portion of human cancers and, employing CRISPR based screening, identified ACOT11/MARCH5 loss as synthetic lethal hit in ATAD1 null tumor cells. The known role of MARCH5 in the regulation (degradation) of specific BCL2 family proteins prompted further testing to assess whether ATAD1 status alters sensitivity to apoptosis modulatory drugs such as the MCL1 inhibitor, AMG176. Mechanistically, they demonstrated that BIM is direct target of ATAD1 and accumulates in ATAD1 null cells. As BIM promotes apoptosis, they argued that ATAD1 normally functions to protect tumor cells from apoptotic stress. Consistent with this hypothesis, loss of ATAD1 correlates with better prognosis in metastatic prostate cancer patients, the main cancer type in which the PTEN/ATAD1 locus in most commonly deleted. This paper points to an intriguing synthetic lethal interaction generated by a collaterally deleted gene, thereby providing a responder hypothesis for the more precise application of available apoptosis inducer drugs in the clinic. This is a nice study that builds on the concepts of collateral and synthetic lethality with translational potential. However, there are several key issues that need to be addressed for consideration including the need for a more relevant cell line in the genomic screen (usual cell line used in the initial screen), the use of limited cell lines, need for analysis in key tumor types, limited in vivo studies, among other issues.

We are grateful for the reviewer’s time and attention to our work. We are encouraged that the reviewer thought this was a “nice study” with “intriguing” genetic interaction data, and recognized the translational potential. We respond here to the points made in the above paragraph.

General comment. Authors employ a confusing array of cell line systems across different experiments, raising questions as to generality of the findings (i.e., would MARCH 5 emerge in screens with other cell type?) and limiting the understanding of the potential role of ATAD1 in cancer biology (i.e., ATAD1 regulates many aspects of mitochondrial biology). Along these lines, a genome wide CRISPR screen were conducted only in Jurkat cells, which is an odd choice as opposed to other cancer cell types with greater relevance such as prostate cancer. As a first step, repeat screens are essential in another cell line model (isogenic ATAD1 WT vs KO, preferably prostate). In addition, ATAD1 or MARCH5 deletion increased both MCL1 (anti-apoptotic) and BIM (pro-apoptotic), and there was no attempt to better understand these opposing elements. Finally, a major deficiency is the limited therapeutic analysis of MARCH5 targeting or apoptotic inducers in ATAD1-null tumors versus controls. Overall, the study extends the universe of collateral lethal interactions but falls short with indepth data to support the conclusions.“Would MARCH5 emerge in screens with other cell type?”

We conducted an additional genome-wide CRISPR-based synthetic lethality screen using a gastric cancer cell line, HGC27, which harbors the Chr10q23 deletion seen in cancer patients. *ATAD1* is deleted in 4.1% of gastric cancer cases. We elaborate on the relevance of cell lines from different cancer types further below.

We transduced HGC27 cells with lentiviral *ATAD1* or empty vector and conducted CRISPR screens using the Brunello sgRNA library. We performed a sgRNA enrichment/depletion screen as we had done before in Jurkat cells. The top hit for synthetic lethality with *ATAD1* was again *MARCH5* (Figure 2F,G). It is remarkable that we identified the same genetic interaction despite all of the differences between our two screens. Notably, with Jurkat cells we screened WT cells vs. knockout clones, while with HGC27 cells we screened Del10q23 cells +/- *ATAD1* re-expression. By pairing these different approaches, we are able to control for: presence/absence of other genes that are co-deleted on Chr10q23; how cells might evolve differently if *ATAD1* loss is an early event in tumorigenesis rather than a perturbation in cell culture; and clonality of knockout cell lines. We highlight the other important differences between the two screens in Author response table 2.

Regarding cell line “relevance:”

In an ideal world, we would have used a prostate cancer cell line, but there are no prostate cancer cell lines available that harbor Del(10q23). We could have used PC3 cells with *ATAD1* deleted via CRISPR, but we felt that there was more to gain by doing the screen in a Del(10q23) cell line, which features other co-deleted genes and does not simply recapitulate our Jurkat screen in a different cell type.

If one considers all cancer types based on two variables, frequency of *ATAD1* deletion and patient mortality, then gastric cancer is the second most “relevant” cancer type, behind prostate cancer. Gastric cancer accounts for nearly 800,000 deaths worldwide each year and *ATAD1* is deleted in 4.1% of cases; Approximately 32,000 patients die every year from *ATAD1*deficient gastric cancer (GLOBOCAN 2020, Extended Data Figure 3). For perspective, this is more than half of the total number of patients who die of melanoma (i.e., *all* deaths attributable to melanoma, irrespective of *ATAD1* status).

As a result of these issues, we consider the screens done using Jurkat and HGC27 cell lines as nicely complementary systems to study *ATAD1* deletion in cancer more generally, and we think that pairing these two contrasting contexts (knockout vs. re-expressing in a Del10q23 cell) is a powerful approach.

Specific points:1) The screen should be conducted in more clinically relevant cancer cell line models to further validate the connection of ATAD1 to apoptosis regulation as well as to ascertain the generality or context-specific nature of their core findings concerning MARCH5. The use of additional cell lines may also uncover additional key actions of ATAD1 in other cancer hallmarks particularly metabolism. It is remarkable that only two hits (ACOTT11 and MARCH5 emerged), would this be the case with other cell line models?

As described above in detail, we conducted an additional complementary screen in HGC27 gastric cancer cells that harbor the Chr10q23 deletion seen in patients. Remarkably, the top hit was again *MARCH5*, clearly demonstrating the reproducibility of this genetic interaction even in distinct cellular situations.

“It would be helpful for the authors to explain why additional apoptosis gene hits (e.g., BCL family) were not observed in the screens”

This is an interesting question posed by the reviewer. The cell has different ways to suppress intrinsic apoptosis, including decreasing expression of genes encoding BH3-only proteins, post-translational modifications of BCL2 family members, altering mitochondrial morphology, and changes in protein stability/half-life. Evidently, ATAD1 is important for suppressing apoptosis in the context of a dysfunctional ubiquitin proteasome system – perhaps the reason for this specificity is that in order for ATAD1 to be relevant, there needs to be abundant BH3-only and other BCL2 proteins present on the OMM.

Additionally, we would point out that tumors, and especially cell lines, have evolved to circumvent apoptosis. Whether that evolution leads to a cell that is highly primed for cell death or one that is invulnerable to all apoptotic stimuli appears to depend on many variables. It may be that these cell types under these conditions have pro- or anti-apoptotic tendencies that are difficult to predict.

2) Along similar lines, the CRISPR screening results should be further validated by depleting MARCH5 and other hits in ATAD1 null models (ATAD1-δ Jarkat, H4, RPMI7951, prostate cancer cells). The mRNA and protein levels of MARCH5 need to be tested in different ATAD1 WT and deleted cancer cell lines. Any differences in MARCH5 level as compensation for ATAD1 deletion?

We thank the reviewer for this comment and agree that this part of the manuscript was under-developed in our initial submission. We have since made significant progress in this area. First, we directly validated the *ATAD1*/*MARCH5* synthetic lethal interaction using focused CRISPR deletion experiments (Figure 1G; Figure 2C-E). Second, we conducted an additional, complementary CRISPR screen using HGC27 cells (Del10q23, *ATAD1*-null gastric cancer cell line) re-expressing *ATAD1* or transduced with the empty vector. Remarkably, the top hit for synthetic lethality with *ATAD1* in this second screen was *MARCH5*. We discuss this screen in more detail above. Third, we demonstrate that BIM (*BCL2L11*) deletion or *MCL1* overexpression partially rescues this synthetic phenotype (Figure 2D,E; Extended Data Figure 5A-C). This genetic experiment demonstrates that BIM is partially responsible for *ATAD1*/*MARCH5* synthetic lethality. That the rescue is partial and not complete indicates that there must be other factors involved, which will be important to study in the future. We speculate that processes independent of apoptosis could also contribute to the functional relationship of

*ATAD1*/*MARCH5*. For instance, our lab and others have demonstrated that loss of *ATAD1* in human cells, *Atad1* in mouse embryonic fibroblasts, or *msp1∆* in yeast can lead to fragmented mitochondria (Chen et al., EMBOJ 2014). Increased mitochondrial fission has a well-established role in promoting apoptosis and could have other implications for cell fitness as manifested by changes in mitophagy, nutrient metabolism, and other pathways.

The reviewer raises an interesting idea that cells might upregulate *MARCH5* to compensate for *ATAD1* deletion. We queried the DepMap database for mRNA expression of *ATAD1* and *MARCHF5* (*MARCH5*). We see that these genes have a tendency toward coexpression, but in the ATAD1-null cell lines there are still *MARCH5* transcripts (see scatter plot in Author response image 3). It would be interesting if cells upregulated *MARCH5* in response to *ATAD1* deficiency, but perhaps that need not be the case.

**Author response image 3. sa2fig3:** MARCH5 vs ATAD1 expression across all DepMap cell lines.

We would favor the stance that if two pathways are truly redundant, then blocking one pathway will not require compensatory upregulation of the other pathway.

3) In generating the ATAD1 isogenic clones, only two clones were used. Given the heterogeneity of cancer cell lines, it is good practice to generate at least 3 independent clones.

This is a helpful comment from the reviewer. Rather than just use more clones of the cell line, we have chosen to ask the more rigorous question of whether the genetic interactions that we’ve observed are found in completely distinct cell lines, as described above. That the synthetic lethal interaction of *ATAD1* and *MARCH5* was observed in such a different context rules out the possibility that we are studying a genetic quirk of a clonal cell line. We have also validated genetic interactions and key functional experiments using multiple cell lines. To our knowledge, it is uncommon -if at all precedented- to conduct whole-genome CRISPR screens on two paired KO clones, representing true biological replicates. We have subsequently verified the key phenotypes across not only these two KO clones but also several other cells lines, both with CRISPR-KO techniques as well as re-expression of *ATAD1* in Del10q23 cell lines.

4) Figure 1C: should be substituted for extended data figure 1A. However, the images shown for patient 1 stained for PTEN and ATAD1 are not the same field of view and should be serial sections to better demonstrate PTEN and ATAD1 status in patient samples. Image quality can also be improved and a high mag inset should be added.

As requested by the reviewer, we are providing higher resolution images from exactly the same area for both stains, for both patient 1 and patient 2 (Extended Data Figure 1C). This study was conducted by a fellowship-trained genitourinary oncology pathologist.

5) Figure 1D to Figure 1E: Figure 1D highlights that PTEN/ATAD1 loss is observed across many cancer types, including prostate, lung adenocarcinoma, GBM etc. Given these observations, coupled with the highly context specific functions of genes in different cancers, it is perplexing as why was the CRISPR-KO screen performed with Jurkat cells derived from T-cell leukemia.

We have taken the reviewer’s advice and repeated a genome-wide CRISPR screen using a completely different cell type, from a tumor type that has the second highest number of deaths attributable to *ATAD1*-deficient tumors. Thus, we feel that this second, complementary CRISPR screen is highly disease-relevant. Again, an additional strength of this second screen that we conducted with HGC27 cells is that this cell line naturally has the Chr10q23 deletion found in human tumors. So, not only do our two screens come from two different tumor types, but they represent fundamentally opposite and complementary approaches to studying the *ATAD1* proficient and deficient states (WT/KO vs. Del(10q23)/+ATAD1).

6) Figure 1G: Do the cell-lines in the lower left corner harbor PTEN and ATAD1-loss? If not, can we still conclude that MARCH5 or MCL1 is essential in ATAD1-null cancer cells in general?

This is a reasonable question, but it points toward a limitation of DepMap. When looking for genetic interactions, a more sensitive approach is to use an isogenic background +/- the gene or variant of interest, as we have done with two complementary screens. The analysis of synthetic lethality using a resource like DepMap can be confounded by the broad diversity of cell lines included in the “denominator.” In other words, the signal cannot overcome the noise.

The gold standard example of synthetic lethality is the interaction between BRCA1/2 mutation and PARP inhibition (reviewed by Lord and Ashworth, Science 2017). Briefly, BRCA1/2 mutant cells are “1000 times more sensitive to PARPi than *BRCA*-wild type cells (depending on the PARPi used and the experimental format).” For instance, a recent paper by Fugger et al. (Science 2021) demonstrates with a colony formation assay that *BRCA2* loss clearly sensitizes DLD1 cells to Olaparib treatment, compared to WT cells:

Moreover, in human ovarian cancer patients with BRCA mutations, PARPi are remarkably effective (Moore et al., *NEJM* 2018):

Similar results have been found with BRCA-mutant pancreatic cancer (Tutt et al., *NEJM* 2021), and BRCA-mutant prostate cancer (de Bono et al., *NEJM* 2020). This is the best example of synthetic lethality in cancer cells and it has already made a meaningful impact on patient care.

Despite the validation of this mechanism with basic research and the clear therapeutic benefit seen in the clinic, DepMap would have completely missed this chemo-genetic interaction. In Author response image 4 we have plotted Olaparib (a clinically-used PARP inhibitor) sensitivity on the x-axis, with increased sensitivity on the left and decreased sensitivity on the right. Each data point represents one cell line. All cell lines with a *BRCA1* mutation (left plot) or *BRCA2* mutation (right plot) are highlighted with a star.

**Author response image 4. sa2fig4:** 

To ask the same question posed by the reviewer, above: If the cell lines on the left side of the graph are not BRCA mutant, then can we still conclude that PARP inhibitors are selectively toxic in BRCA-mutant cancer cells in general? Clearly, the starred cell lines do not cluster on the left side of the distribution in either plot in Author response image 4. The result looks the same if we plot PARP1 gene dependence (CRISPR score, CHRONOS) on the x-axis instead of PARP inhibitor sensitivity.

We present these data simply to point out that DepMap is powerful but imperfect. If it misses the gold standard synthetic lethal interaction between BRCA and PARP, then we should not be concerned that the interaction between ATAD1 and MARCH5 does not emerge either. The DepMap is an unparalleled resource for identifying co-essential genes and generating hypotheses about how a given gene might relate to cellular processes. At least in its current iteration, DepMap is not as well suited to identify synthetic lethal interactions.

7) Figures 1H, 1I, Figure 2A-2G, only Jurkat cells were used for all the experiments. These findings need to be validated in different cancer cell lines with WT and/or null ATAD1. Figure 2H and 2I were done with H4 cells. Again, this needs to be repeated in other cancer cell lines.

We have now added key experiments using multiple cell lines, including genome-wide CRISPR screens (Figure 1D-G), proteasome inhibitor treatment (Figure 3A-H, Extended Data Figure 14A-F, Extended Data Figure 15A,B), epistasis experiments of proteasome inhibition and caspase inhibition in *ATAD1*-null cells (Figure 3H, Extended Data Figure 15C-F), and tumor xenografts (Figure 4A-C; Extended Data Figure 18). We also validate our mechanistic work using purified proteins in a reconstituted system that transcends the variability inherent to different cell lines (Figure 2H-K; Extended Data Figure 7).

We would also like to emphasize our new data, where two, independent, complementary, and unbiased genetic screens uncovered the same synthetic lethal interaction between *ATAD1* and *MARCH5*. This robust result forms an important part of the foundation for our project.

8) Given the differential impact of AMG176 and nivitoclax, what is the basis for this mechanistic difference given that march5 regulates MCL1, Noxa and Bim levels? Please clarify.

We think it is likely that ATAD1 and MARCH5/MCL1 mediate parallel pathways that suppress apoptosis. It is well described that inhibiting BCL2 or BCLXL can be synergistically toxic in combination with inhibiting MCL1 (Bhatt et al. Cancer Cell 2020) Insofar as *MARCH5* deletion approximates MCL1 inhibition, we speculate that increased BIM activity in *ATAD1*-null cells may neutralize other anti-apoptotic factors like BCLXL, in addition to MCL1.

9) Figure 2A: The differences in BIM expression in both the wildtype and ATAD1-null cells are rather meager by western blot. The authors should check the expression of BIM in both mitochondrial and cytoplasmic fractions. Also, ATAD1 deletion increases both MCL1 and BIM, in a way similar to MARCH5 deletion as described in the text. While the author studied the mechanism of how ATAD1 extracts BIM from mitochondrial, there was no attempt to understand findings concerning MCL1 which would seem important.

As mentioned by the reviewer, in Jurkat cells, ATAD1 affects BIM and MCL1 levels (Figure 2A-C; Extended Data Figure 5A). However, in PC3 cells, ATAD1 affected BIM but not MCL1 levels (Figure 4C; Extended Data Figure 16C; Extended Data Figure 17). Thus, we think the effect of ATAD1 on MCL1 might be indirect, through BIM-mediated stabilization of MCL1. It could also be that cells upregulate MCL1 to compensate for *ATAD1* deletion in some contexts (e.g., Jurkat cells) but not in others (e.g. PC3 cells).

Nonetheless, as pointed out by the reviewer, it is important to address the functional relationship of ATAD1 and MCL1. To address this, we did genetic experiments, which strongly argue against the notion that ATAD1 extracts or otherwise antagonizes MCL1. Specifically, overexpression of MCL1 partially rescues the synthetic lethality of *ATAD1*/*MARCH5* in Jurkat cells, and exogenous MCL1 does not differentially accumulate in the presence vs. absence of *ATAD1* (Figure 2E, Extended Data Figure 5A,B). If MCL1 were an ATAD1 substrate, we might even expect to see that MCL1 overexpression is *toxic* in *ATAD1*-deficient cell lines, as is seen when Msp1 substrates are overexpressed in *msp1*∆ yeast (e.g., Pex15; Chen et al., EMBOJ 2014). We would expect to observe a functionally antagonistic relationship between ATAD1 and a substrate, but MCL1 is instead functionally complementary with ATAD1. In sum, our *MCL1* overexpression studies strongly support a model in which ATAD1 and MARCH5 act upstream of MCL1, and on BIM and NOXA, respectively.

Furthermore, as requested by the reviewer, we conducted fractionation experiments using differential centrifugation but could not detect BIM in the cytoplasmic fraction (Author response image 5). As expected, we observed increased BIM in *ATAD1*∆ cell lines, which is particularly noticeable when one compares BIM to VDAC, a specific loading control for the mitochondrial/organellar fraction (crude mitochondria, “CM”). The effect on BIM is specific, since BAK levels do not increase in *ATAD1*∆ CM fractions. In contrast to these data, our imaging experiments and several published manuscripts describe a non-mitochondrial population of BIM. Possible explanations of this discrepancy include: (i) BIM binds to microtubules that cofractionate with mitochondria (Luo et al. Mol Cell 2012) (ii) ATAD1 deposits BIM onto other nearby membranes that co-fractionate with mitochondria (Matsumoto et al., Mol Cell 2019), (iii) extracted BIM is rapidly degraded by the proteasome, or most likely (iv) a combination of these phenomena is occurring and preventing us from observing non-mitochondrial BIM is this experiment.

**Author response image 5. sa2fig5:** 

10) Figure2F should include the expression levels of exogenous BIM. It would be important to show that the inducible expression system generates similar levels of BIM expression in different cell lines.

As requested, we have performed anti-GFP immunoblots that demonstrate dose dependent induction of GFP-BIM_EL_ (Figure 2F). As shown in Figure 2F, minimal basal expression of GFP-BIM is detectable in both WT and *ATAD1∆* cells in the absence of doxycycline. Increasing doxycycline concentration to 50 ng/mL or 500 ng/mL increases levels of GFP-BIM, but these increases are the same in WT and *ATAD1*∆ cells. Intermediate doses of GFP-BIM expression reveal that the loss of *ATAD1* significantly sensitizes cells to BIM, as measured by PARP cleavage (Figure 2F) and viability assays (Figure 2G). Thus, these results demonstrate that similar levels of BIM expression are induced in WT and *ATAD1∆* cells and that the presence or absence of *ATAD1* dictates the cellular response to those equivalent BIM levels.

11) Figure 2H: Is more cytochrome c release of ATAD1-null cells with EV or ATAD1E193Q than WT cells due to greater accumulated of endogenous BIM (without active ATAD1), but not only the presence of nanomolar BIM peptide (the description in the manuscript is confusing)? For several other BH3 peptides (A133, venetoclax, HRKy, MS1, BAD), cytochrome c release is actually higher with mutant ATAD1E193Q than WT. Why is that?

We appreciate that interpretation of the BH3 profiling assay can be confusing when viewed in the context of our other experiments. We have now included a much more explicit and clear description and interpretation. To summarize, we do not think that ATAD1 acts directly on the BH3 peptide derived from BIM. Rather, the data suggest that ATAD1 acts on endogenous BIM, which changes the apoptotic priming “set-point” underlying the response to challenge with exogenous BIM BH3 peptide. There are pools of pro and anti-apoptotic proteins on the OMM, and the relative abundances and activation states of these proteins dictates how a cell will respond to further pro-apoptotic stimuli, such as treatment with a BH3 peptide. Increased activity (de-phosphorylation, localization, and abundance) of BIM due to the loss of *ATAD1* will “fill up” the reservoirs of anti-apoptotic proteins and make the cell more sensitive to exogenous challenge with BIM peptide. This experiment simply demonstrates that the absence of *ATAD1* primes cells for apoptosis, using an assay that is widely used in the field. Although these data support the overarching hypothesis, in the revised manuscript we have deemphasized this assay to help focus the attention of the reader on more easily interpretable data that also strongly support the model.

As the reviewer points out, the E193Q mutant has slightly higher cytochrome C release than the EV (*ATAD1*-null) cells for several peptides. This could either be noise (most likely) or could derive from the E193Q mutant acting as a dominant negative (Chen et al., EMBOJ 2014) to stabilize BIM and other ATAD1 substrates on the OMM. In any event, the key comparisons that are relevant to our model are between ATAD1^WT^ cells and EV (*ATAD1*-null) cells and between ATAD1^WT^ and ATAD1^E193Q^ cells. These comparisons clearly demonstrate the same result, that cells with functional ATAD1 have suppressed apoptotic priming, particularly in response to BIM peptide.

12) Figure 3A, the difference is not clear and needs more objective quantification. The authors should consider separate mitochondria and cytosolic fractions to determine the amount of BIM in each fraction.

We have now included many more examples of this phenotype, which we hope the reviewer will agree is now more clearly demonstrated (Extended Data Figure 10,12). Specifically, we discovered that the presence of ATAD1 leads to accumulation of GFP-tagged BIM in the cytoplasm, and this phenotype is most striking under conditions of proteasome inhibition. What this result tells us is that ATAD1 promotes the non-mitochondrial localization of BIM, consistent with our other data that demonstrate ATAD1 can extract BIM as a direct substrate. To reiterate, the observation that ATAD1 promoted non-mitochondrial localization of GFP-BIM strongly supports our model, and can be seen whether we label mitochondria with mitotracker red or genetically label them with a mito-mCherry transgene. We hypothesized that these nonmitochondrial GFP puncta might localize to the lysosome for degradation, but they do not colocalize with Lysotracker Blue (Extended Data Figure 12). We also wondered whether ATAD1 might extract BIM and relocalize it to peroxisomes, but again the GFP puncta did not colocalize with mRFP-SKL, which is imported into and labels peroxisomes (Extended Data Figure 11). Thus, our current hypothesis is that these puncta represent aggregates of extracted GFP-BIM that has accumulated upon inhibition of the proteasome. Future investigation into the fate of extracted ATAD1 substrates will be interesting, but is beyond the scope of the current manuscript.

13) Figure 3C, BIM protein and ubiquitylation levels should be compared between EV and ATAD1 in the absence or presence of BTZ, which would help to understand the mechanism. After BIM is extracted from mitochondria to puncta by ATAD1, there is no evidence that BIM is degraded more efficiently and rapidly. The model conveys that ATAD1 extracts BIM for degradation, but there is no evidence to support this point. Is extraction of BIM from mitochondria to puncta (inactivation) enough for its anti-apoptotic effect or is further degradation is required?

We suggest that our data strongly support the hypothesis that degradation is not required for ATAD1 to neutralize BIM, since ATAD1 is most protective to cells in the *presence of a proteasome inhibitor*. If the sole fate of ATAD1-extracted substrates were proteasomal degradation, then ATAD1 would be inconsequential in the context of an inhibited proteasome. Instead, we think that extracted and phosphorylated BIM represents a detoxified intermediate between the OMM and the proteasome.

BIM protein levels increase upon loss of *ATAD1* in many cell contexts (for example, Figure 2A, 2B, Extended Data Figure 16C,D). Nevertheless, we have made the reproducible observation that ATAD1 regulates BIM via mechanisms beyond simply degradation of the protein. Specifically, we have added new experiments using high-resolution imaging, which demonstrate that ATAD1 promotes the non-mitochondrial localization of BIM (Extended Data Figure 9-12). BIM is known to be inactivated by changes in localization (Kale, Osterlund, and Andrews, *Cell Death & Differentiation* 2018), but how it translocates from the OMM to the cytosol was previously unknown. BIM requires OMM localization to bind and inhibit anti-apoptotic proteins, such as MCL1, and to activate the pore-forming proteins BAX/BAK. Hence, BIM localization is a key variable that dictates its pro-apoptotic activity.

Finally, we have made the exciting new discovery that ATAD1 promotes the phosphorylation of BIM_EL_ by cytosolic kinases (Figure 2B; Extended Data Figure 4A,B; Extended Data Figure 13A-F). Phosphorylation of BIM_EL_ has been clearly established in several papers as an important mechanism by which kinases like ERK promote cell survival. Phosphorylation of BIM_EL_ reduces its toxicity by decreasing its binding affinity to pro-survival proteins like MCL1, preventing its activation of BAX, and accelerating its degradation (e.g., PMID: 17151701; PMID: 29775995; PMID: 18498746). We now show that *ATAD1* loss decreases BIM_EL_ phosphorylation, while re-expressing *ATAD1* increases BIM_EL_ phosphorylation, and identify residues Ser69 and Ser77 as being phosphorylated in this context (Figure 2B; Extended Data Figure 4A,B; Extended Data Figure 13A,D-F). Of course, ATAD1 is not a kinase, so our model is that ATAD1 extracts BIM and, by so doing, puts it into a biochemical environment that promotes its phosphorylation by kinases such as ERK and Aurora A/B. In the context of proteasome inhibition, *ATAD1*-null cells accumulate unphosphorylated (active) BIM, while ATAD1-positive cells accumulate phosphorylated (inactive) BIM (Extended Data Figure 13A,D-F). We propose that ATAD1-mediated extraction of BIM from the OMM immediately prevents it from activating apoptosis, and subsequent phosphorylation of BIM triggers either its degradation and/or inactivation via other mechanisms (enumerated above).

Figure 4A and B, authors should use additional cell lines (H4, RPMI7951) and validate the conclusion with ATAD1-KO models. Also, the use of the glioma cell model appears to be conducted in the subQ setting which is inadequate and nonphysiological. Although beyond the scope of this study, future studies should attempt to further validate the hypothesis by characterizing the therapeutic vulnerability of tumors arising in GEMS (PTEN/ATAD1 null versus PTEN null PCa model).

We conducted additional xenograft experiments using *ATAD1*-knockout PC3 cells and found two main results. First, *ATAD1* was not essential for tumor growth basally in this cell type, which reflects the fact that *ATAD1*-deficient tumors can form in humans. Second, loss of *ATAD1* conferred sensitivity to bortezomib treatment in vivo. Thus, the role of *ATAD1* in tumor growth under basal conditions is not consistent in all cancer cell lines: in SW1088 cells it is required, while that is not the case in PC3 cells. We include a brief discussion of this point in the revised manuscript. Given that the PC3 experiment is more reflective of the human tumor situation, particularly in prostate cancer, and is more relevant to the overall premise of the manuscript, we have moved the SW1088 xenograft data to a supplemental figure.

We would love to conduct a more physiologically relevant model such as a PrAd GEMM, but as the reviewer points out that is beyond the scope of the current study. We hope the reviewer agrees that subcutaneous xenografts of a mCRPC cell line (Figure 4A,B) in conjunction with a striking overall survival phenotype in human patients (Figure 4D-F) build nicely on the bulk of our data, which are more biochemical and genetic in nature.

14) Figure 4, authors should show therapeutic impact of CRISPR hits or apoptosis inducers to ATAD1 deleted tumors rather than the function of ATAD1 as tumor suppressor. Will ATAD1 inhibitor treatment of cancer cells mimic the effect of ATAD1 deletion in promoting apoptosis thereby inhibiting tumor growth for longer survival?

As described above (#14), we conducted another xenograft experiment, this time using PC3 cells transduced with Cas9 + non-targeting sgRNA or sgRNA targeting *ATAD1*. In this context, loss of *ATAD1* was compatible with tumor engraftment (Figure 4B). We treated mice harboring *ATAD1*-proficient (sgNT) or deficient (sgATAD1) PC3 tumors with saline or bortezomib, and found that bortezomib slowed the growth of sgATAD1 tumors specifically (Figure 4B), while having no effect on the sgNT tumors (Figure 4A). These data represent a clear in vivo validation of our model in which ATAD1 protects cells from apoptosis induced by proteotoxic stress, which has direct implications for hundreds of thousands of cancer patients whose tumors have Chr10q23 deletion. Although we see that proteasome inhibition is preferentially toxic to cells lacking *ATAD1* both in culture and in murine xenografts, the clinical utility of proteasome inhibitors in cancer patients is somewhat limited by on-target toxicity. We think that a MARCH5 inhibitor would be an exciting area for drug development, since we would predict that such an agent would achieve a maximal therapeutic window by selectively targeting *ATAD1*-null tumor cells while sparing the rest of the cells in the patient’s body, which have *ATAD1* intact. We hope that our work will stimulate further investigation in this space.

Referee #4 (Remarks to the Author):The manuscript by Winter et al. describes the identification of ATAD1 as a driver of a novel collateral lethal vulnerability in PTEN deficient tumours. The authors present evidence that the loss of ATAD1, which co-occurs at very high frequency in PTEN-deleted tumours, sensitizes cells to proteasome inhibitors through accumulation of BIM. Their data show that ATAD1 acts as a direct extractase for mitochondrial BIM, which prevents BIM from causing cytochrome c release and apoptosis. The authors also conclude that the loss of ATAD1 primes tumour cells for apoptosis which explains why prostate cancer patients with PTEN/ATAD1 co-deleted tumours have longer overall survival compared to those with ATAD1-proficient ones.While the concept of collateral lethality is not new, the identification of a new collateral lethal vulnerability associated with PTEN loss is important given the frequency of PTEN inactivation in human cancer. The manuscript provides important mechanistic insight into how ATAD1 regulates BIM physiology, and identifies proteasome inhibitor treatment as a potential therapeutic strategy for patients with PTEN/ATAD1-deficient tumours.Despite these insights, the manuscript suffers from attempting to unnecessarily explore too many aspects of the ATAD1/PTEN functional relationship. The result is a seeming loss of focus in some areas, and a lack of depth (both conceptual and experimental) in others. Below is a list of the most important omissions.

We greatly appreciate that the reviewer considers our work to offer “important mechanistic insights,” which we agree are the strength of our work. We have carefully revised our manuscript in response to the reviewer’s critiques, including that we “unnecessarily explore too many aspects of the *ATAD1*/*PTEN* functional relationship.” To that end, we have focused on the demonstration that deficiency of *ATAD1* sensitizes cells to apoptosis triggered by proteasome dysfunction, and how direct extraction of BIM is a mechanistic driver of that phenotype.

We would like to comment on this critique in particular, in order to provide clarity: we do not attempt to address the “functional relationship” of *ATAD1* and *PTEN*. We have no doubt that there would be interesting biology to study in this area, but it is outside the scope of our current manuscript. Moreover, how the biology we discovered varies based on the presence or absence of *PTEN* is a fascinating question, but our data suggest that the functions of ATAD1 are similar in either setting. The clinical/translational potential of our work is more obvious in the context of *PTEN*-null tumors, so we think it is reasonable to focus on that context for our mechanistic and cell culture experiments. We hope the reviewer agrees with this decision to focus on how ATAD1 suppresses apoptosis induced by proteotoxic stress.

1) Despite the high incidence of PTEN/ATAD1 co-deletion in mCRPC (40% frequency of 10q23 loss), clinical trials of proteasome inhibitors have shown little to no activity in these patients (DOI:10.1200/JCO.2018.36.6_suppl.224; DOI: 0.1200/jco.2005.23.16_suppl.4633). Similarly a large phase III study of the brain-penetrant proteasome inhibitor marizonib in glioblastoma (which has 10% frequency of 10q23 deletion) also failed to show any improvement in overall survival. Based on the authors’ results, one might expect to see some signal in these trials. This does not seem to be the case.

We thank the reviewer for their clinical and translational expertise, and agree that this is an important area of literature to discuss, which we do in the revised manuscript. First, to clarify: the reviewer states that *ATAD1*/*PTEN* co-deletion occurs in “40%” of mCRPC and “10%” of glioblastoma, while the actual values are ~25% and ~7%, respectively, as shown in our first figure (Figure 1B,C). The differences in these percentages are important, given the reviewer’s suggestion that prior trials of proteasome inhibitors should have detected an effect of *ATAD1* loss in a subset of patients. We discuss specific studies below, but in the end, it is evident that prior trials of proteasome inhibitors on unselected patients had inadequate sample sizes to reliably see efficacy on this fraction of patients with statistical confidence.

Here we highlight three trials of proteasome inhibitors in patients with advanced prostate cancer. First let us consider the phase II trial by Naik et al. (JCO, 2018) of carfilzomib in mCRPC patients (n = 28 patients), which was cited by the reviewer. The rationale for doing this study was that “bortezomib, a proteasome inhibitor, has exhibited activity in a subset of [mCRPC] patients,” but that carfilzomib, a second-generation proteasome inhibitor might have improved efficacy. Depending on the genomic study, *ATAD1* is deleted in 13-27% of patients with mCRPC, so one might expect that approximately 4-7 patients in this 28-patient cohort to have tumors that lacked *ATAD1*. The study found that “minor PSA declines and CTC declines were observed in 3 patients each, and one patient exhibited both PSA and CTC declines.” The authors conclude: “Minor and transient activity may be present in a small subset of patients.” It may have been that the 7 patients (25% of cohort) who had some degree of response all had *ATAD1*-null tumors. We wanted to test that hypothesis, so we contacted Dr. Guru Sonpavde, the lead investigator on the trial. Unfortunately, he informed us that specimens from that study were unavailable for us to analyze by immunohistochemistry due to his recent move from UAB to Boston.

Morris et al. (Journal of Urology, 2007) conducted a trial of bortezomib and prednisone in patients with mCRPC, but only 13 or 9 patients were “evaluable for response” in the two different dosing regimen cohorts. One patient out of the 9-patient cohort reached the primary endpoint.

Papandreou et al. (JCO 2004) conducted a phase I/II study of dose-escalation of bortezomib (alone) in 53 patients with advanced prostate cancer (AIPCa). The authors found that “biological activity and antitumor activity is seen in AIPCa at tolerated doses of bortezomib,” and concluded: “This agent should be further explored with chemotherapy agents in advanced prostate cancer.”

These three studies of proteasome inhibitors in patients with advanced prostate cancer (a rather aggressive disease) demonstrate *some* degree of signal, and this signal could be suspected to be enriched in patients with *ATAD1*-null tumors. These studies support the idea that it is worthwhile to investigate targeting *ATAD1*-deficient tumors with proteasome inhibitors. In order to truly test our model, one would need to conduct a large prospective study that pre-selected patients based on biopsy-proven *ATAD1* deficiency in their tumors.

In the MIRAGE trial of marizomib in patients with glioblastoma multiforme, 749 patients were randomized in a 1:1 ratio to receive standard of care (temozolomide and radiotherapy) with or without marizomib. If 375 patients were in the +MRZ group, one could expect approximately 26 patients (7%) to have tumors that lack *ATAD1*. What kind of change in overall survival (OS) would be reasonable to expect in these cases if our model is correct? For perspective, consider that temozolomide added to radiotherapy increased OS from 12.1 to 14.6 months in the landmark trial by Stupp et al., published in *The New England Journal of Medicine* in 2005 (DOI: 10.1056/NEJMoa043330). Suppose that MRZ treatment in patients with ATAD1null tumors conferred an equivalent increase of 2.5 months in OS – surely a rigorous benchmark, given that temozolomide represents the greatest effect size on OS ever seen in a trial of adjuvant chemotherapy in patients with GBM. The null hypothesis in this case is that marizomib plus TMZ/RT does not affect OS compared to RT alone. We think it is extremely unlikely that this null hypothesis would be refuted if the only difference between these groups was that 7% of patients (those with ATAD1-deficient tumors) lived 2.5 months longer. Of course, an effect specific to *ATAD1*-deficient tumors very well could have occurred, but the analysis was not designed to detect an effect of that nature. It may be possible to retrospectively assess ATAD1 levels in the tumors of “responders” vs. “nonresponders” of this trial, which could reveal differences hidden in the initial analysis on all patients.

The lead investigator of the MIRAGE trial is a physician scientist named Dr. Patrick Roth. Thinking along the same lines as the reviewer, we reached out to Dr. Roth in August, 2021. He kindly agreed to have a virtual meeting with us where we discussed our project and our interest in collaborating with him to use the MIRAGE trial data to perform a retrospective analysis of survival in patients whose tumors lacked *ATAD1*. He was “very interested in collaborating” with us, but in 6 months the European Organization for the Research and Treatment of Cancer (EORTC) has made zero progress in this direction. Dr. Roth wrote to us in an email in November: “…The problem is that the tissue samples are stored in a central biobank. Currently, EORTC is exploring if an additional IRB approval is required to release them. I would be more than happy to provide these samples but I cannot predicit [sic] when we will be able to do so.”

We hope that in the future we will be able to test our model retrospectively using human data, but that is likely years away. We hope at least that the publication of our work will motivate further studies by investigators with access to these samples.

2. Their analysis of mCRPC patient data shows that PTEN/ATAD1-co-deleted cases have longer overall survival compared to PTEN-null and unaltered cases. In fact, their data show no significant difference between PTEN-null and unaltered cases. However there is evidence that PTEN loss is associated with poorer prognosis in mCRPC (PMID: 25454616; DOI: 10.1200/JCO.2021.39.6_suppl.58). Even if PTEN/ATAD1-deficient and PTEN-deficient cases in their dataset were assessed together, their analysis would show a trend for improved OS (i.e. for all PTEN-deficient tumours irrespective of ATAD1 status).

This is an insightful point, and the reviewer is correct that across prostate cancer generally, *PTEN* loss confers a negative prognosis. Of course, the answer to the question of how *PTEN* loss affects prognosis depends on the patient population in which the analysis is performed; in other words, what is the denominator? Here we use mCRPC patients with quite advanced disease. It may be that in this population, all tumors are aggressive, such that *PTEN* loss itself is insufficient to distinguish tumors based on prognosis. We believe that we have been appropriately conservative in our analysis of these patient data, but would be happy to implement any suggestions from the reviewer on the methods or result.

Of interest is a recent preprint by the Speed group at WEHI that develops a new algorithm to normalize RNA-Seq datasets. The authors used colorectal cancer patient data for their study and found that high *ATAD1* expression in tumors predicted shorter median overall survival compared to low *ATAD1* expression, corroborating our findings in a different tumor type. Link to paper: https://www.biorxiv.org/content/10.1101/2021.11.01.466731v1.full.pdf After seeing this preprint we examined publicly available colorectal cancer (CRC) survival data to test their hypothesis. We examined overall survival in all colorectal cancer datasets available on cBioPortal (https://bit.ly/3IOUARH). Specifically, we examined whether *ATAD1* deletion (“ATAD1” group in Author response image 6 and Author response image 7) predicted overall survival, and found, to our surprise, that these data agree with those of Speed and colleagues. *ATAD1* deletion in tumors correlates strongly with increased overall survival in CRC patients, similar to what we observed in prostate cancer.

**Author response image 6. sa2fig6:** 

**Author response image 7. sa2fig7:** LEFT: PARP inhibitor sensitivity (Olaparib, AUC) in DepMap cell lines based on BRCA1/2 status. “In group” consists of all DepMap cell lines harboring BRCA1 and/or BRCA2 mutation. The “out group” is all remaining cell lines. The parameter shown is sensitivity to Olaparib, as measured by area under the curve (AUC) from the CTD^2^ dataset. Lower values (left direction) on the x-axis indicate lower AUC and increased sensitivity. Based on extensive literature, we would expect BRCA mutant cell lines to be more sensitive than the “out group.” That is not the case (Q = 0.867; Author response table 3).

3. There is no explanation for the choice of Jurkat cells in the original CRISPR screen. The relevance of PTEN/ATAD1 co-deletion in T-cell leukemia is unknown. Although PTEN is frequently inactivated in TALL, the most common mechanisms involve point mutations, hemyzygous deletions, or intragenic microdeletions. Also, parental Jurkat cells are already sensitive to proteasome inhibitors (PMID: 24301524). It seems using a prostate cell line such as LNCaP (which has a deep PTEN-ATAD1 co-deletion according to the Cance Cell Line Encyclopedia) where ATAD1 is reconstituted would perhaps have been a more sensible choice given that prostate cancer has the highest incidence of 10q23 loss.

We agree with this reviewer that Jurkat cells are not the most disease relevant cell type to use for our initial screen. The reason we used Jurkat cells is that when we conducted the screen, in 2018, it was very uncommon to use adherent cell lines for genome wide CRISPR screening. We chose Jurkat cells as the best available option at the time, since they are PTENnull, and because *ATAD1* is deleted in T-ALL, albeit infrequently.

To address this shortcoming, we conducted an additional genome-wide CRISPR-based synthetic lethality screen using a gastric cancer cell line, HGC27, which harbors the Chr10q23 deletion seen in cancer patients. *ATAD1* is deleted in 4.1% of gastric cancer cases. We elaborate on the relevance of cell lines from different cancer types further below.

We transduced HGC27 cells with lentiviral *ATAD1* or empty vector and conducted CRISPR screens using the Brunello sgRNA library. We performed a sgRNA enrichment/depletion screen as we had done before in Jurkat cells. The top hit for synthetic lethality with *ATAD1* was again *MARCH5* (Figure 2F,G). It is remarkable that we identified the same genetic interaction despite all of the differences between our two screens. Notably, with Jurkat cells we screened WT cells vs. knockout clones, while with HGC27 cells we screened Del10q23 cells +/- ATAD1 re-expression. By pairing these different approaches, we are able to control for: presence/absence of other genes that are co-deleted on Chr10q23; how cells might evolve differently if *ATAD1* loss is an early event in tumorigenesis rather than a perturbation in cell culture; and clonality of knockout cell lines. We highlight the other important differences between the two screens in Author response table 2.

Regarding cell line “relevance:”

In an ideal world, we would have used a prostate cancer cell line, but there are no prostate cancer cell lines available that harbor Del(10q23). We could have used PC3 cells with *ATAD1* deleted via CRISPR, but we felt that there was more to gain by doing the screen in a Del(10q23) cell line, which features other co-deleted genes and does not simply recapitulate our Jurkat screen in a different cell type.

If one considers all cancer types based on two variables, frequency of *ATAD1* deletion and patient mortality, then gastric cancer is the second most “relevant” cancer type, behind prostate cancer. Gastric cancer accounts for nearly 800,000 deaths worldwide each year and *ATAD1* is deleted in 4.1% of cases; Approximately 32,000 patients die every year from *ATAD1*deficient gastric cancer (GLOBOCAN 2020, Extended Data Figure 3). For perspective, this is more than half of the total number of patients who die of melanoma (i.e., *all* deaths attributable to melanoma, irrespective of *ATAD1* status).

As a result of these issues, we consider the screens done using Jurkat and HGC27 cell lines as nicely complementary systems to study *ATAD1* deletion in cancer more generally, and we think that pairing these two contrasting contexts (knockout vs. re-expressing in a Del10q23 cell) is a powerful approach.

4. The synthetic lethal phenotype of MARCH5 and ACOT11 with ATAD1 inferred from the CRISPR screen was never confirmed with individual gRNAs (or si/shRNAs) and viability/apoptosis assays. In fact, there is no follow up work on either of these two genes (which are the actual hits from the CRISPT screen) beyond the screen itself. Instead, the authors moved to explore MCL1 inhibitors, proteasome inhibitors, and BIM overexpression.

We thank the reviewer for this comment and agree that this part of the manuscript was under-developed in our initial submission. We have since made significant progress in this area. First, we directly validated the *ATAD1*/*MARCH5* synthetic lethal interaction using focused CRISPR deletion experiments (Figure 1G; Figure 2C-E). Second, we conducted an additional, complementary CRISPR screen using HGC27 cells (Del10q23, *ATAD1*-null gastric cancer cell line) re-expressing ATAD1 or transduced with the empty vector. Remarkably, the top hit for synthetic lethality with *ATAD1* in this second screen was *MARCH5*. We discuss this screen in more detail above

5. There is evidence that in BRAF-mutant melanoma cells, BRAF inhibitors can induce BIM expression in a PTEN-sensitive manner (PMID: 21317224). It is therefore possible that PTEN loss may influence the ability of ATAD1 to regulate BIM levels. As such, assessing the role of PTEN on whether ATAD1 is able to regulate BIM is warranted.

We appreciate the reviewer asking about how the presence or absence of PTEN might affect the interaction of ATAD1 and BIM, which is an insightful question with important physiological implications that would be interesting to address in the future. Unfortunately, this is outside of the scope of our current study – we focus on *ATAD1* loss in cancer, which means our experiments are done in a *PTEN*-null background in order to isolate the variable of *ATAD1* status and to mimic the genetic background seen in human tumors. We hope the reviewer will accept our approach in light of their critique that our initial submission attempted “to unnecessarily explore too many aspects of the ATAD1/PTEN functional relationship.”

The reviewer states: “It is therefore possible that *PTEN* loss may influence the ability of ATAD1 to regulate BIM levels.” We conduct all of our experiments in *PTEN*-null cell lines, so clearly the **loss** of *PTEN* does not block the ability of ATAD1 to extract BIM. Again, we conducted our experiments in this genetic background because that is what is seen in human tumors that lack *ATAD1*. Understanding the functional relationship of *ATAD1* and *PTEN* in physiology and/or tumorigenesis would be a fascinating question to address in future work with genetically engineered mouse models.

6. Unclear why SW1088 cells did not form tumours in NOD/SCID given that there’s evidence they can grow in the less immunocompromised strain nu/nu (PMID: 25255031).

We thank the reviewer for their thorough review of relevant literature. In the paper by Lee et al. that the reviewer cites, the authors injected 10 million SW1088 cells into mice and then waited an unclear amount of time before beginning their experiments. The injection of ten million cells is quite extreme and more than 3x the number we injected (3e6 cells, which is standard), which may explain why this paper had better success engrafting SW1088 cells. Again, it would be helpful here if the Lee et al. described how long they waited until they observed tumors. As we cite in our manuscript, others have also found that SW1088 cells fail to engraft when injected subcutaneously in mice. For instance, Mercapide et al. (*International Journal of Cancer*, 2003) reported that SW1088 cells were nontumorigenic when injected subcutaneously into *Scid* mice.

There is no in vivo evidence of the potential therapeutic benefit of proteasome inhibitors in ATAD1-deficient tumours (i.e. using some of the isogenic models generated by the authors, but not SW1088 given that the parental cells will not form tumours in their hands).

As suggested by the reviewer, we conducted another xenograft experiment, this time using PC3 cells transduced with Cas9 + non-targeting sgRNA or sgRNA targeting *ATAD1*. Loss of *ATAD1* was compatible with tumor engraftment (Figure 4B). We treated mice harboring *ATAD1*proficient (sgNT) or deficient (sgATAD1) PC3 tumors with saline or bortezomib, and found that bortezomib slowed the growth of sgATAD1 tumors specifically (Figure 4B), while having no effect on the sgNT tumors (Figure 4A). These data represent in vivo validation of our model in which *ATAD1* protects cells from apoptosis induced by proteotoxic stress, which has direct implications for hundreds of thousands of cancer patients whose tumors have Chr10q23 deletion. Although we do see that proteasome inhibition is preferentially toxic to cells lacking *ATAD1* both in culture and in murine xenografts, the success of proteasome inhibitors in human patients is somewhat limited by on-target toxicity. We think that a MARCH5 inhibitor would be an exciting area for drug development, since we would predict that such an agent would achieve a maximal therapeutic window by selectively targeting *ATAD1*-null tumor cells while sparing the rest of the cells in the patient’s body, which have *ATAD1* intact. We hope that publication of our work will stimulate further investigation in this space.

Minor points:1. In Figure 2 A/B Authors claim BIM protein goes up (based on a-tubulin normalisation) while FIS1 does not. Can they show the normalised FIS1 data? Visually, it would seem FIS1 also changes.

We now immunoblot for several OMM localized and tail-anchored proteins in PC3 cells and have quantified all of these data (Extended Data Figure 16C,D). As suggested by the reviewer, FIS1 appears to be marginally increased. FIS1 may be an ATAD1 substrate or the cell might upregulate FIS1 in order to trigger compensatory mitochondrial fission in response to ATAD1 loss. This could reflect a component of the BIM-independent pro-survival function of ATAD1 as mitochondrial fission is known to be pro-apoptotic.

2. Figure 2C. There is no rescue on cPARP in sgBIM/ATAD1KO cells

We now use quantitative measures of cell viability to demonstrate that deletion of BIM partially rescues the synthetic lethality of *ATAD1* and *MARCH5* (Figure 2D). This represents a direct test and validation of our model in which ATAD1 protects cells from apoptosis induced by ubiquitin proteasome system dysfunction in part by extracting BIM from the OMM.

3. In Figure 2I, input blot suggests the ATAD1-FLAG/HA lane is underloaded, based on b-actin levels. If so, and considering there is some background GFP signal in the FLAG-IP EV lane, it is not entirely conclusive that there is co-IP. Perhaps the reverse IP/Western should be tried for confidence (IP with GFP, blot with HA/FLAG).

Our revised manuscript includes not only reciprocal Co-IP, as suggested by the reviewer, which more clearly demonstrates a physical interaction. More importantly, we have now included experiments that detect binding of ATAD1-FLAG with endogenous BIM (Extended Data Figure 6E). We hope the reviewer agrees that these data together more clearly demonstrate a physical interaction of ATAD1 and BIM. Of course, these data culminate with those from our in vitro extraction system, which show that BIM is a specific substrate for ATAD1-mediated extraction from proteoliposomes.

4. In Extended Data Figure 5, panels M and N show significantly different profiles for PC3 sg NT vs PC3 + EV. Explain

Because wildtype PC3 cells are not highly sensitive to bortezomib normally (hence, deletion of *ATAD1* confers sensitivity), we had to increase treatment duration to 36 hr rather than overnight. Our purpose in this particular experiment is to test whether ATAD1 overexpression decreases bortezomib sensitivity, and in order to do we had to use conditions in which the control cells were sensitive.

Referees' comments, Second Round:Referee #1 (Remarks to the Author):Many tumors harboring deletions in PTEN lack also the neighboring ATAD1 gene. The authors report that ATAD1 loss renders PTEN-deficient cancer cells more susceptible to apoptosis, which is consistent with the improved prognosis of PTEN-cancer patients with co-deletions of ATAD1. In agreement with the previously described role of ATAD1 for quality control in the OMM, ATAD1 is shown to extract the pro-apoptotic BIM from the membrane allowing its proteasomal inactivation. ATAD1- deficient cells accumulate BIM at mitochondria but are susceptible to proteasome inhibition, which the authors explain by the parallel, MARCH5-dependent proteasomal degradation of BIM. These findings point to therapeutic possibilities using proteasome (or MARCH5) inhibitors in cancer patients with co- deletions of PTEN and ATAD1.The authors have significantly advanced their analysis of the effect of a co-deletion of ATAD1 with PTEN in cancer. Additional experiments strengthen the importance of BIM for the effect of ATAD1 loss on the apoptotic vulnerability of cancer cells. They provide compelling evidence for ATAD1 specificity showing that it is able to extract Bim but not BIK or PUMA from liposomes. At the same time, the new experiments also establish that the increased apoptotic vulnerability of ATAD1 deficient cells involves additional proteins besides BIM, as is illustrated by the only partial restoration of the apoptotic resistance of ATAD1 deficient cells by depletion of BIM. Nevertheless, the experiments reveal an important role of BIM in this context and I feel that the authors carefully and appropriately discuss this issue in the revised manuscript. The authors also extended the analysis of the genetic interaction of BIM and MARCH5, leading to the significantly revised model that MARCH5 acts via NOXA while BIM5 controls BIM. This offers the intriguing possibility that cancer cells lacking both PTEN and ATAD1 might be targeted by inhibition of the UPS or MARCH5.Overall, I think the manuscript reports on the exciting discovery that co-deletion of ATAD1 increases the apoptotic vulnerability of PTEN deficient cancers. Although some mechanistic questions remain to be addressed, the reported findings will be of broad interest and are likely of significant medical relevance.

We thank the reviewer for their positive feedback and for their constructive insights during the first round of review that helped us improve our manuscript.

Referee #2 (Remarks to the Author):The revised manuscript includes some new data that help address some of the minor concerns previously raised. However, the main concerns I had regarding the contribution of ATAD1-mediated extraction and inactivation of BIM to the sensitization of ATAD1-deficient cancer cells to apoptosis remain unresolved. Although BIM contributes partially to the synthetic lethality of ATAD1 and MARCH5 in Jurkat cells (Figure 2D), the new data revealed little contribution of BIM to the sensitization of ATAD1- deficient PC3 cells to bortezomib (Extended Data Figure 16A). Furthermore, the previously presented data of increased BIM-EL levels in ATAD1-deificient compared to ATAD1-proficient SW1088 and RPMI7951 cells have been replaced with new data that no longer showed altered BIM-EL levels in response to ATAD1 loss or restoration (Extended data Figure 13A, 13D, and 16C). My overall impression is that the authors have identified a few interesting and seemingly related findings of ATAD1, such as the synthetic lethality of ATAD1 and MATCH5, ATAD1-mediated extraction of BIM, the sensitization of ATAD1-deficient cells to proteasome inhibitors, and the association of ATAD1 loss to improved clinical outcomes. However, key data to integrate these findings together as a coherent story are still missing. Specific Comments:1. The impact of ATAD1 loss on BIM-EL levels was only present in Jurkat cells (Figure 2A) but not in SW1088 (Extended Data Figure 13A) and PC3 cells (Extended data Figure 13D and 16C).

BIM protein levels increase upon loss of ATAD1 in many cell contexts (for example, Figure 2A, 2B, Extended Data Figure 16C). Nevertheless, we have made the reproducible observation that ATAD1 inactivates BIM via mechanisms beyond simply degradation of the protein. Specifically, we have added new experiments in the revised manuscript using high-resolution imaging, which demonstrates ATAD1 promotes the non-mitochondrial localization of BIM (Extended Data Figure 9-12). BIM has been previously shown to be inactivated by changes in localization (Kale, Osterlund, and Andrews, Cell Death & Differentiation 2018), but how it translocates from the OMM to the cytosol was previously unknown. BIM requires OMM localization to bind and inhibit anti-apoptotic proteins, such as MCL1, and to activate the pore- forming proteins BAX/BAK. Hence, BIM localization is a key parameter that dictates its pro- apoptotic activity.

In addition, we have made the exciting new discovery that ATAD1 promotes the phosphorylation of BIMEL by cytosolic kinases (Figure 2B; Extended Data Figure 4A,B; Extended Data Figure 13). Phosphorylation of BIMEL has been clearly established as an important mechanism by which kinases like ERK promote cell survival. Phosphorylation of BIMEL reduces its pro-apoptotic function by decreasing its binding affinity for pro-survival proteins like MCL1, preventing its activation of BAX, and/or accelerating its degradation, depending on the context (e.g., PMID: 17151701; PMID: 29775995;PMID: 18498746). We now show that ATAD1 loss decreases BIMEL phosphorylation, while re-expressing ATAD1 increases BIMEL phosphorylation, and identify residues Ser69 and Ser77 as being phosphorylated in this context (Figure 2B; Extended Data Figure 4A,B; Extended Data Figure 13A,D-F). Of course, ATAD1 is not a kinase, so our model is that ATAD1 extracts BIM and, by so doing, puts it into a biochemical environment that promotes its phosphorylation by kinases such as ERK and Aurora A/B. In the context of proteasome inhibition, ATAD1-null cells accumulate unphosphorylated (active) BIM, while ATAD1 positive cells accumulate phosphorylated (inactive) BIM (Extended Data Figure 13A,D-F).

We propose that ATAD1 extraction of BIM from the OMM prevents it from activating apoptosis, and subsequent phosphorylation of BIM enforces its inactivation via multiple mechanisms. We propose that this may partially explain how ATAD1 is protective against apoptosis even in the context of proteasome inhibition – if the sole fate of ATAD1-extracted substrates were immediate proteasomal degradation, then ATAD1 would be inconsequential in the context of an inhibited proteasome (e.g., bortezomib treatment). Instead, we clearly demonstrate that ATAD1 is most important to cell fitness in the context of ubiquitin-proteasome system dysfunction (Figure 1E, G; Figure 2D,E; Figure 3; Figure 4A,B).

2. Can the authors explain why sgBIM had no effects on bortezomib-induced apoptosis of PC3 cells (± sgATAD1)?

Despite our ruling out all of the prime suspects, the genetic data suggest that other factors beyond BIM are also involved in the pro-survival function of ATAD1. To acknowledge that ATAD1 likely acts through proteins in addition to BIM, we have now revised the manuscript to more precisely indicate that while BIM is clearly an important and direct substrate of ATAD1,there are likely other substrates that are relevant to the cytoprotective function of ATAD1. We state the following in our Results section:

“Although caspase inhibition completely rescued the bortezomib phenotype of ATAD1- deficient cells, the same was not true for deletion of BIM in PC3 cells, which means that there must be other factors in addition to BIM that mediate this phenotype (Extended Data Figure 16A,B). We examined several OMM-localized or tail-anchored proteins in PC3 cells with and without bortezomib, and reduced BIM phosphorylation was the only consistent change caused by *ATAD1* deletion (Extended Data Figure 16C). Other BH3-only proteins reported to be activated by proteotoxic stress include BIK, PUMA, and NOXA, but our in vitro extraction assay ruled out a direct action of ATAD1 to extract these proteins. Increased mitochondrial fragmentation in ATAD1-null cells^2^ could be one potential explanation, since mitochondrial dynamics are intimately connected to apoptosis. Nonetheless, these results indicate that the protective effects of ATAD1 during proteasome inhibition can be explained exclusively by limiting apoptosis, with BIM extraction playing a key role.”

The identities of these other hypothetical substrates are unknown. Deficiency of *ATAD1* has been previously shown to increase mitochondrial fragmentation (Chen et al., EMBOJ 2014), so it may be that ATAD1 extracts substrates that regulate mitochondrial dynamics. Changes in mitochondrial fission and fusion have an established role in regulating the apoptotic threshold, which has been known and studied for nearly 20 years (Youle and Karbowski, Nat Rev Mol Cell Bio, 2005). Our lab and others have shown that ATAD1 is an evolutionarily conserved protein quality control factor, so this more generic protective function could also contribute to the pro- survival role of ATAD1 in the context of protein stress. However, given the strong synthetic lethality with *MARCH5* across multiple cell types, we are confident in our initial focus on a direct role in the regulation of apoptosis.

3. Can the authors explain why BIM was not stabilized/increased in bortezomib-treated PC3 cells (Extended data Figure 13D and 16C)?

We appreciate the reviewers pointing this out so that we can clarify this issue. BIM is stabilized in BTZ-treated PC3 cells, but the phosphorylation-induced shift makes it difficult to appreciate. We quantified the BIM_EL_ intensity for all phosphorylated and unphosphorylated bands in sum and replotted the data in Extended Data Figure 16D.

4. The quality of anti-NOXA western blots shown in Figure 2C and Figure 4C is suboptimal.

We apologize for the quality of the NOXA western blots in the manuscript, which we have been unable to improve in spite of significant effort. NOXA is a very small (10-12 kD) protein that shows vertical diffusion when resolved by SDS-PAGE with a high percentage of acrylamide. Alternatively, using SDS-PAGE gels with low acrylamide concentration led to poor resolution of NOXA, with the protein migrating along with the dye front. Our blots look very similar to those of others.

5. The in vivo anticancer efficacy of bortezomib in ATAD1-deficient PC3 mouse xenografts is not impressive, which may limit its translational potential.

We appreciate the reviewer pointing out this limitation of bortezomib studies in mice, which seems to have a narrower therapeutic window than humans. We used 1 mg/kg, which is the maximum tolerated dose in mice. Other studies have found that even this dose is not always tolerated, depending on mouse strain and other experimental variables (PMID: 17374740; 28800777). Nevertheless, bortezomib is an approved cancer therapy and is efficacious in humans, so we anticipate that the effect might be larger in humans where it is not masked by toxicity. Furthermore, there are other experimental therapeutics in clinical development that inhibit this pathway, broadly defined, such as VCP/p97 inhibitors (CB-5083; Anderson et al.Cancer Cell 2015). ATAD1 status of tumors could be an important consideration in designing clinical trials for these drugs, since our work predicts that patients whose tumors lack ATAD1 might be more susceptible to those drugs as well. Finally, our work provides rationale for the development of a specific inhibitor of MARCH5, which does not yet exist. Our genetic screening experiments found a synthetic lethal interaction between ATAD1 and MARCH5, specifically, so the ideal drug target in ATAD1-null tumors might be MARCH5, not the proteasome. We have initiated efforts in this direction.

6. The presence of ATAD1 does not appear to alter the correlation between GFP-BIM and mito- mCherry (Extended Data Figure 10B), which argues against the statement that ATAD1 promotes non- mitochondrial localization of GFP-BIM.

We thank the reviewer for pointing this out and agree that the presence or absence of ATAD1 does not alter the localization of GFP-BIM under **basal** conditions. GFP-BIM and mito- mCherry show a decrease in colocalization upon treatment with bortezomib, but only in the ATAD1+ condition. Furthermore, we quantify the appearance of non-mitochondrial GFP+ puncta, which occur at a much higher frequency and number in ATAD1+ cells treated with bortezomib compared to ATAD1(–) treated with the same (P < 0.001).

When taken together, these data support the conclusion that ATAD1 promotes non- mitochondrial localization of GFP-BIM under conditions of proteasome inhibition. The data suggest that there are multiple redundant mechanisms that can affect BIM localization and degradation, which is why the ATAD1-dependent mechanism becomes more important when the proteasome is inhibited.

Referee #3 (Remarks to the Author):In the revised manuscript “co-deletion of ATAD1 with PTEN sensitizes cancer cells to protein stress” by Jacob Winter et al., the authors conducted many new experiments to address the concerns of the reviewers. Despite these valiant efforts, key issues remain which collective failed to generate a coherent thesis reinforced across multiple and physiological relevant systems. Several issues including the need to conduct CRISPR screens in more clinically relevant cancer cell lines model including prostate cancer cell lines; the need to document synthetic lethality (as opposed to synthetic ‘sick’ interactions between MRH5) and ATAD1 both on the genetic and pharmacological levels in multiple cell lines; the remaining uncertainties surround the mechanisms of how ATAD1 regulates BIM and MCL1; and the inconsistent data in different cell lines and limited in vivo studies among different types of tumors. Collectively, the story presents a contorted model based on retrofitted data that may not stand the test of time. Thus, while this reviewer deeply appreciates many aspects of the story and good components in the experimental data, I hold concerns that this thesis is not sufficiently robust to merit publication.

We are glad to hear that the reviewer “deeply appreciates many aspects of the story,” and recognizes “good components of the experimental data.” We thank the reviewer for acknowledging our “valiant” efforts” to conduct “many new experiments” to address their original concerns.

Major points:1. “Would MARCH5 emerge in screens with other cell type”?In the revised manuscript the authors included gastric cancer cell line HGC27 (to complement their work in Jurkat cells) to conduct another CRISPR screen. They claim that the top hit is still MARCH5 by comparing log2 fold change of sgRNA abundance. This is not convincing. The authors should analyze the CRISPR screening data by scoring the fitness of the gene deletion via BAGEL analysis. In addition, the authors state that HGC27 is a gastric cancer cell line, representing an ATAD-deficient tumor type with the second highest number of deaths. It should be noted that HGC27 harbors a large Chr10q23 deletion, making it a less ideal screening platform given the large genetic deletion harboring many genes. It is disappointing that the CRISPR screens were not conducted in prostate cancer cell lines (such as PC3 with isogenic WT and deleted ATAD1 and/or LNCaP which has a deep PTEN-ATAD1 co-deletion according to the CCLE). This is particularly relevant given that this cancer type harbors highest frequency of PTEN/ATAD1 deletion. This remains a major concern as to whether the conclusions hold true in other cancer types (i.e. prostate cancer, glioblastoma, and lung squamous cell carcinoma) that appear to be more relevant in terms of PTEN and ATAD1 co-deletion. It is perhaps telling that some experiments in PC-3 did not mirror those of others systems or showed meager differences (Extended Figure 16). To prove that the findings are robust and of broad interest, the central conclusions must be confirmed in other relevant cancer types.

There are many points embedded in the above paragraph, and we address them here.

1) The top-ranked gene in terms of differential essentiality based on the presence or absence of *ATAD1*, was *MARCH5*. We also conducted MAGeCK VISPR and determined that *MARCH5* was an essential gene in EV cells (*ATAD1* deficient) but not in *ATAD1* re-expressing cells. Presenting data based on differential CRISPR score is common practice in the field. We include the raw data from our CRISPR screens so that readers can analyze the data in any way that suits their predilections.

More importantly, *MARCH5* was the only hit shared by the two screens that we performed, HGC27 and Jurkat. Whether *MARCH5* truly represents the best genetic vulnerability in all extant *ATAD1*-null cells is a nearly impossible question to answer and seems to be beyond the scope of our current study. We identified a strong genetic interaction independently in two distinct cancer cells and, from that starting point, developed a deeper understanding of a biochemical network that appears to be an important and targetable vulnerability of a sizeable subset of *PTEN*-null cancers.

2. Along similar lines, the CRISPR screening results should be further validated by depleting MARCH5 and other hits in ATAD1 null models (ATAD1-δ Jarkat, H4, RPMI7951, prostate cancer cells). The mRNA and protein levels of MARCH5 need to be tested in different ATAD1 WT and deleted cancer cell lines. Any differences in MARCH5 level as compensation for ATAD1 deletion?Review, 2^nd^ round:This essential point is not addressed and synthetic lethality between MARCH5 and ATAD1 was not validated in multiple cell lines including prostate cancer cells as suggested. In Figure 2D, deletion of MARCH5 resulted in a decreased cell viability in both WT and ATAD1 KO Jurkat cells, and ATAD1 KO Jurkat cells exhibited higher sensitivity to MARCH5 deletion; however, it is notable that up to 30% to 40% of Jurkat cells were still alive when ATAD1 and MARCH5 were co-deleted, consistent with a synthetic disadvantage as opposed to a true synthetic lethal relationship. In addition, Figure 4B clearly showed that proteasome inhibition only caused a delayed growth; the tumor neither shrank nor plateaued. A synthetic lethal interaction typically shows evidence of tumor shrinkage or extensive cell death (e.g., BRCA/PARP). Regarding the MARCH5 mRNA/protein levels, the authors did not validate in ATAD1 WT or deleted cancer cells. The DepMap plot showed that there is still MARCH5 expression in ATAD1 deleted cells, more specifically, the expression level is similar to cells with intact ATAD1.

That *MARCH5* was synthetic lethal with *ATAD1* in both Jurkat cells and HGC27 cells, despite taking two completely different approaches to study *ATAD1* deficiency (KO vs. re-expression) in two very different cell types, validates the genetic interaction. It is unclear what would be gained by repeating the screen or validating this genetic interaction in several other cell lines. Instead, we have taken the approach to understand the basis for this genetic interaction, which led to the discovery that ATAD1 inactivates BIM to limit the propensity for apoptosis. The reviewer is correct that, at 4 days post-transduction, there is a viability decrease in WT cells upon *MARCH5* deletion. However, this is significantly less than the viability decrease observed in *ATAD1*∆ cells, which is the reason for including that comparison as a control. Our study was focused on identifying genes that differentially impact cell fitness—and *MARCH5* certainly fits that criterion. By 8 days post-transduction, there is an even greater difference in viability upon *MARCH5* deletion in WT vs. *ATAD1*∆ cells.

We thank the reviewer for noticing the distinction between “synthetic sick” and “synthetic lethal”. Indeed, by the most strict definition *ATAD1* and *MARCH5* do not exhibit a “synthetic lethal” genetic interaction. However, the term “synthetic lethal” has been adopted in the literature as a general descriptor that is synonymous with a negative genetic interaction, rather than specifically describing true lethality. We appreciate the reviewer’s use of the BRCA/PARP relationship as a gold standard of synthetic lethality, and we will return to this example below.

The reviewer seems to expect that negative genetic interactions should be reflected in changes in MARCH5 mRNA/protein levels in *ATAD1*-deficient cells. Synthetic lethal (or negative genetic) interactions often involve two redundant, parallel pathways, as we think occurs with *ATAD1* and *MARCH5*. If two genes are redundant, then loss of one gene does not require upregulation of the second gene. Or, as we stated in our first rebuttal, “if two pathways are truly redundant, then blocking one pathway will not require compensatory upregulation of the other pathway.” It seems that the reviewer’s expectation is that if *ATAD1*/*MARCH5* are synthetic lethal, then *MARCH5* should be upregulated in *ATAD1*-null cells. This would not be unexpected if true, but need not be the case. Indeed, as described below, the classic BRCA/PARP genetic relationship doesn’t exhibit such changes in mRNA abundance.

3. The authors claimed that BIM was sufficient to trigger apoptosis preferentially in ATAD1 KO cells; loss of MARCH5 and ATAD1 antagonized the anti-apoptotic function of MCL1 via increased abundance and activity of BIM and NOXA. Figure 2C also showed that deletion of ATAD1 and/or MARCH5 increased MCL1. These and other data together indicate that MARCH5 and ATAD1 converge on MCL1. However, the authors indicate in the rebuttal letter that tested cell lines show inconsistency in response to MCL1 inhibitor; one ATAD1 KO Jurkat cell clone is insensitive to MCL1 inhibitors; ATAD1 protects HGC27 cells but not PC3 cells from MCL1 inhibitors. These observations again raise the concern relating the general nature of the conclusions across cancer types.

Before we address the specific concern, we will first make a general comment about the relationship between evolution and apoptosis in cancer cells. Given the nature of all cancer cell lines, they are under profound selective pressure to minimize their susceptibility to apoptosis. The exact mechanisms through which they accomplish this necessity seem to be idiosyncratic. As a result, it is typical for different cell lines to be differentially dependent on (or sensitive to) anti-apoptotic and pro-apoptotic members of the BCL^-^2 family as well as the genes and proteins that regulate them. Therefore, the fact that different cell types exhibit differences in the genetic relationships between BIM, NOXA, MCL1, etc. is not at all surprising due to their differences in natural history.

As it relates to our manuscript specifically, the BIM/NOXA/MCL1 axis is clearly a major component underlying the synthetic lethality of *ATAD1* and *MARCH5*. However, as we state multiple times, this mechanism cannot fully explain the genetic interaction that we have discovered. For instance, overexpression of *MCL1* only partially rescues the synthetic lethality of *ATAD1*/*MARCH5* (Extended Data Figure 5). It seems likely that MCL1 antagonism as well as other insult(s) add up to account for the phenotype of *ATAD1*/*MARCH5* double knockout cells. In light of those data, it is not surprising that different cell types are more or less susceptible to direct and specific antagonism of MCL1.

Pharmacologic inhibition of MCL1 differs from *MARCH5* knockout or proteasome inhibition. MCL1 inhibitors unleash whatever pro-apoptotic binding partners happen to exist on the OMM at the time of drug treatment. *MARCH5* knockout or proteasome inhibition causes an increase in the levels of specific proteins, especially NOXA, which is a fundamentally different mechanism of action. Thus, BH3 mimetics can only exploit a pre-existing imbalance in pro- and anti-apoptotic members of the BCL2 family, while *MARCH5* knockout or proteasome inhibition can **generate** an imbalance. What appears to be uniquely toxic to *ATAD1*-null cells is the latter, and not always the former.

4. Figure 2A: The differences in BIM expression in both the wildtype and ATAD1-null cells are rather meager by western blot. The authors should check the expression of BIM in both mitochondrial and cytoplasmic fractions. Also, ATAD1 deletion increases both MCL1 and BIM, in a way similar to MARCH5 deletion as described in the text. While the author studied the mechanism of how ATAD1 extracts BIM from mitochondrial, there was no attempt to understand findings concerning MCL1 which would seem important.Referee, 2^nd^ round:It is still not clear how ATAD1 extracts BIM from OMM and how this extraction facilitates BIM downregulation/degradation. In the rebuttal figure, the authors detected increased BIM in the crude mitochondria of ATAD1 deleted cells compared to ATAD1 WT cells, while no BIM was detected in cytosol for both WT and ATAD1 deleted cells. Why is that? Further validation is warranted. No clear change in localization of GFP-BIM could be observed despite addition imaging pictures in extended data Figure 10, 12. Higher quality and higher resolution images and quantitation are required. In addition, the assessment of interaction between ATAD1 and BIM is not convincing as it was evaluated in the context of ATAD or BIM1 overexpression. Does endogenous ATAD1 interact with endogenous BIM in ATAD1 WT cells? It iss not known how ATAD1 deletion increases NOXA and MCL1 levels, nor how increased BIM/NOXA antagonizes increased MCL1 to induce apoptosis.

The post-translational regulation of BIM is extensive and has been the subject of much work. One aspect of this regulation is BIM localization. The canonical model is that BIM binds to microtubules in an inactive state, and then becomes active and translocates to the OMM. Other studies have shown that BIM on the surface of other nearby organelles. In untreated Jurkat cells, we detect endogenous BIM in the crude mitochondria fraction. This fraction is enriched for mitochondria, other organelles and cytoskeletal elements. BIM is likely undetectable in the cytoplasm, even when it is non-mitochondrial by imaging, due to its stable association with other membranes or microtubules. This hypothesis is supported by our imaging data, where we see the accumulation of cytoplasmic puncta of BIM upon treatment with bortezomib.

The reviewer states: “the assessment of interaction between ATAD1 and BIM is not convincing as it was evaluated in the context of ATAD or BIM1 overexpression.” We wish to point out that this is the first time anyone has shown co-immunoprecipitation between endogenous levels of a tail-anchored protein (in this case BIM) and overexpressed *ATAD1*. It is difficult to observe interactions between AAA+ ATPases and their substrates, since these are typically transient interactions and multiple substrates often exist for a particular AAA+ ATPase. We propose that the fact that we see reciprocal co-immunoprecipitation in cells and direct and specific extraction in vitro altogether provides compelling evidence that BIM is an ATAD1 substrate.

“It is not known how ATAD1 deletion increases NOXA and MCL1 levels, nor how increased BIM/NOXA antagonizes increased MCL1 to induce apoptosis.” Here the reviewer points out interesting unanswered questions that are beyond the scope of the current manuscript.

Regarding the second point, since both BIM and NOXA can antagonize MCL1, it stands to reason that an increase in all three would lead to increased levels of free BIM that could activate BAX/BAK oligomerization and trigger MOMP.

5. The authors concluded that BIM could be phosphorylated by cytosolic kinases without convincing data. BIM phosphorylation shown as increased mobility band via PhosTag detection is faint and not clear (Figure 2B, extended Figure 4). Changes in expression levels of total Bim and phosphorylated Bim following ATAD1 KO was meager compared to control. In addition, how these samples were prepared? From whole or cytosolic or mitochondrial fraction? Did IP show the interaction between BIM and potential kinases ERK/Aurora or other kinases in cytosol? How ATAD1 mediated extraction of BIM from mitochondria facilitated BIM phosphorylation and how this phosphorylation regulated/promoted degradation? Would BIM mutant with the two potential phosphorylation sites (Ser69/77) mutated to Ala be more resistant to degradation, or the phosphorylation mimic mutants would be degraded more? Given that BIM could be inactivated by cytosolic kinases (i.e. ERK – particularly in the context of PTEN deficiency), the authors also need to evaluate the level of these kinases in ATAD1 WT and KO cells as they can potentially compensate the inhibitory effect of MARCH5 and ATAD1 on BIM.

We appreciate these many excellent questions posed by the reviewer. Answering many of them will be the subject of multiple future papers from our lab. However, we will address those that lie within the scope of the current manuscript. To begin, unless explicitly stated, all western blots from cell culture experiments were conducted using whole-cell lysates (this will be further emphasized in the revised manuscript). We observe the most striking changes in BIM phosphorylation based on *ATAD1* status when we treat cells with proteasome inhibitors. We have quantified these data and explicitly annotated an example blot to demonstrate to the reader which bands are important. The phosphorylation effects seem quite apparent and obvious to us; they are consistent with the published data showing BIM phosphorylation in the literature.

6. Figure 1G: Do the cell lines in the lower left corner harbor PTEN and ATAD1-loss? If not, can we still conclude that MARCH5 or MCL1 is essential in ATAD1-null cancer cells in general?Referee, 2^nd^ round:I do not agree with the statement – “The analysis of synthetic lethality using a resource like DepMap can be confounded by the broad diversity of cell lines included in the “denominator. In other words, the signal cannot overcome the noise.” Actually it would instead enhance the robustness of the data if it were a true genetic interaction of broad relevance. The authors at least could have cross-referenced with other many other synthetic lethal datasets to further validate their observations with Jukart. Authors stated that “at least in its current iteration, DepMap is not as well suited to identify synthetic lethal interactions”, which amplifies my concerns as to the validity of the model.

Rebuttal, 2^nd^ round:

As present in the round 1 rebuttal, using the analysis requested by the reviewer, DepMap fails to identify the two most validated synthetic lethal interactions, that between BRCA and PARP inhibitors. It is, therefore, clear that to conclude that lack of evidence for an *ATAD1/MARCH5* interaction means it is not broadly relevant is a mis-use of the DepMap resource.

We feel that the reviewer neglected our example of how DepMap is ineffective at identifying synthetic lethal relationships. The reviewer states: “Actually it would instead enhance the robustness of the data if it were a true genetic interaction of broad relevance.” One can easily imagine how a database like DepMap that compiles gene expression and gene dependency could *theoretically* be used to find relationships where low expression of gene X predicts increased dependency on gene Y. In practice, such analyses are murky at best.

We have presented the data from a DepMap analysis wherein we compared Olaparib (PARP inhibitor) sensitivity between two groups of cell lines: BRCA mutant vs. wildtype. We grouped all cell lines that have *BRCA1* and/or *BRCA2* mutations, then used all remaining cell lines as the “out group.” Author response table 3 which display the results, using two distinct drug sensitivity datasets (PRISM and CTD^2^) to conduct the same analysis. DepMap includes a two-class comparison algorithm which consists of linear hypothesis test followed by a multiple hypothesis correction step. The code used for this two-class comparison is available on github: https://github.com/broadinstitute/cdsr_models/blob/master/R/linear_association.R

**Author response table 3. sa2table3:** 

		*Olaparib sensitivity (PRISM AUC)*			
In Group	Out Group	Effect size	P-Value	Q-Value	Number of cell lines
BRCA1 and/or BRCA2 mutant	All other cell lines	-0.000769	0.203	**0.826**	338
					
*Olaparib sensitivity (**CTD^2^ AUC**)*					
**In Group**	**Out Group**	**Effect size**	**P-** **Value**	**Q-** **Value**	**Number of cell lines**
BRCA1 and/or BRCA2 mutant	All other cell lines	0.00537	0.905	**0.867**	772

To further emphasize the point that tools like DepMap (which are excellent at finding co- essential genes) are not suited to identify synthetic lethal interactions, we conducted an additional analysis using established gene pairs of “collateral lethality.” For these tests we compared expression of gene 1 with dependency of gene 2, where gene 1 is deleted in cancer and gene 2 is the potential therapeutic target (“Achilles’ Heel”). We also included *BRCA1* vs. *PARP1* and *BRCA2* vs. *PARP1*.

The best performing gene pair by this analysis is *ENO2*/*ENO1* (Muller et al., 2012). The Pearson and Spearman correlation coefficients are 0.279 and 0.296, respectively, with a P-Value (Pearson) of 2.27E-18. This is the pioneering example of collateral lethality, in which one isoform of a glycolytic enzyme is co-deleted with a tumor suppressor locus, which makes the other become more important for cell fitness. It is difficult to imagine a more straightforward and clearer example of conditional gene essentiality.

Another example of collateral lethality is *ME2*/*ME3* (Dey et al., 2017). We can use this robust example (an elegant study published in the last 5 years by one of the top labs in the field) to demonstrate the limits of DepMap. In Author response image 8, and we plot *ME2* expression on the x-axis and *ME3* gene effect on the y-axis.

**Author response image 8. sa2fig8:** 

If DepMap were able to identify this genetic relationship, then we would expect to see a positive slope: the data points that are on the very left side of the x-axis (low *ME2* expression) should have lower y values than the rest of the data points, indicating preferential dependence on *ME3*. Here, however, we see that there is no correlation between *ME2* expression and *ME3* dependency.

The Chronos gene dependency algorithm classifies a value of (-1) as being the median of essential genes, which is a useful yardstick with which to judge the dependency of other genes. Here, we see that not one of the cell lines in DepMap has a dependency on *ME3* anywhere close to -1, and certainly one could not justify the claim that *ME3* is uniquely essential in the *ME2*-null cells (leftmost on the x-axis). That is to say, not one of the cell lines in DepMap shows a genetic dependency on *ME3*, irrespective of *ME2* expression. Even so, Dey et al. perform many elegant and robust experiments to demonstrate that *ME3* is selectively essential in pancreatic ductal adenocarcinoma (PDAC) tumors that co-delete *ME2* along with the tumor suppressor *SMAD4*. The correlation remains insignificant (P = 0.7721) even when we filter cell lines to select only those derived from pancreas tumors (as was the focus of the Dey et al. paper).The correlation between *BRCA1/2* expression and *PARP1* dependency is statistically significant, but with unconvincing coefficients (shown in Author response table 4). As in the BRCA/PARP examples, there is a weak but statistically significant correlation between *ATAD1* expression and *MARCH5* dependency. To reiterate, the coefficients and statistical significance of the correlation between *ATAD1*/*MARCH5* is very similar to those of *BRCA2*/*PARP1*.

**Author response table 4. sa2table4:** DepMap: Expression of Gene 1 vs.Dependency on Gene 2.

Paper	Gene 1 (Deletedgene)	Gene 2 (Syntheticlethal partner)	Pearson	Spearman	P-Value
Dey et al. Nature 2017	*ME2*	*ME3*	-0.011	0	7.46E-01
Zhao et al., Nature 2017	*PTEN*	*CHD1*	0.049	0.039	1.36E-01
Fan et al., *eLife*					
2017	*FXR2*	*FXR1*	0.022	-0.011	5.07E-01
Kryukov et al.,					
Science 2016; Mavrakis et al.,					
Science 2016	*MTAP*	*PRMT5*	0.170	0.162	1.52E-7
Muller et al., Nature					
2012	*ENO2*	*ENO1*	0.279	0.296	2.27E-18
	*BRCA1*	*PARP1*	0.151	0.159	3.14E-06
	*BRCA2*	*PARP1*	0.096	0.090	3.12E-03
Winter et al.	*ATAD1*	*MARCH5*	0.086	0.116	**8.14E-03**

Follow the links below to reproduce the indicated DepMap plots

ME2/ME3:

https://depmap.org/portal/interactive/?filter=&regressionLine=true&associationTable=false&x=slice%2Fex pression%2F17519%2Fentity_id&y=slice%2FChronos_Achilles%2F17521%2Fentity_id&color=PTEN/CHD1:

https://depmap.org/portal/interactive/?filter=&regressionLine=true&associationTable=false&x=slice%2Fex pression%2F26155%2Fentity_id&y=slice%2FChronos_Achilles%2F4408%2Fentity_id&color=FXR2/FXR1:

https://depmap.org/portal/interactive/?filter=&regressionLine=false&associationTable=false&x=slice%2Fe xpression%2F9438%2Fentity_id&y=slice%2FChronos_Achilles%2F9437%2Fentity_id&color=MTAP/PRMT5:

https://depmap.org/portal/interactive/?filter=&regressionLine=true&associationTable=false&x=slice%2Fex pression%2F20467%2Fentity_id&y=slice%2FChronos_Achilles%2F25783%2Fentity_id&color=ENO2/ENO1:

https://depmap.org/portal/interactive/?filter=&regressionLine=true&associationTable=false&x=slice%2Fex pression%2F7668%2Fentity_id&y=slice%2FChronos_Achilles%2F7661%2Fentity_id&color=BRCA1/PARP1:

https://depmap.org/portal/interactive/?filter=&regressionLine=true&associationTable=false&x=slice%2Fex pression%2F2479%2Fentity_id&y=slice%2FChronos_Achilles%2F23906%2Fentity_id&color=BRCA2/PARP1:

https://depmap.org/portal/interactive/?filter=&regressionLine=true&associationTable=false&x=slice%2Fex pression%2F2481%2Fentity_id&y=slice%2FChronos_Achilles%2F23906%2Fentity_id&color=ATAD1/MARCH5:

https://depmap.org/portal/interactive/?filter=&regressionLine=true&associationTable=false&x=slice%2Fex pression%2F1697%2Fentity_id&y=slice%2FChronos_Achilles%2F17328%2Fentity_id&color=

Altogether, we hope these analyses convey that negative data from DepMap should not be used to invalidate more focused work with extensive follow up studies.

To be clear, we are not arguing that DepMap is ineffective, generally. On the contrary, it is a remarkably effective tool to identify ***co-essential*** genes, or genes that tend to be important for fitness in the same cell types. For example, consider two subunits of mitochondrial complex I, *NDUFC1* and *NDUFC2*. These genes function in a similar capacity – therefore, the same cell lines that require *NDUFC1* for fitness will also require *NDUFC2* for fitness. This is clearly demonstrated in DepMap (Pearson coefficient = 0.791, Linear regression P-value = 1.93E-223) (Author response image 9) :

7. Authors conducted extra xenograft experiment using PC3 cells transduced with Cas9 + non-targeting sgRNA or sgRNA targeting ATAD1 and they claimed that loss of ATAD1 conferred sensitivity to bortezomib treatment in vivo (Figure 4B). However, the in vivo data is not convincing – the effect is marginal and with high variation (is there statistical significance), perhaps relating to the limited sample size.

**Author response image 9. sa2fig9:** 

Although there is spread of the data points, we observe a statistically significant difference in tumor size over time with bortezomib treatment in sgATAD1 tumors, but not sgNT tumors. The statistical significance is indicated by the asterisk, as described in the figure legend. The therapeutic window of proteasome inhibitors appears to be narrower in mice than in humans.

We used the maximum tolerated dose (1 mg/kg), but even this dose is sometimes toxic by itself as described in the literature, depending on other experimental parameters. We also hypothesize that the development of a MARCH5 inhibitor would enable superior targeting of *ATAD1*-null tumors. Nonetheless, the proteasome inhibitor data is a valuable proof of principle, and given that proteasome inhibitors are already used safely in human patients, the barrier is low for establishing a clinical trial for proteasome inhibition in patients with *ATAD1*-null tumors.

8. It is not clear how MARCH5 functions as synthetic lethal target of ATAD1. How does MARCH5 regulate NOXA in some cells but both BIM and NOXA in other cells? The claim that MARCH5 deletion increased NOXA levels is not convincing (Figure 2C). In addition, the knockout efficiency of MARCH5 needs to be confirmed (Figure 2D).

The posttranslational regulation of the BIM/NOXA/MCL1 axis is complex. The central claims of our paper are that BIM is an ATAD1 substrate and that *ATAD1* loss sensitizes to BIM and to inhibition of the ubiquitin proteasome system. Understanding the precise mechanisms of MARCH5 substrate selection in different contexts is outside the scope of the present manuscript, but would be an interesting area for further study. As requested, we now confirm the knockout efficiency of sgMARCH5 transduction by western blot. MARCH5 is the lower band, indicated by the arrow on the left-hand side, and not the nonspecific band, which is indicated by the asterisk (Author response image 10).

**Author response image 10. sa2fig10:** 

9. “These experiments showed us that our initial model was incomplete, since the main effect of MARCH5 deletion in this context was an increase of NOXA, rather than BIM. It is clear that loss of MARCH5 increases BIM in some cell types (Arai et al. eLife 2020), but apparently not in Jurkat cells, which we used for our mechanistic follow up experiments. Based on our new data, we have revised our model to propose that MARCH5 inhibits NOXA, while ATAD1 inhibits BIM, and these two pathways converge on MCL1. Thus, co-deletion of ATAD1 and MARCH5 triggers a concomitant increase of BIM and NOXA, which combine to inhibit MCL1 and activate apoptosis (schematic in Figure 4G).”The authors presented a contorted model with a patch quilt of cell lines to retrofit the mechanism into the model, which was mainly based on the results from Jurkat cells. BIM levels were not substantially different in ATAD1 intact or deleted PC3 and SW1088 cells (extended Figure 16C). The authors elected to omit the experiments involving AMG176 due inconsistencies in different cell lines -- such as ATAD1 protects HGC27 cells but not PC3 cells from this agent, reinforcing the insufficient validation of the synthetic lethal relationship of ATAD1 and MARCH5 either genetically or pharmacologically in vitro or in vivo.

We are disappointed to hear the reviewer’s characterization of our model. From our point of view, we performed many additional experiments in response to the 4 reviewers’ critiques, gained new insight, and *changed the model to fit the new data*. We learned more about how ATAD1 affects cell survival under conditions of protein stress, incorporated this new and exciting data into our manuscript, and necessarily added to our model to represent those new data.

The reviewer is correct that the primary cell line we use to study the mechanistic and genetic questions in our manuscript is Jurkat T cells. We intentionally used this same cell line to conduct biochemical experiments in a genetically tractable system. Cell lines are model systems just like *S. cerevisiae* and mice are model systems. We started our work using Jurkat cells for a genetic screen, conducted more mechanistic genetic work in Jurkat cells, and then expanded to other cell types to test for relevance in the cancer types of relevance for Chr10q23 deletion.

The reviewer places undue emphasis on using an MCL1 inhibitor, AMG176, as a sole means of testing our mechanism. We know that the *ATAD1*/*MARCH5* genetic relationship converges on MCL1, because when we overexpress *MCL1* we can rescue the phenotype.

Similarly, deleting an antagonist of MCL1, BIM, partially rescues the phenotype. Different cell lines have different genetic backgrounds, and in the case of the BCL2 family, strong selective pressures act to suppress apoptosis during the course of cell line (or clonal KO line) establishment and maintenance. For instance, SW1088 cells lack NOXA, PC3 cells lack PUMA, while Jurkat cells lack BAX – if different cell lines differ in these fundamental components of the BCL2 family, one should not be surprised that they differ phenotypically as well. We do not fully understand why AMG176 gives inconsistent results across cell lines, but that point is peripheral to the fundamental observations of our manuscript. That loss of *ATAD1* sensitizes to dysfunction of the ubiquitin proteasome system (be it *MARCH5* deletion, proteasome inhibition, etc) is consistent across cell lines and is the key insight underlying our paper.

Referee #4 (Remarks to the Author):The revised manuscript by Winter et al. demonstrates a real attempt to address the issues I raised in my original review. As a result, many improvements have been made. However, there are still some major issues with the focus of the paper, and with the somewhat inconsistent use of various cell line/xenograft models as well as discrepancies with previous studies.

We appreciate the reviewer acknowledging our efforts and that our resubmission “demonstrates a real attempt to address issues.” We are grateful that the reviewer recognizes that “many improvements have been made.” We hope to address the reviewer’s remaining concerns below.

Firstly, I agree with the authors that the strength of the paper rests on the immediate therapeutic potential of proteasome inhibitors in ATAD1-deficient tumours and on the mechanistic basis of this sensitivity (and not on the contribution of PTEN). Thus, I understand why the authors have decided to focus on this particular topic. However, this also narrows the audience to whom this topic will appeal. Furthermore, there should have been an adjustment in the text to reflect the change in focus.Specifically, the manuscript’s title suggests that it is the co-deletion of ATAD1 and PTEN that drives the sensitive phenotype. But, would it not be more accurate to say, “ATAD1 loss sensitises cancer cells to protein stress”, or “ATAD1/PTEN co-deletion is associated with sensitivity to protein stress”? Ultimately, in the absence of any functional PTEN data, PTEN loss is nothing more than a less-than-perfect marker of ATAD1 loss (as I mentioned in my first review, collateral lethality itself is no longer a novel concept).

This is a helpful comment from someone with a fresh perspective on our project. We agree that the clearest and simplest title would be: “ATAD1 loss sensitizes cancer cells to protein stress.” The reason we did not choose this title or something like it is that we wanted to include the context of *ATAD1* being co-deleted with *PTEN*. Without knowing the connection between *ATAD1* and *PTEN*, we are concerned that it might be difficult to understand the implications of a phenotype associated with *ATAD1* loss. Perhaps it is sufficient that this information is conveyed in the abstract. If the reviewer and editor feels strongly that our manuscript would be better served by an alternate title, we will oblige.

With regards to issue #1, I commend the authors for having attempted to access clinical trial material to explore the predictive value of ATAD1 loss in the response to proteasome inhibitors. Just to clarify, the numbers I was referring to (40% in mCRPC and 10% in GBM) were for 10q23 loss, and not necessarily ATAD1/PTEN co-deletion. One thing that could still be informative (if the data could be accessible for the MIRAGE trial at least) is the response rates. I do agree that any positive signal that comes from only 7% of patients would likely get lost in the noise if we only look at OS.This is an excellent suggestion and we agree with the reviewer that response rates would be informative. Unfortunately, Dr. Roth, the lead investigator, has not published such information and has not been willing to provide it as unpublished data to our group.With regards to issue #2, It seems the authors may have missed the fact that the reference I provided was of a retrospective study done in advanced disease. My point still stands that part of their analysis is not consistent with previous observations in a similar patient population.

We agree with the reviewer that loss of *PTEN* alone appears to have a different effect in our analysis compared to in the study referenced, by De Bono and colleagues. The source of this discrepancy is not clear to us, but could derive from the fact that we assess *PTEN* status at the genomic level, while the other study conducted IHC on patient samples. It could be that there are some tumors that appear to be *PTEN*-positive by sequencing but who lack PTEN protein due to a deletion in an important non-coding region of the gene. We have observed this phenomenon in cell lines on CCLE and verified it using DepMap (e.g. C32 melanoma). Generally, the overall survival of patients in the cohort that we have used is longer than that of the patients in the study by De Bono and colleagues, so there may be an underlying difference in patient demographics. However, the most important point of our analysis is that patients whose tumors lack *PTEN*/*ATAD1* appear to survive longer than their counterparts with only *PTEN*-null tumors. Therefore, we suggest that (i) any irregularity of *PTEN* status is controlled for, since we are comparing +/– *ATAD1* in the background of *PTEN* deficiency, (ii) even if our analysis somehow *overestimates* the overall survival of the “*PTEN*-only” cohort, that would act against the hypothesis we are testing (making it appear as though there is less of a difference between *PTEN* and *PTEN*/*ATAD1*).

With regards to issue #3, the authors make use of the gastric cancer cell line HGC27 and provide justification in their rebuttal letter stating that ATAD1 is deleted in 4.1% of gastric cancer cases. It would be best if an analysis similar to what’s presented in Figure 1B were done for gastric cancer to make the relevance of the cell line choice clearer to readers.

We thank the reviewer for this helpful suggestion. We have now replaced the plot showing PTEN/ATAD1 status of lung squamous cell carcinoma with one showing that of stomach adenocarcinoma in Figure 1B, as requested.

Also, as I mentioned in my first review, LNCaP cells are characterised as having deep deletions in both the ATAD1 and PTEN loci in CCLE. It would have been simple to just run qPCR\western blots to check for expression and make an empirical determination.

We thank the reviewer for pointing out this oversight and regret not previously including a western blot showing ATAD1 status in different cell lines (namely PC3 and LNCaP). See Author response image 11. Clearly LNCaP and PC3 cells both maintain ATAD1 protein levels, as predicted by DepMap (but in disagreement with CCLE). These cell lines both have absent PTEN protein. We used a knockout-verified monoclonal antibody for ATAD1, as in the rest of the manuscript, and included the control cell line SW1088, which lacks ATAD1.

**Author response image 11. sa2fig11:** 

What is also concerning about the choice of HGC27 (this is relevant to issue #4) is that according to DEPMAP (which is the resource the authors used to rule out LNCaP as a useful model), HGC27 cells are completely insensitive to MARCH5 depletion.

To ensure that we are referring to the same data as the reviewer, we included a plot from DepMap in Author response image 12 and Author response image 13:

https://depmap.org/portal/interactive/?filter=&regressionLine=true&associationTable=false&x=slice%2Fexpression%2F1697%2 Fentity_id&y=slice%2FChronos_Achilles%2F17328%2Fentity_id&color.

**Author response image 12. sa2fig12:** 

**Author response image 13. sa2fig13:** 

HGC27 cells have a *MARCH5* gene effect of -0.86. For reference, -1 is defined as “the median value of all common essential genes” (DepMap). A value of -0.86 corresponds to a significant dependency. As a comparison, consider *PARP1* gene effect vs. *BRCA2* expression. Irrespective of *BRCA2* expression, the cell line with the highest magnitude dependency on *PARP1* has a value of only -0.76.

And the authors do not provide any evidence of MARCH5 depletion (either by qPCR or western) for the experiments shown in Figures 2D and 2E.

As requested, we now confirm the knockout efficiency of sgMARCH5 transduction by western blot. MARCH5 is the lower band, indicated by the arrow on the left-hand side, and not the nonspecific band, which is indicated by the asterisk, see Author response image 10.

The same lack of sensitivity to MARCH5 depletion in HGC27 cells is reported in DRIVE.

Project DRIVE is a compendium of RNAi screens, which are less sensitive than CRISPR screens and –although irrelevant to the current point– suffer from off-target effects. We think it could be the case that there would be cell lines that are insensitive to knockdown of gene X but sensitive to deletion of gene X. This seems to be the best explanation for the *MARCH5* data in the DRIVE resource.

In fact, there is a high number of cell lines (e.g. SNU1105 (a GBM line), LN235 (a GBM line), UMUC3) which according to CCLE (and DEPMAP) express vanishingly low levels of ATAD1 and are insensitive to MARCH5 depletion (based on DEPMAP data).

We acknowledge the reviewer’s point that there are examples of cell lines in DepMap that have low/zero ATAD1 expression and yet are insensitive to MARCH5 deletion. While on its face, that would appear to be inconsistent with synthetic lethality, however, we have found that most if not all examples of known synthetic lethal pairs have similar cases of resistant cell lines that do not fit the expected trend, when analyzed via DepMap. As a result, we think it is inappropriate to conclude that our findings on ATAD1/MARCH5 are incorrect or not representative.

With regards to issue #4 (and #3), the type of analysis (and/or data display) for the new CRISPR screen is different from what was originally done in the first version of the manuscript. In fact, ACOT11 no longer shows up, or at least is no longer highlighted in the data plots. Why was the HGC27 data not analysed in the exact same way the Jurkat data was analysed in the original version of the manuscript? (the initial Jurkat data was shown as dCS vs p-value, but in the revised version with the two screens, the data was shown as dCS in WT vs dCS in ATAD1delta).

We thank the reviewer for pointing out this change from the first manuscript. We originally displayed the Jurkat screen data as -log10(P-value) vs. dCS because we thought that was the most robust way to present the data from a screen where we had two biological replicates. With our HGC27 screen we did not have the same statistical power (as we used only one cell line as is the convention in the field), and our sgRNA library had only 4 sgRNAs per gene instead of 10, as in the library we used for our Jurkat screen. We therefore moved the previous Jurkat screen plot to the supplemental figure and instead showed the dCS WT vs. dCS ATAD1∆ in order to be consistent with our HGC27 figure. We have always been completely transparent with our screening data and have included all raw data used to make the figures.

We think the current figure format conveys the results in the most transparent and interpretable manner. It also emphasizes the two complementary screening approaches to show the robustness of the MARCH5/ATAD1 interaction.